# Experimental determination of partial charges with electron diffraction

Soheil Mahmoudi[1,2], Tim Gruene[1✉], Christian Schröder[3✉], Khalil D. Ferjaoui[4], Erik Fröjdh[4], Aldo Mozzanica[4], Kiyofumi Takaba[1], Anatoliy Volkov[5], Julian Maisriml[1,6], Vladimir Paunović[7,8], Jeroen A. van Bokhoven[7,8] & Bernhard K. Keppler[1]

Atomic partial charges, integral to understanding molecular structure, interactions and reactivity, remain an ambiguous concept lacking a precise quantum-mechanical definition[1,2]. The accurate determination of atomic partial charges has far-reaching implications in fields such as chemical synthesis, applied materials science and theoretical chemistry, to name a few[3]. They play essential parts in molecular dynamics simulations, which can act as a computational microscope for chemical processes[4]. Until now, no general experimental method has quantified the partial charges of individual atoms in a chemical compound. Here we introduce an experimental method that assigns partial charges based on crystal structure determination through electron diffraction, applicable to any crystalline compound. Seamlessly integrated into standard electron crystallography workflows, this approach requires no specialized software or advanced expertise. Furthermore, it is not limited to specific classes of compounds. The versatility of this method is demonstrated by its application to a wide array of compounds, including the antibiotic ciprofloxacin, the amino acids histidine and tyrosine, and the inorganic zeolite ZSM-5. We refer to this new concept as ionic scattering factors modelling. It fosters a more comprehensive and precise understanding of molecular structures, providing opportunities for applications across numerous fields in the chemical and materials sciences.

Properties of molecules and materials and chemical synthesis pathways require a comprehensive understanding of their chemical bonds. A deeper understanding leads to greater control over properties and chemical reactions. Weakening and cleavage of chemical bonds are the first steps to forming new bonds. Controlling these basic processes of chemical reactions is essential for engineering new materials and fine-tuning their structural properties, a highly active area of research[5–8]. Electronegativity and atomic charge are cornerstone concepts for rationalizing trends in charge transfer, bond polarity, bond strength, reactivity and various chemical properties[9]. Yet, until now, there is no generally applicable experiment that determines partial charges in chemical compounds. Here we present ionic scattering factors (iSFAC) modelling, an experimental method to assign partial charges to the atoms in molecules. iSFAC modelling is based on crystal structure analysis with three-dimensional (3D) electron diffraction. It determines partial charges at an absolute scale for every atom of the molecular compound. Our results indicate that iSFAC modelling is available with generally the same crystal quality as used for conventional single crystal structure determination. This method empowers chemists to determine the partial charges of all atoms of a chemical compound in its chemical environment.

The chemical bond is not a fixed property but depends on and can be modified by its chemical surroundings, forming the basis for all chemical reactions that involve the breaking of existing bonds and the formation of new bonds. The distribution of electrons in molecules affects bond strengths and reactivity, enabling predictions and modifications of bond behaviour in various chemical environments. The characterization of chemical bonds through crystallographic methods has traditionally been limited to instances with extremely high-resolution data of exceptionally high quality, enabling multipole refinement and analysis of bond topologies by, for example, Bader's quantum theory of atoms in molecules[10–12]. The required resolution and quality are rare, which limits the applicability of this analysis. X-ray diffraction is the primary technology for crystal structure analysis, contributing to more than 1.5 million structures in public databases, which continue to grow exponentially[13–15]. However, X-rays are relatively insensitive to fine electronic details. Unlike X-rays, electrons are charged particles and therefore interact with the charge distribution, that is, the electrostatic potential or Coulomb potential, of the crystal. Thus, electron diffraction provides intrinsic access to partial charges in molecular structures. In the case of radiation-hard materials, this possibility has been demonstrated with a technique called converged beam electron diffraction pioneered by Spence with the quantitative determination of the electrostatic potential of rock salt and germanium[16–18]. A recent study was published with five inorganic compounds—quartz, natrolite, borane, lutecium aluminium garnet and caesium lead bromide—based

[1]Department of Inorganic Chemistry, University of Vienna, Vienna, Austria. [2]Vienna Doctoral School in Chemistry (DoSChem), University of Vienna, Vienna, Austria. [3]Department of Computational Biological Chemistry, University of Vienna, Vienna, Austria. [4]Center for Photon Science, Paul Scherrer Institute, Villigen, Switzerland. [5]Department of Chemistry, Middle Tennessee State University, Murfreesboro, TN, USA. [6]Quantum Optics, Quantum Nanophysics and Quantum Information, University of Vienna, Vienna, Austria. [7]Institute for Chemical and Bioengineering, ETH Zurich, Zurich, Switzerland. [8]PSI Center for Energy and Environmental Sciences, Paul Scherrer Institute, Villigen, Switzerland. ✉e-mail: tim.gruene@univie.ac.at; christian.schroeder@univie.ac.at

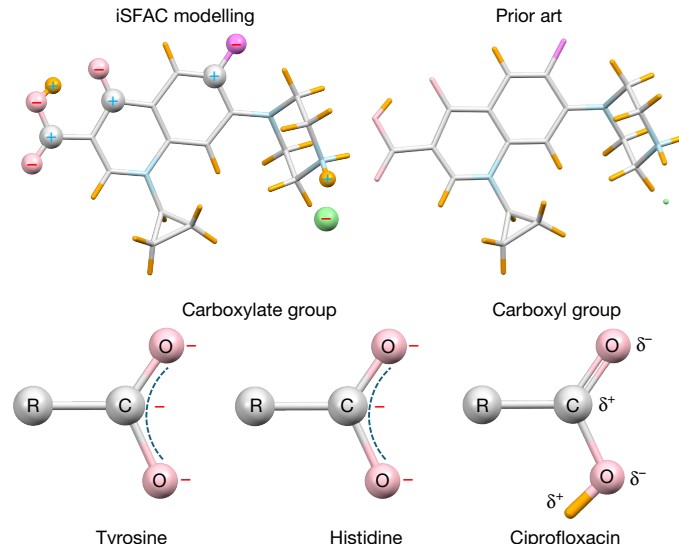

**iSFAC modelling** **Prior art**

**Carboxylate group** **Carboxyl group**

Tyrosine Histidine Ciprofloxacin

**Fig. 1 | With iSFAC, the chemical environment in the crystal structure matters.** The symbols '−' and '+' refer to negative and positive partial charge values of the respective atoms, respectively.

on $\kappa$ refinement[19]. For macromolecules, qualitative discrimination between oxidation states of metals in enzymes was reported[20,21].

In this study, we introduce a broadly applicable method for the experimental determination of partial charges, named iSFAC modelling against crystallographic electron diffraction data. Our method is universally applicable to all classes of chemicals. iSFAC modelling provides absolute values for the partial charge of each atom of the crystallographic structure (Fig. 1). We present the results for the antibiotic ciprofloxacin, the amino acids tyrosine and histidine, as well as the inorganic structure of ZSM-5. For all three organic compounds presented in this study—ciprofloxacin, tyrosine and histidine—the values show a strong Pearson correlation of 0.8 or higher with quantum chemical computations (Supplementary Tables 12–29).

When the crystal structure of a chemical compound is determined, each atom is typically described by nine parameters: its three coordinates $x$, $y$, $z$, and six parameters for its thermal vibration, the atomic displacement parameters. Each atom contributes to each reflection of the crystallographic diffraction experiment. The scattering factor, which describes the contribution of each chemical element, is usually hard-coded into the refinement software and not refined. The iSFAC modelling method presented in this study introduces one additional parameter for each atom. The scattering factor for each atom is a combination of the theoretical scattering factor of the neutral atom and the theoretical scattering factor of its ionic form. One parameter, which describes the fraction of the ionic scattering factor, equivalent to its charge, is refined along with the conventional nine parameters. This parameter balances the contribution of the ionic form with the contribution of the neutral form for the atom (see section 'iSFAC modelling: quantification of partial charges' for the technical details). As the scattering factors are based on the Mott–Bethe formula[22], the resulting partial charges are on an absolute scale. There is one value for each individual atom in the structure, resulting in individual scattering factors for each atom (Fig. 2a,b). A welcome consequence of iSFAC modelling is its marked improvement in fitting models to diffraction data. For instance, it consistently improves the fit of the chemical model with the observed reflection intensities (Extended Data Fig. 8), and it shows enhanced structural details, including the possibility to refine coordinates and ADP values of protons (see Extended Data Fig. 9a–f), which is rather associated with dynamical structure refinement[23].

To illustrate our results, we highlight the differences in the partial charge distribution between carboxylic acid groups and carboxylate

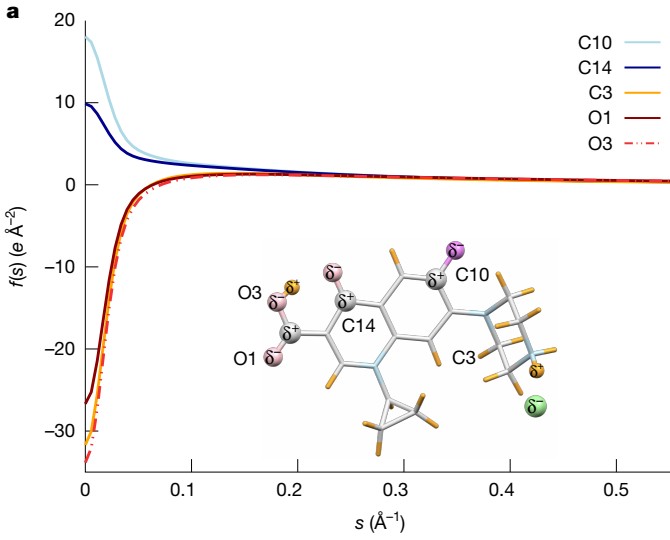

**a**

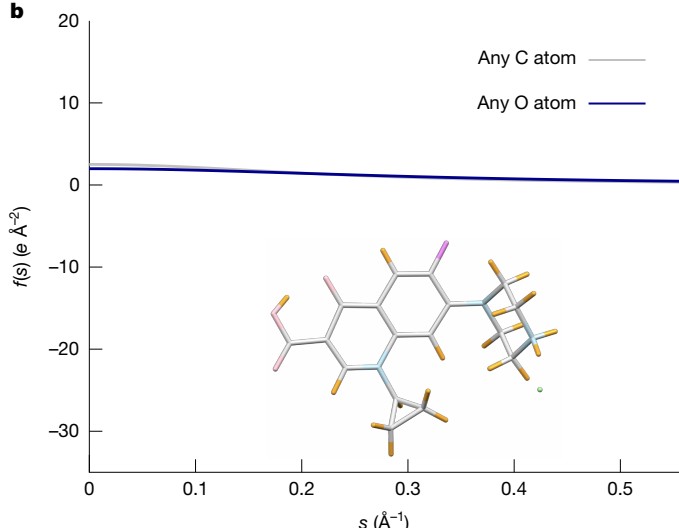

**b**

**Fig. 2 | Plot of atomic scattering factors $f(s)$ compared with data resolution $s$ (Å⁻¹). a**, iSFAC modelling refines individual scattering factors $f(s)$ for each atom. This enables the extraction of partial charges for each atom. **b**, Conventional refinement assigns one scattering factor per element type.

groups in ciprofloxacin and two zwitterionic amino acids, respectively. According to classical theory, the proximity of the amino group, which formally takes over the proton by a free electron pair, leads to the formation of a zwitterion. Hence, there is an excess of negative charge on two oxygen atoms that delocalizes over the carbon atom, making its partial charge negative. Both amino acids include a carboxylate group. Consequently, the carbon atoms C9 (−0.19$e$) in tyrosine and C6 (−0.25$e$) in histidine carry a negative partial charge. This seems counterintuitive for chemists but plausible considering the delocalized electron of the COO⁻ group. The crystal structure of ciprofloxacin includes a carboxylic acid group COOH. There is no delocalized electron, but a well-defined C18=O1 double bond and a C18–O3 single bond. Consequently, C18 carries a positive partial charge (+0.11$e$) (Figs. 3 and 4).

## Ciprofloxacin

Ciprofloxacin, a second-generation fluoroquinolone, exhibits marked antimicrobial activity and favourable pharmacokinetic properties[24]. In this study, the hydrochloride salt was crystallized in EtOH/MeCN, featuring a protonation NH₂ on the piperazine ring, and a carboxyl group (−COOH). The proton H3 of the carboxyl group forms an intramolecular hydrogen bond with the oxygen atom of the ketone group

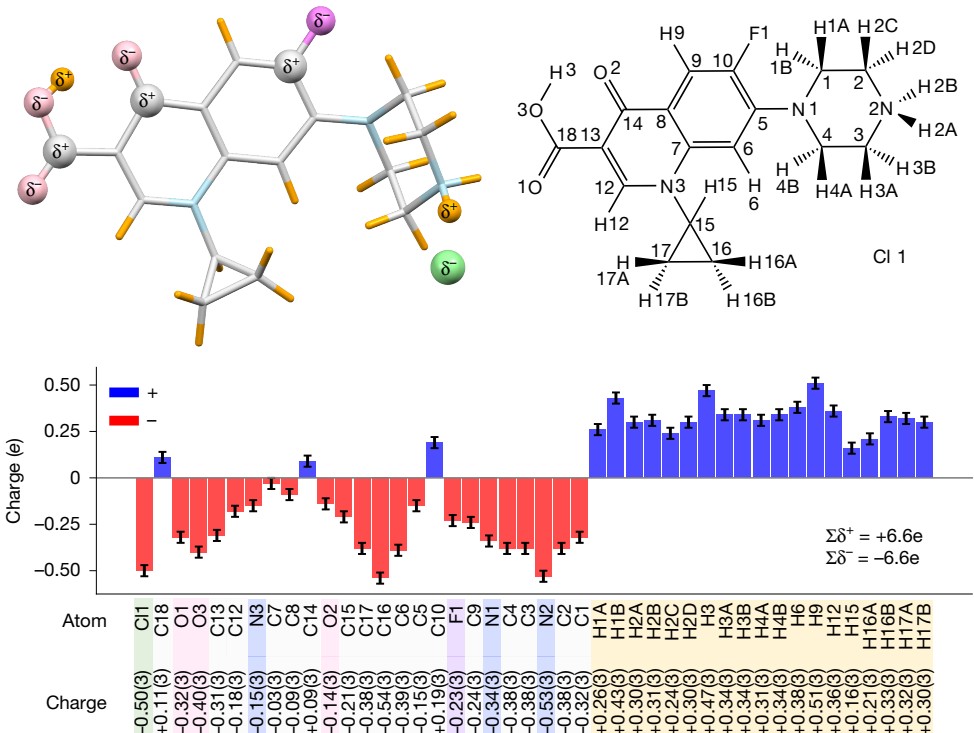

**Fig. 3 | Experimental partial charges for ciprofloxacin hydrochloride.** The numbers in brackets denote the standard uncertainties of the last significant digit. For example, −0.50(3) for Cl1 means −0.50e ± 0.03e.

C14=O2. Figure 3 shows the molecule and atomic partial charges and highlights several results of the iSFAC modelling. As we would expect, all hydrogen atoms show positive charges. These partially positive charges balance the negative ones of nearly all non-hydrogen atoms, including the Cl⁻ counterion. Only three carbon atoms exhibit a positive partial charge—namely, C18 as part of the carboxyl group, C14 with a double bond to O2 and C10 bonded to F1. The overall charge of $NH_2$ on the piperazine ring is positive, which makes a potent group to form a hydrogen bond with Cl⁻ counterion. This explains the strong negative charge on the N2 atom.

## Amino acids

The amino acids histidine and tyrosine are frequently used in electron diffraction studies because of their notable radiation stability and availability, allowing for direct use from the bottle[25,26]. Figure 4 shows their experimental partial charges. Both histidine and tyrosine are zwitterionic structures, with positive and negative charges on their amine and carboxylate groups, respectively. This affects their chemical properties and interactions within crystal structures.

**Tyrosine.** The nitrogen atom N1 within the amine group carries a relatively high negative charge of −0.46e. This may seem surprising. Yet, N1 is associated with the protons H1A, H1B and H1C with positive partial charges of +0.39e, +0.32e and +0.19e, respectively. The amine group as a whole is thus positive and a potent proton donor, as seen in intermolecular N1−H1A⋯O3, N1−H1B⋯O2, and N1−H1C⋯O3 hydrogen bonds (Extended Data Fig. 9g). This distribution results in an electron-rich carboxylate group with a total negative charge of −0.80e. The oxygen atoms are more electronegative than the carbon atom, resulting in a greater negative charge on the oxygen atoms compared with the carbon atom (−0.29e and −0.21e for O1 and O3, respectively, compared with −0.16e for C9). As discussed above, the delocalized electrons result in the observation that the absolute value for the partial charge of C9 is negative. Moreover, the hydroxyl group (−OH) within the phenol group functions as both a hydrogen bond acceptor O2⋯H1B−N1 and donor

O2−H2A⋯O1, with O2 bearing a charge of −0.27e and H2A bearing a charge of +0.29e.

**Histidine.** Our experimental data show that the N1 atom within the imidazole ring is neutral (+0.02e ± 0.04e). The O1 atom within the carboxylate group exhibits a strong negative charge of −0.31e, correlating well with tyrosine. This negative charge facilitates the formation of two intermolecular hydrogen bonds N3−H3A⋯O1 and N3−H3B⋯O1. By contrast, O2 exhibits a partial charge of −0.18e that is associated with the N1−H1⋯O2 hydrogen bond. This polymorph has the space group $P2_12_12_1$. We also collected data from a different polymorph with space group $P2_1$ and a c-axis half the length of the first polymorph. The chemical environment in this polymorph is similar, but not identical. Consequently, the partial charges are similar, but not identical (Extended Data Fig. 2). This observation highlights the sensitivity of iSFAC modelling not only to the molecule itself but also to its chemical environment.

## A new role for hydrogen atoms in crystal structures

Usually in crystallography, hydrogen atoms are treated differently than non-hydrogen atoms, because their contribution to the data is relatively weak. Their positions are calculated from the bonding geometry, and the ADP values are equal to the ADP value of the bonded heavier atom, multiplied by a fixed number (for example, the ADP value for the hydrogen atom in an N−H bond would be 1.2× the ADP value of the nitrogen atom). When this crystallographic habit is combined with iSFAC modelling, it renders the resulting partial charges of the non-hydrogen atoms meaningless. Including the charge of the proton is crucial in iSFAC modelling (see sections 'Comparison with quantum-mechanical calculations' and 'iSFAC modelling: quantification of partial charges'). This treatment frees hydrogen atoms from their geometric constraints and enables their free refinement, similar to any non-hydrogen atom. When needed, for example, when data resolution is not quite atomic, mild restraints can be applied[27]. To explain the consequences of this emancipation of hydrogen atoms, we discuss the observation of two different configurations of the intermolecular N1−H1⋯O2 hydrogen bond revealed by

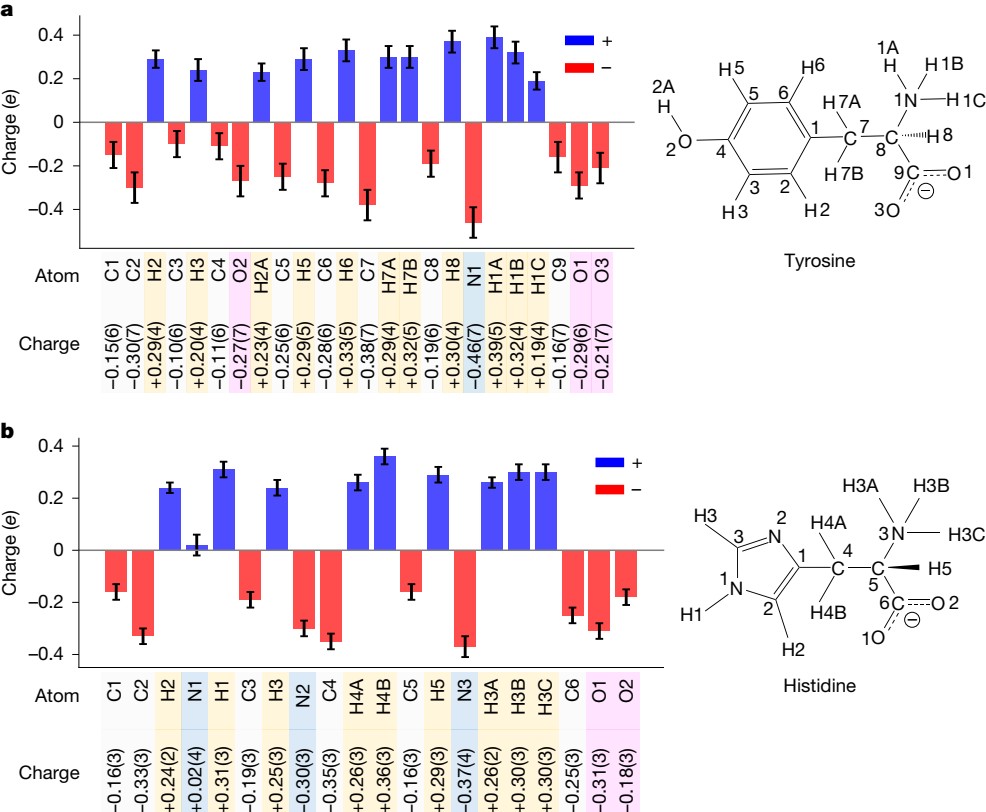

**Fig. 4 | Experimental partial charges for amino acids. a,b,** Experimental partial charges for tyrosine (**a**) and histidine (**b**). Element types are grouped by colour boxes: oxygen (light red), nitrogen (blue), hydrogen (pale yellow) and carbon (white).

iSFAC modelling: For histidine, the experimental partial charges O1 and O2 in the carboxylate group varied among different crystals (Supplementary Table 5). In dataset DS5, the strong negative charges on O2 ($-0.24e$) and N1 ($-0.13e$) are associated with an N1–H1···O2 angle of $175° \pm 2°$, an expected rather straight hydrogen bond. In crystal dataset DS1, with slightly positive charges on O2 ($+0.10e$) and N1 ($+0.12e$), the hydrogen bond angle is $155° \pm 2°$ (Fig. 5 and Supplementary Table 6). We investigated these dimer configurations, characterized by an N1–H1···O2 hydrogen bond, with quantum-mechanical calculations. The analysis included four distinct configurations: two experimentally derived configurations from datasets DS1 and DS5; one configuration based on the conventional riding atom model (DS1_HFIX); and one configuration 'OPT' obtained after geometrical optimization with Gaussian16 (ref. 28). As shown in Fig. 5, the straight configuration with DS5 has a relative energy of 69 kcal mol$^{-1}$ and is 59 kcal mol$^{-1}$ more stable than DS1 with a relative energy of 128 kcal mol$^{-1}$. The placement of H1 in the riding atom position increases the relative energy to 155 kcal mol$^{-1}$, suggesting that the bending of H1 towards O2 in the N1–H1···O2 hydrogen bond configuration contributes to a more stable bond between the histidines in DS1. Although the optimized dimer OPT has the lowest energy, its two histidines differ from each other. The bottom histidine is a zwitterionic molecule. The protons H3A, H3B and H3C are bonded to the nitrogen, resulting in a positive charge of the NH$_3$ group compensated by the negative charge of the carboxylate group. By contrast, in the top histidine of the optimized structure, the hydrogen H3A has moved to the carboxyl group, yielding a non-zwitterionic configuration with an NH$_2$ and a COOH moiety, which is not observed in any of our crystal structures. This shows the depth of the insight that iSFAC modelling provides into the interaction between a molecule and its chemical environment.

### Zeolite ZSM-5

Zeolite ZSM-5 was chosen to illustrate that iSFAC modelling also applies to inorganic compounds. Zeolites are aluminosilicate structures forming a great variety of different frameworks[29]. Zeolite ZSM-5 has an MFI-type framework in space group *Pnma* and is composed of 12 independent T-sites and 26 oxygen atoms. The unit cell is composed of (T$_{12}$O$_{24}$)$_8$ because four oxygen atoms occupy special positions with half occupancy (Extended Data Fig. 4b). The T-sites are mainly occupied by Si$^{IV}$ and can be replaced with Al$^{III}$, creating a charge imbalance that affects their catalytic activity. Our experimental results reflect the expectations, in that the T-sites are expected to have a positive partial charge whereas the oxygen atoms are expected to be negatively charged (Fig. 6).

### Robustness and resolution dependence

Parameters derived from crystallographic modelling can vary with the resolution of the dataset—equivalent to the number of experimental data points—or they can depend on the quality of the crystals. Certain variations are expected with any experimental data, and iSFAC modelling is no exception. Figure 6 is an example, showing slight variations of partial charges between data sets of zeolite ZSM-5. Comparison of partial charges derived from a series of resolution cut-offs shows a systematic trend. At resolutions worse than 1 Å, deviations become visible as shown in Extended Data Fig. 3. This limit coincides with the limit known as Sheldrick's rule[30]. The largest variations are accompanied by geometric irregularities, mainly for protons. Mild restraints to keep these atoms at their original positions recover the reliability of partial charge (Extended Data Fig. 3e,f).

As the charge information of scattering factors of ions is the greatest at low resolution (at $s \to 0$ Å$^{-1}$; Fig. 2), we need to ensure high data completeness at low resolution. In case the detector cannot capture the entire resolution range, the high-resolution pass should be followed by a low-resolution pass, as is common practice for good data collection[31]. This also applies to the temperature of data collection, as cryogenic temperatures generally improve data quality and protect better against radiation damage. In case of the zeolite ZSM-5, we collected data both

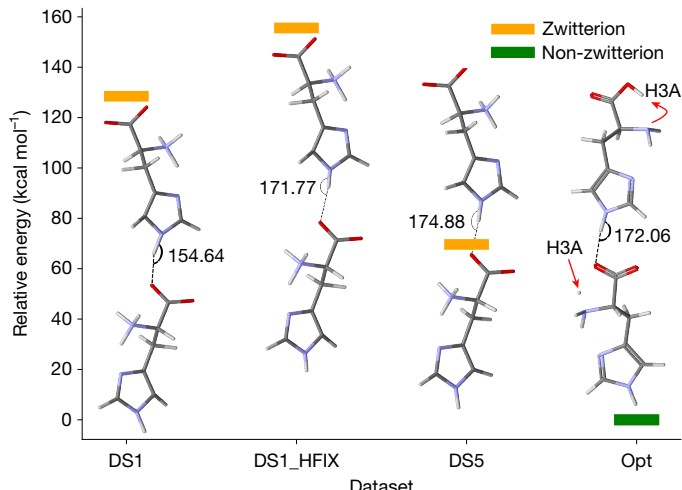

**Fig. 5 | Comparison of computed energies for histidine conformations.** Energy for four different possible configurations of the N1–H1···O2 hydrogen bond calculated with DFT. DS1 and DS5 are observed in the crystal structure of histidine. DS1_HFIX has H1 fixed at N1 in the riding atom position, and an optimized configuration of the dimer (opt).

at room temperature and at −110 °C. Except for the expected variation and reduction in estimated standard uncertainties, there is no notable trend for the partial charges (Extended Data Fig. 4a).

Another indicator for the stability of iSFAC refinement results from the computation of $R_{complete}$ (ref. 32), in which each reflection, in turn, is omitted from the refinement (leave-one-out cross-validation). In case of histidine, this results in monitoring the effect of each of its 1,710 data points on the values of the partial charges. Extended Data Fig. 3g shows $\pm 3\sigma$ boxes about the mean value $\langle \delta q \rangle$ for each atom, as well as the respective maximum and minimum values. The $\pm 3\sigma$ boxes are tiny, as the variance of the partial charge is minute. Next to it, Extended Data Fig. 3h shows the same type of analysis, not for the partial charge values but for the atomic displacement parameters $U_{ij}$ in case of a conventional refinement without iSFAC modelling. The spread of maximum and minimum outliers for $U_{ij}$ compares with those of the partial charge values, showing that missing certain reflections does not affect the reliability of iSFAC modelling beyond a standard crystallographic refinement.

For electron diffraction data, the dynamical theory of diffraction produces significantly more accurate calculated intensity values than the kinematical theory of diffraction[33], but only kinematical refinement is available with the program SHELXL, which was used for this study. The discrepancy between dynamical and kinematical refinement increases with sample thickness. Therefore, Extended Data Fig. 5 shows the partial charges of four individual histidine crystals with varying thickness. Although the absolute thickness of these crystals is unknown, the transparency to the underlying lacey carbon and the crystal thickness vary between these samples. Within the range of selected crystal size, the plot of partial charges in Extended Data Fig. 5a does not indicate systematic errors related to the crystal thickness. Instead, to reduce random errors, it is advisable to always merge data from several crystals[34].

## Reliance of iSFAC on electron diffraction over X-ray diffraction

We also tested whether iSFAC modelling would be suitable for X-ray data. We chose a very high-quality dataset of $Ca^{2+}$ tartrate, as well as a published structure of L-histidine with high-quality data (CCDC ID KAGGEI; ref. 35). Although with electron diffraction data, iSFAC modelling converges quickly to stable values for the partial charge, iSFAC modelling against X-ray data does not converge, leaving non-zero shifts of the parameters even after thousands of cycles, with partial charges

much greater than 1$e$, and negatively charged protons, neither of which is chemically plausible (Extended Data Fig. 6).

## Comparison with quantum-mechanical calculations

The electrostatic characterization inherent to our iSFAC methodology aligns closely with the results derived from quantum-mechanical computations. A rigorous quantitative analysis can be achieved by computing the Pearson correlation coefficient[36] for the experimental and computational partial charge distributions in the case of the amino acids and the ciprofloxacin. As the quantum-mechanical calculations may depend on the functional and basis set, several combinations have been tested (see section 'Computational approaches to determine partial charges'). On average, the various experimental datasets and the partial charges from the methods ADCH, CM5 and Natural Population Analysis (NPA), exhibit Pearson coefficients of 0.8 for ciprofloxacin, tyrosine and histidine—except for CM5 and NPA in the case of histidine. These high values underscore that our method for determining the partial charges coincides with quantum-mechanical calculations, consequently proving the validity of the iSFAC partial charges.

NPA and both Hirshfeld variants, ADCH and CM5 (refs. 1,37), determine the partial charges of the atoms based on the electron density (see section 'Computational approaches to determine partial charges'). As shown in Extended Data Fig. 1d for histidine, the high Pearson coefficients extend to most electron-density-based approaches (left group, methods Mulliken to Becke, labelled 'Electron Density'), with the exceptions of the atoms-in-molecule (AIM) method and the Becke approach. The comparatively lower efficacy of the AIM method may stem from its dual dependency on wave functions and the partitioning of the molecule into discrete atomic volumes. Conversely, methods that derive partial charges based on the electrostatic potential (middle group in Extended Data Fig. 1d, methods MK, CHELPG and RESP, labelled 'Electrostatic Potential') exhibit moderate performance, with average Pearson coefficients around 0.5, which might be due to grid artefacts[38]. Particularly, at the polar surface of the molecules, grid-based approaches are more sensitive to the local environment than electron-density-based methods. Also, conformational changes may cause problems for grid-based charges[1,39]. Charges derived from the electronegativity scale, specifically the PEOE (partial equalization of orbital electronegativity) method, manifest the lowest Pearson coefficients. In Extended Data Fig. 1d, the green boxes refer to the above analysis with iSFAC modelling that includes hydrogen atoms, as expressed in equation (3). Notably, omitting special iSFAC treatment for protons results in Pearson correlation coefficients falling below 0.2 with any computational method (dark red boxes in Extended Data Fig. 1d), highlighting the indispensable role of protons in iSFAC modelling (see section 'iSFAC modelling: quantification of partial charges'). This observation has led to the acknowledgement of the unique and important contributions of protons in iSFAC modelling. The poor performance of the partial charges without the inclusion of iSFAC treatment for hydrogens may be attributed to an imbalanced dataset, in which most of the non-hydrogen atoms exhibit negative charges. As a result, a linear regression focused solely on non-hydrogen partial charges is inherently more constrained compared with that which includes all partial charges. To address this hypothesis, the average charge of hydrogen atoms was calculated for the experimental partial charge sets not containing hydrogen atoms, effectively counterbalancing the charges of the non-hydrogen atoms. A Pearson correlation analysis was then re-conducted using an augmented dataset that included both hydrogen and non-hydrogen atoms, represented by the orange boxes in Extended Data Fig. 1d. Although incorporating the average hydrogen partial charges significantly improved the Pearson coefficients, the green boxes still demonstrated superior correlations, as indicated by higher average Pearson coefficients, reduced spreads of the corresponding boxes and lower standard deviations. The success of the average hydrogen partial charges (orange boxes) can be

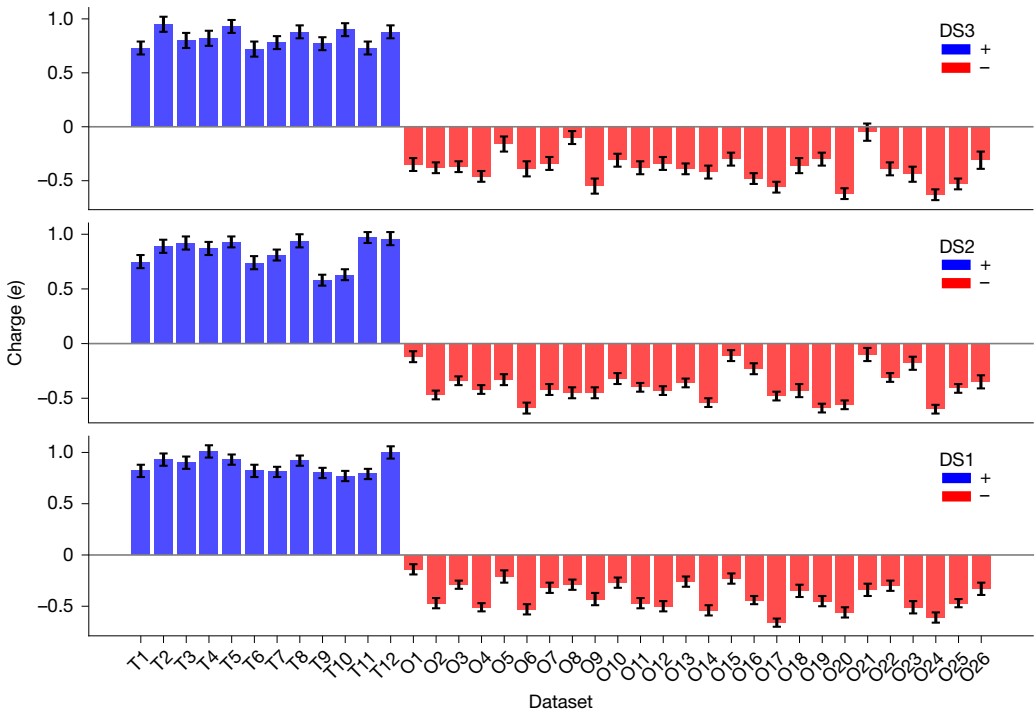

**Fig. 6 | Experimental partial charges for ZSM-5 across three different datasets (DS1, DS2 and DS3).** The red bars represent negative charges, and the blue bars denote positive charges.

attributed to the relatively narrow range of hydrogen partial charges ($0.1e \leq q_{\mathrm{H}} \leq 0.4e$). Consequently, the actual partial charges of hydrogen atoms do not deviate substantially from the computed average for all protons. However, the partial charges of non-hydrogen atoms are considerably influenced by whether or not protons are included in the analysis. Specifically, the average Pearson coefficient of non-hydrogen partial charges between these two partial charge analyses is 0.63 in histidine, underscoring that the iSFAC model, which incorporates protons, not only improves the accuracy of hydrogen charges but also enhances the reliability of partial charges for all other atoms.

Partial charges are the direct result of iSFAC modelling. Together with the phases computed from the crystallographic model, the Fourier transformation of the electron diffraction data produces a map that corresponds to the electrostatic potential map (ESP). Quantum chemical calculations calculate the electron density map first, and the partial charges are derived second (see section 'Computational approaches to determine partial charges'). Extended Data Fig. 10 juxtaposes experimental with computed ESPs for tyrosine and histidine. Both for tyrosine (Extended Data Fig. 10a,b) and histidine (Extended Data Fig. 10d,e), the carboxylate group shows positive patches induced by hydrogen bonds. This is also reflected by the hydroxyl group of tyrosine. The respective protons are highlighted in Extended Data Fig. 10c,f. The strong match of the experimental ESP from iSFAC modelling with the computational ESP shows with a Pearson correlation coefficient of 69.0% for tyrosine and 71.4% for histidine. Extended Data Fig. 10c,f shows the pseudo-crystal environment used for the calculated ESPs.

## Conclusions

This study presents iSFAC modelling as an innovative experimental methodology for determining the partial charges of atoms within chemical compounds based on 3D electron diffraction. With remarkable versatility, iSFAC modelling applies to any crystal structure resolved through electron diffraction at atomic resolution, spanning a broad range of compounds, including inorganic, bio-inorganic and organic molecules. What will be the ultimate limitation of the application of

iSFAC modelling, such as the size of accessible molecules, remains to be determined. As a standout feature, it requires neither exceptionally high-resolution data nor unusually high data quality, as demonstrated by the antibiotic ciprofloxacin. A marked advantage of iSFAC modelling is its ability to determine partial charges within the true chemical environment of the molecule, capturing essential chemical interactions rather than modelling the molecule as an isolated entity in a vacuum. Different crystal forms have different chemical environments, resulting in different partial charges. This provides a much deeper understanding of molecules and their functional groups than a conventional crystal structure. Furthermore, several quantum-mechanical methods align well with the experimentally derived partial charges, particularly those that are calculated based on electron density rather than electrostatic potential.

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

# Methods

## Chemical reagents

The chemical reagents used in the synthesis experiments include tyrosine ($C_9H_{11}NO_3$, 99.0 wt%, Sigma-Aldrich), histidine ($C_6H_9N_3O_2$, 99.0 wt%, Fluka), ciprofloxacin HCl ($C_{17}H_{18}FN_3O_3 \cdot HCl \cdot H_2O$, Merck), n-hexane ($C_6H_{14}$, 97.0 wt%, Honeywell) and distilled water. All chemicals were used without further purification.

## Sample preparation and data collection

Single crystals of ciprofloxacin hydrochloride were obtained through vapour diffusion of EtOH/MeCN. These crystals were then directly suspended by sonicating for several minutes. For the sample preparation of amino acids, a small amount of polycrystalline samples was delicately crushed between two microscope slides. The crushed material was then suspended in n-hexane and subjected to ultrasonication for approximately 1 min to achieve uniform dispersion.

Subsequently, a droplet of the resulting suspension was placed onto a TEM grid featuring a lacey carbon film (copper, Ted Pella, 200-mesh). Evaporation of n-hexane was observed with a light microscope. The grid was transferred to an ELSA698 tomography holder (Gatan) at room temperature, which was inserted into the JEM2100Plus TEM (200 keV LaB$_6$, JEOL), equipped with a 1,024 × 1,024 SINGLA 1M detector with an effective area of 1,028 × 1,062 pixels at 75 μm side length per pixel (DECTRIS). The SINGLA detector and the TEM were controlled with the graphical user interface singlaGUI, with source code available on GitHub[40]. Data were collected at room temperature or 163 K. The beam current was confined with a 50-μm condenser lens aperture and spot size 5. This corresponds to a current of about 20 pA. The sample was illuminated with a beam diameter of about 2.2 μm. Data were collected at 1.0° s$^{-1}$ and an effective readout rate of 10 Hz (0.1° per frame). The rotation range was up to 140°. In some cases, two or three datasets, with effective detector distances of 860 nm, 1,030 nm and 1,370 mm, were obtained to capture low- and high-resolution data for the samples.

ZSM-5 data were acquired using a JUNGFRAU detector with an effective area of 1,030 × 514 pixels at 75 μm side length per pixel, with the TEM operated by the epoc-GUI graphical interface (source code available on GitHub; ref. 41). Data collection took place at either room temperature or 163 K. A 50-μm condenser lens aperture and spot size 4 were used to confine the beam current. The sample was illuminated with a beam diameter of about 2.2 μm. Data were recorded at a speed of 1.0° s$^{-1}$, with an effective readout rate of 20 Hz (0.05° per frame). The rotation range extended up to 140°. In some cases, two or three datasets were collected with effective detector distances of 665 mm and 1,025 mm to capture both low- and high-resolution data.

## Data processing and model refinement

Data were processed with XDS[42]. The detector distance for each dataset was refined during the CORRECT step of the XDS program to improve the accuracy of the unit cell parameters. The structures were solved by using SHELXT[43] for ab initio phasing, followed by refinement and model building using SHELXL and ShelXle, respectively[44,45]. Full-matrix least-squares refinement with SHELXL using the L.S. command was performed in all cases. Estimated standard uncertainties for all parameters, including the free variables for partial charge assignment, are printed in the SHELXL log file (.lst file) as well as the .mat file, which can be created with the MORE command with a negative parameter. In case of results being combined from multiple datasets, standard uncertainties were estimated as the weighted mean[46].

## Atomic and ionic electron scattering factors

iSFAC modelling of the amino acid structures offered a great revelation underpinning the physical difference between electron and X-ray diffraction. The iSFAC modelling of non-hydrogen atoms includes the electron that the hydrogen contributes to its donor atom. Consequently, as an essential aspect of iSFAC modelling and unlike all other atoms, hydrogen atoms are refined with the scattering factor of H$^+$ (see section 'iSFAC modelling: quantification of partial charges' for the technical details). This approach resulted in consistency with chemical intuition and with quantum-mechanical calculations, and at the same time bore some surprises, such as a negative partial charge of the C-atom in carboxylate groups.

Electron scattering factors in Cromer–Mann parametrization,

$$f(s) = c + \sum_{i=1}^{4} a_i \exp(-b_i s^2) \tag{1}$$

suitable for the SFAC command in SHELXL were generated by fitting the Cromer–Mann parameters $c$, $a_1$, $b_1$, $\cdots$, $a_4$, $b_4$ against the version of the Mott–Bethe formula published in ref. 22:

$$
\begin{aligned}
f(s) &= \frac{m_0 e^2}{8\pi\varepsilon_0 h^2} \frac{Z_0 + \Delta Z - f_X(s)}{s^2} \\
&= \frac{1}{8\pi^2 a_0} \frac{Z_0 + \Delta Z - f_X(s)}{s^2} \\
&= 0.02393366\text{Å}^{-1} \frac{Z_0 + \Delta Z - f_X(s)}{s^2}
\end{aligned}
\tag{2}
$$

In equation (2), $Z_0$ represents the atomic number of the neutral atom, $\Delta Z$ is the additional charge in case of ions, for example, for Si$^{IV}$: $Z_0 = 14$ and $\Delta Z = 4$, for C$^-$: $Z_0 = 6$ and $\Delta Z = -1$. The term $f_X(s)$ is the X-ray scattering factor of the respective atom (not ion), evaluated at the magnitude of the scattering vector **s** (ref. 47). The terms $m_0$, $e$, $\varepsilon$ and $h$ are the rest mass of the electron, the elementary charge, the dielectric permittivity in vacuum and Planck's constant, respectively. Equation (2) has the advantage of being suitable for all ions, without dependency on computed X-ray scattering factors for ions such as C$^-$ or N$^+$, which may not always be available. Examples of the scattering curves $f(s)$ are shown in Extended Data Fig. 7. Note the offset at high resolution between neutral and ionic scattering curves, which may contribute to the stability of iSFAC modelling. Originally, our first attempts with iSFAC modelling computed $f_e(s)$ with the classical Mott–Bethe formula, where the ionic charge is expressed by $f_X(s)$ of the ion instead through $\Delta Z$. This, however, did not produce plausible results. Modelling the oxygen of the hydroxyphenyl side chain of tyrosine with either the scattering factor for O$^-$ or O$^+$ results in stable refinement, which would mean a negative or a positive partial charge depending on the choice of the crystallographers (data not shown). It was only when scattering factors $f_e(s)$ were computed based on equation (2) that iSFAC modelling would unambiguously result in a stable value for the partial charge for all atoms in the model.

In the case of ionic scattering factors, the Cromer–Mann parametrization cannot match the entire resolution range because of the divergence at low resolution ($f(s) \rightarrow \pm\infty$ for $s \rightarrow 0$ Å$^{-1}$). To optimize the fit, the resolution range of the fit was restricted to the resolution range of the particular data sets, 15 Å–0.75 Å in most cases. Note that the Mott–Bethe formula (equation (2)) enables the parametrization of seemingly nonphysical charges, such as O$^+$, C$^+$ or C$^-$, making it suitable for the fitting of positive and negative partial charges for any atom. We used the Levenberg–Marquardt algorithm as implemented in the program GNUPLOT[48] for fitting. Future development may result in even better scattering factors, leading to better results from iSFAC modelling[49].

**iSFAC modelling for quantification of partial charges.** We used the program SHELXL for the implementation of iSFAC modelling[44]. Those who are familiar with the syntax of SHELXL, and thus other programs based on its syntax[50], can easily carry out the required steps. Crystallographic refinement compares the observed reflection intensities

$I_{obs}(hkl)$ with calculated intensities $I_{calc}(hkl)$. In the kinematical approximation, as used by SHELXL, $I_{calc}(hkl) \propto |F_{calc}(hkl)|^2$. SHELXL is based on the independent atom model, IAM, in which each atom contributes independently to $F_{calc}(hkl) = \sum_{atoms\,j} f_j(s)e^{-8\pi^2 U_j(s)}e^{-2\pi i\mathbf{hx}_j}$, where the sum runs over all atoms in the unit cell. We abbreviated $\mathbf{hx}_j = hx_j + ky_j + lz_j$. The subscript $j$ refers to the $j$th atom, $f_j(s)$ is the (element-specific) electron scattering factor of atom $j$, calculated with the Cromer–Mann parametrization as explained above. The scattering vector $\mathbf{s}$ can be computed from the Miller index $(hkl)$. $U_j(s)$ is the isotropic or anisotropic Debye–Waller term, and $(x_j, y_j, z_j)$ are the fractional coordinates of atom $j$. Conventionally, both the coordinates and ADP values $U_j(s)$ of an H atom are computed from the respective parameters of its geometric environment (AFIX command in SHELXL), rather than refined. iSFAC modelling modifies the conventional handling of H atoms twofold:

1. the partial charge for every non-H atom is computed using a linear superposition between neutral and ionic scattering factor[21] and
2. the partial charge for H atoms is computed from their ionic scattering factor.

In the course of our work, we realized that hydrogen atoms have to be included in iSFAC modelling in this special manner. This treatment showed that iSFAC modelling stabilizes H atoms, and their parameters can be freely refined without constraints (possibly supported by mild restraints). When hydrogen atoms are ignored, the resulting partial charges are not very meaningful (Extended Data Fig. 1d, dark red boxes, and section 'Comparison with quantum-mechanical calculations'). However, when hydrogen atoms are refined as a linear superposition, similar to all other atoms, the refinement does not converge and stops as unstable. Possibly, the scattering factor of $H^0$ is too weak for superposition with the scattering factor of $H^+$. Hence, iSFAC modelling is based on the expression

$$
\begin{aligned}
F_{calc}(hkl) = &\sum_{\text{non-H atoms}\,j} (v_j f_j^{ionic}(s) + (1-v_j)f_j^{neutral}(s)) \\
&e^{-8\pi^2 U_j(s)}e^{-2\pi i\mathbf{hx}_j} \\
&+ \sum_{\text{H atoms}\,k} v_k f^{H^+}(s)e^{-8\pi^2 U_k(s)}e^{-2\pi i\mathbf{hx}_k}
\end{aligned}
\tag{3}
$$

For each atom $j$, the fraction $v_j$ was refined together with its other parameters $\mathbf{x}_j$ and $U_j$. Note that each atom was refined individually, and no grouping or other kind of simplification was applied. In SHELXL, the fractions $v_j$ and $v_k$ are refined through its FVAR mechanism. Equal coordinates and ADPs were imposed with the commands EXYZ and EADP, respectively. This way, iSFAC modelling requires only one extra parameter, $v_j$ for each atom in the model.

The fraction $v_j$ of the ionic scattering factor for the $j$th atom in equation (3), and the respective charge offset $\Delta Z$ from equation (2) that was used to calculate its ionic scattering factor $f_j^{ionic}(s)$ define the partial charge of the $j$th atom as

$$
\delta q_j = v_j \Delta Z
\tag{4}
$$

It is noteworthy that refinement with SHELXL can continue only when all free variables fall within the range $0 \leq v_j \leq 1$. In case of negative values, the respective ionic scattering factor can be replaced with its opposite sign, for example, $C^-$ with $C^+$ for that particular $j$th atom. Our study illustrates this feature with the atoms C10, C14 and C18 for ciprofloxacin (Fig. 3). For values >1, the Mott–Bethe formula (equation (2)) allows for the computation of higher oxidation states. Our study shows this feature with the use of scattering factors for $Si^{IV}$ in the case of zeolites.

**Cation or anion robustness of assignment.** iSFAC modelling does not presume whether an atom is an anion or a cation. We tested both possibilities by replacing the respective ionic scattering factors. If the opposite type was modelled (cation instead of anion, or vice versa), the respective FVAR for the fraction $v_j$ would usually go negative. This was the case for the three carbon atoms $C_{10}$, $C_{14}$ and $C_{18}$ in ciprofloxacin. At first, these were modelled with a negative ionic scattering factor similar to all other non-hydrogen atoms. Refinement resulted in negative free variables, which showed the correctness of the opposite sign, that is, a positive partial charge. In some cases, an atom turned out to be neutral, fluctuating about $\pm 0e$. This renders the refinement with SHELXL unstable, as SHELXL refuses to start with a negative free variable (although it returns negative occupancy values once refinement has started). In these cases, the partial charge of the atom is set fixed to zero, and its FVAR is unused.

**Restraining the total charge using SUMP.** Physically, the total charge of the content of a unit cell should be neutral. This knowledge can be added to iSFAC modelling with the SHELXL SUMP command. SUMP is a relatively soft restraint, but in some cases, it reduces parameter shifts. With SUMP, the sum of all partial charges can be restrained to zero. For example, for the amino acid histidine, hydrogen atoms were refined with positive ionic scattering factors and non-hydrogen atoms were refined with negative partial charges modelled with a negative ionic scattering factor. Expressed with the free variable syntax of SHELXL, a neutral molecule satisfies

$$
0 = -1 \times fv(O_1) - 1 \times fv(N_1) \ldots + 1 \times fv(H_1) + \ldots + 1 \times fv(H_5)
\tag{5}
$$

In SHELXL, this condition can be expressed with the SUMP instruction SUMP 0 0.001 −1 2 −1 3 −1 4 −1 5 −1 6 −1 7 −1 8 −1 9 −1 10 −1 11 −1 12 1 13 1 14 1 15 1 16 1 17 1 18 1 19 1 20 1 21.

**Single crystal X-ray and ED data for Ca tartrate.** Tartrate crystals were collected from a glass of Frizzante (2022), vineyard Bioweinbau Thomas Berger, 2212 Großengersdorf, Austria. A crystal was isolated in NVH oil (cat. no. NVHO-1, Jena Bioscience) and mounted on a Bruker D8 diffractometer equipped with a CuKα source (Incoatec) and an EIGER R2 500 detector (DECTRIS). Data were collected with Apex v.3, integrated with XDS and scaled with SADABS[51]. The structure was solved with SHELXT and refined with SHELXL and the ShelXle GUI. Ionic scattering factors in Cromer–Mann parametrization were produced by fitting the Cromer–Mann function against theoretical X-ray scattering curves, both for the neutral and for ionic forms[47,52]. ED data from the same batch of crystals were collected at −110 °C, in the same way as the other ED data. Data from 12 crystals were merged to form the final dataset. The samples were very radiation sensitive (possibly because of the presence of four ordered water molecules). Owing to radiation damage, only about 30°–40° of the data were useful from each crystal. The partial charge for $Ca^{2+}$ was fluctuating around $0.10e$, in some cycles close to, but below $0e$, which prevents further processing. It was set fixed to $0.10e$, similar to the oxygen atom of the water molecule in residue 3 (Extended Data Fig. 6).

## Computational approaches to determine partial charges

Numerous methodologies exist for the computation of partial charges in molecular systems. These techniques predominantly rely on electron density[1,2,37,53–56], the electrostatic potential[2,38,53,57–59] or the electronegativity[60] as shown in Extended Data Fig. 1a–c. In the context of electron density-based methods, the partial charge $q_\beta$ of an atom $\beta$ is quantitatively defined by the equation

$$
q_\beta = Z_\beta - \int_{V_\beta} \rho(\mathbf{r})\, d\mathbf{r}
\tag{6}
$$

Here, $Z_\beta$ represents the charge of the nucleus, whereas the integral over $\rho(\mathbf{R})$ pertains to the electron density within a specific volume $V_\beta$. Challenges arise in the accurate delineation of this volume, as well as in the distribution of the electron density between bonded atoms. Various methodologies, including those by Mulliken[54], Löwdin[56], Hirshfeld[1,37] (and its successors CM5 and ADCH), Becke[61] and NPA[55], offer distinct approaches to addressing these issues. The Mulliken population analysis is an archetypal approach, leveraging overlap integrals and the electron density matrix to accomplish this task. Notably, the derived partial charges show a pronounced sensitivity to the choice of basis set. In response, Löwdin population analysis has been developed, using basis set orthogonalization to mitigate basis set dependency, thus enhancing the stability and reproducibility of molecular dipole moments.

The Hirshfeld population analysis represents a marked advancement towards basis set independence. This method deploys a stockholder partitioning mechanism, segregating the electron density predicated on the proportion an atom would contribute within a hypothetical overlay of isolated atomic charge densities. A notable limitation, however, is the dependence of Hirshfeld analysis on atomic reference states. It is worth mentioning that CM5 and ADCH are progressive iterations following the Hirshfeld scheme. Furthermore, NPA operates through natural atomic orbitals derived from a transformation of the molecular wave function. This method offers a more appropriate segmentation, resulting in charge distributions that resonate more intuitively with chemical principles. A caveat to consider with NPA is its elevated computational demand, alongside a potential residual basis set dependency.

Although also based on the electron density, AIM and Becke charges rely on the topology of the electron density rather than molecular orbitals or empirical rules. Becke charges are obtained from a fuzzy atom-centred division of electron density, thereby attenuating the influence of basis set choice. AIM[62] divides the molecule into atomic volumes, which allows for a charge determination in equation (6).

By contrast, methods using the electrostatic potential

$$V(\mathbf{r}) = \sum_\beta \frac{Z_\beta}{|\mathbf{r} - \mathbf{R}_\beta|} - \int \frac{\rho(\mathbf{r})}{|\mathbf{r} - \mathbf{R}_\beta|}\, d\mathbf{r} \qquad (7)$$

try to minimize the least-square between $V(\mathbf{r})$ and a charge-based electrostatic potential $V_q(\mathbf{r})$

$$V_q(\mathbf{r}) = \sum_\beta \frac{q_\beta}{|\mathbf{r} - \mathbf{R}_\beta|} \qquad (8)$$

These methods differ in determining points in space to compute the electrostatic potentials. The most popular methods are Merz–Kollman (MK)[58], CHELPG[57] and RESP[38]. CHELPG charges often provide values more in line with experimental observations, but the grid quality and placement may notably affect the results[63]. Alternatively, the RESP methodology incorporates supplementary constraints to enhance the fidelity with which molecular electrostatic attributes are replicated.

Beyond methods that rely on electron density and electrostatic potential, the PEOE model, introduced by Mar and others[60], uses electronegativity and ionic potential to distribute partial atomic charges in a molecule.

Quantum-mechanical computations were conducted using Gaussian16 (ref. 28) using the density functionals B3LYP (ref. 64) and $\omega$B97XD (ref. 65). The Pople basis sets 6-31G and 6-311G, along with the Karlsruhe basis set def2-tzvp, were used. We systematically explored all permutations of functionals and basis sets to assess basis set dependence. Geometry optimization started from crystallographic coordinates, and the presence of an energy minimum was confirmed through vibrational frequency analysis. For the amino acids, the zwitterionic state was explicitly enforced. The values for partial charges were derived from Gaussian check files using the MultiWFN 3.8 (ref. 66) software. Notably, except for Mulliken and Löwdin charges, minimal dependence on the selected functional or basis set was observed, aligning with expectations.

**Pearson coefficient.** The Pearson coefficient is the ratio between the covariance of two datasets (here, the computational and experimental partial charges of atoms within the molecule) and the product of their standard deviations. A Pearson coefficient of one signifies a perfect linear correlation between the datasets.

### Generation of electrostatic potential maps
The following steps were taken for the visualization of electrostatic potential maps for tyrosine and histidine, shown in Extended Data Fig. 10.

Experimental ESP maps in CUBE format were exported directly from ShelXle. Experimental maps are scaled to the value $F(s = 0)$, which is ill-defined for ionic scattering factors. Therefore, the value for $F(s = 0)$ from conventional refinement was used by the parameters REM OVERRIDEF000 acknowledged by ShelXle when exporting cube files, for example, the line REM OVERRIDEF000 121.5 was written into the SHELXL instruction file.

The crystallographic RES file was loaded into the program MERCURY (ref. 67). Next neighbours for the central molecule were hand-picked graphically. The resulting composition was exported from MERCURY as an XYZ file. ORCA v.6.0.1 was run in parallel mode from these coordinates with the following script:
- ! B3LYP 6-311G(d) CHELPG keepdensity
- %maxcore 18000
- %pal
- nprocs 20
- end
- * xyz 0 1
- ...
- *

where '...' was replaced by the content of the XYZ file. ORCA writes a GBW file. This was converted to an input file for MultiWfn with the command

```
#> orca_2mkl HIS_mrg_4orca -molden
```

where HIS_mrg_4orca is the basename of the previous ORCA input and output files. This produces the file HIS_mrg_4orca.molden.input, which was used as input to the program MULTIWFN v.3.8 (ref. 66) to compute the ESP and to write this ESP as file totesp.cub in CUBE format. We refer to this CUBE file as totesp.cub, which is the default output name of MULTIWFN. Note that the grid spacing in the QM-ESP and E-ESP cube files is not precisely the same. The Pearson correlation coefficients were computed with the program cubemaps, available on GitHub.

Images in Extended Data Fig. 10 were generated with VMD v.2.0.0a5 as follows. For tyrosine, the colour scaling was set between −0.06 and 0.15 for the E-ESP and between −0.19 and 0.40 for the QM-ESP map. For histidine, the values ranged between −0.04 and 0.09 for the E-ESP and 0.04 and 0.25 for the QM-ESP. The E-ESP maps and the QM-ESP maps were oriented manually to match the following:

First, to load the files, use: File → New Molecule → Browse.
1. Select the electrostatic potential cube file totesp.cub for the QM-ESP.
2. Select the respective cube file exported by ShelXle for the E-ESP, and also load as New Molecule.

Second, for representations, use:
1. Graphics → Representations → Create, Drawing method: CPK
2. Graphics → Representations → Create, Drawing method: Surf; with the following parameters:

- Colouring method: Volume
- Draw style: Solid surface
- Trajectory tab → Colour Scale Data Range: min–max
- Colouring method: Volume

For the QM-ESP totesp.cub, it is necessary to select the atoms of the central molecule only. For histidine, these are 20 atoms, INDEX 0 to 19, for tyrosine, these are 24 atoms, INDEX 0 to 23.

Third, for lighting, use:

1. Display → Light 1, Light 2 and Light 3
2. Display → Background → Gradient

**Note on simulation software.** We computed the ESP and partial charges using the programs ORCA and GAUSSIAN for single molecules, without taking crystallographic symmetry into account. For the ESP comparison of Extended Data Fig. 10, we mimicked crystal space with a next-neighbour approximation. This confirmed the experimental partially charged and the derived ESP. However, for systems that are electronically more complex than the amino acids and metal-organic compounds investigated here, for example, conducting inorganic materials, our approximation might have failed. During peer review, we were made aware of programs that take crystal symmetry into account (for example, NWChem, SIESTA or QUANTUM ESPRESSO) and are more suitable for modelling periodic crystalline systems. We believe future work involving more complex molecules should involve these tools.

## Software

The visualization tasks were performed using VMD[68], Mercury[67] and Platon[69].

Computational analyses were conducted using MULTIwfn[66,70], ORCA[71] with the B3LYP functional and the 6-311G(d) basis set and Gaussian[28], following the manuals of these programs.

## Data availability

The following data were used in this study. CIF files available through https://www.ccdc.cam.ac.uk/structures/?. CCDC histidine: 2365064, 2365068, 2365071, 2365077, 2365081, 2365065, 2364979, 2365082, 2365070, 2365066 and 2427850 (space group $P2_1$); CCDC tyrosine: 2365067, 2365033, 2365083, 2365069, 2365072, 2365166, 2365167, 2365161, 2365162 and 2365163; CCDC ciprofloxacin: 2365108; CSD ZSM-5: 2365164, 2365168, 2365169, 2365165 and 2365176; and CCDC Ca tartrate: 2427273 (X-ray diffraction) and 2427849 (electron diffraction).

## Code availability

The GNUPLOT scripts for the generation of SHELXL SFAC commands for both ions and neutral atoms are available at GitHub[72] (https://github.com/CF-CSA/edSFAC.git). The repository makes available the data from ref. 47, recomputed with increased s-spacing. Small Python programs for the creation of plots and figures are available at GitHub[73] (https://github.com/CF-CSA/iSFAC_code). This includes the program cubemaps to compute the Pearson correlation coefficient between two cube files. The epocGUI code for the graphical user interface data collection with the JUNGFRAU detector is available at GitHub[41] (https://github.com/epoc-ed/GUI). The SINGLAGUI code for the graphical user interface data collection with the SINGLA detector is available at GitHub[40] (https://github.com/tgruene/SinglaGUI.git).

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

**Acknowledgements** We are grateful to DECTRIS for the loan lending a SINGLA 1M detector. We thank J. H. Holstein for the discussion and for initiating the loan from DECTRIS. We thank R. Henderson, who triggered the idea of refining scattering factors for electron diffraction (Murnau Conference 2016). This research was funded in part by the Swiss National Science Foundation (SNSF) grant no. 215650 ('Measuring the electrostatic potential of complex chemical compounds and functional materials by electron diffraction'). This research was funded in part by the Austrian Science Fund (FWF) (I 6546-N; 'Electrostatic potential through charge-integration'; https://doi.org/10.55776/I6546). We are grateful to D. Kaiser for detailed suggestions on the manuscript. We thank S. van Terwingen and M. N. M. Milunovic for reading and discussing the manuscript. We thank the referees of our paper for their suggestions that markedly improved the manuscript between submission and acceptance. We dedicate this work in memoriam G. M. Sheldrick (1942–2025), who was often surprised by the unusual ideas of the users of his programs.

**Author contributions** S.M. contributed to the experimental design, data collection, data analysis and computational studies; T.G. helped with the experimental design and data analysis; C.S. did the computational studies; K.D.F. implemented the epocGUI user interface; E.F. provided the backend implementation for epocGUI user interface; A.M. developed the JUNGFRAU detector; K.T. implemented the epocGUI user interface; A.V. computed the (ionic)

X-ray scattering factors; J.M. did the programming of SINGLAGUI for data collection; V.P. synthesized the zeolite samples and discussed the results of zeolites; J.A.v.B. and B.K.K. discussed and validated the results; S.M., T.G. and C.S. wrote the manuscript. All authors contributed to the paper and discussed the meaning of our results.

**Funding** Open access funding provided by University of Vienna.

**Competing interests** The authors declare no competing interests.

**Additional information**

**Correspondence and requests for materials** should be addressed to Tim Gruene or Christian Schröder.

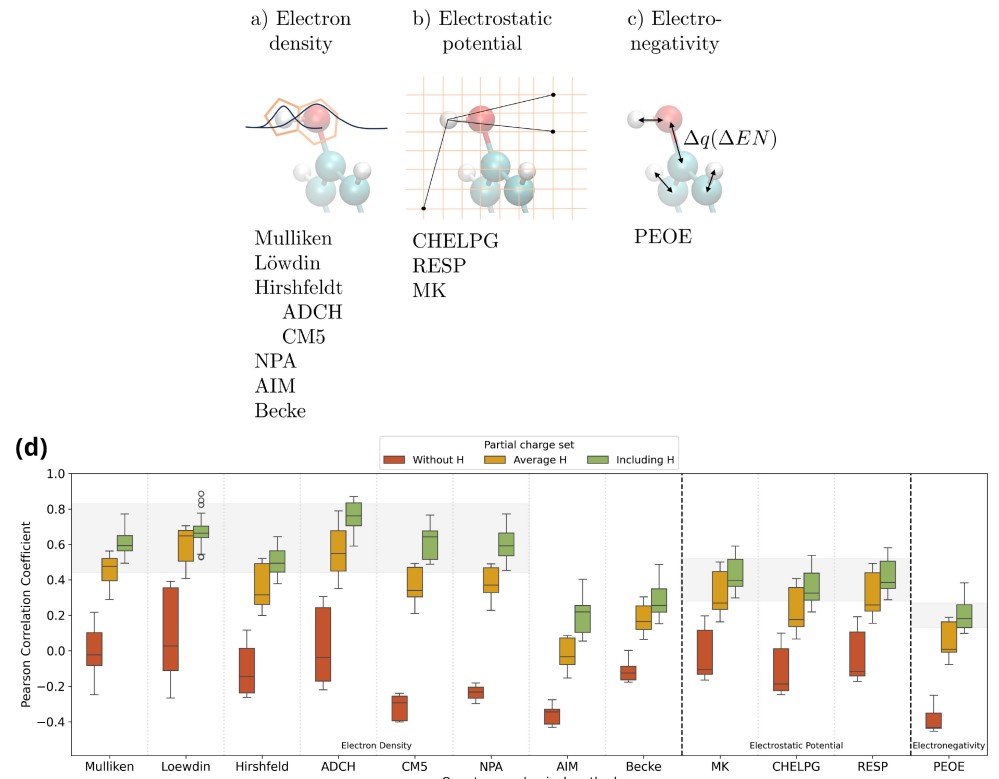

**Extended Data Fig. 1 | Comparison between iSFAC modelling and computational methods.** Most of the partial charges are either determined via a) electron density-based, b) electrostatic potential-based, or c) electronegativity-based approaches. (d) Pearson correlation coefficients of the individual experimental partial charge distributions in histidine with quantum-mechanical calculations using B3LYP or $\omega$B97XD with the basis sets 6-31G, 6-311G, and def2-tzvp. Plot grouped into electron-density-based, electrostatic-potential-based, and electronegativity-based approaches, respectively. Dark red boxes ignore the contribution of protons, only modelling partial charges for non-hydrogen atoms. Green boxes represent complete iSFAC modelling of partial charges for both hydrogen and non-hydrogen atoms based on Eq (3). Orange boxes are based on the values as for the dark red boxes, amended by one average value for the partial charge of each hydrogen atoms. See Sec. 1.7 for an in-depth discussion.

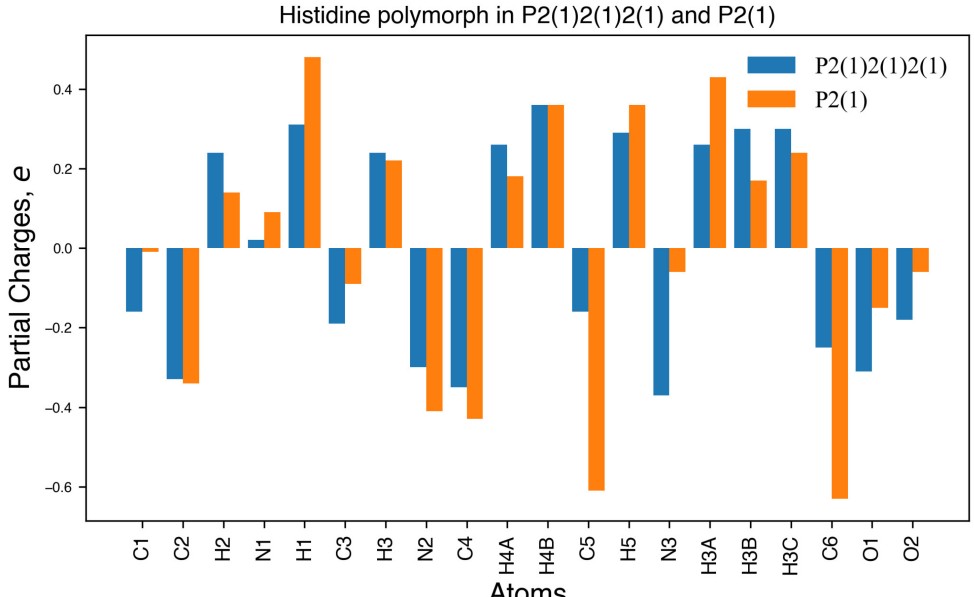

**Extended Data Fig. 2 | Juxtaposition of partial charges for L-histidine from two polymorphs.** Both polymorphs have similar crystal packing, but one is in spacegroup $P2_12_12_1$, the other one in spacegroup $P2_1$ with $\beta = 98°$.

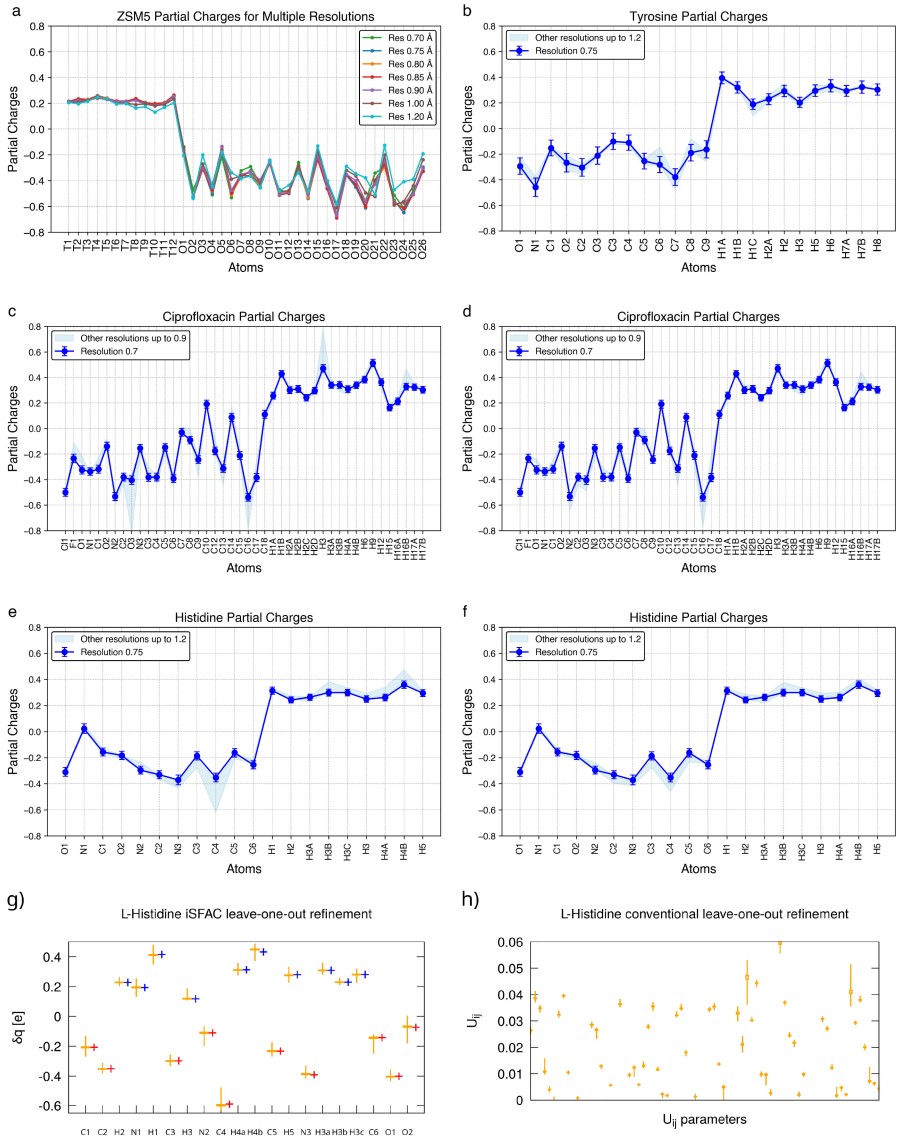

**Extended Data Fig. 3 | Robustness of iSFAC modelling.** (a)-(f): with respect to data resolution, (g)/(h): with respect to data completeness. (a) Resolution dependence of partial charges for ZSM-5. The high-resolution limit was systematically reduced from 0.7Å to 0.95 Å in steps of 0.05 Å, and to 1.00 Å and 1.20 Å. Results remain consistent up to 0.95 Å; deviations become noticeable only at 1.00 Å and above. (b) Resolution dependence of partial charges for tyrosine. The high-resolution limit was systematically reduced from 0.75Å to 1.20 Å in steps of 0.05 Å. (c) For ciprofloxacin, at resolutions worse than 0.85Å, deviations in the positions of O3 and H3 become apparent. These deviations arise from the instability of the H3 position when the resolution is worse than

0.85 Å. (d) Restraining the O3-H3 bond distance recovers the partial charge values close to the high-resolution data. (e) For histidine, at resolutions worse than 1.00Å, deviations in the positions of C4 and H4B become evident, affecting their partial charges. (f) Restraining the H4B-C4 distance recovers the partial charge values close to the high-resolution data. (g) Spread of partial charges with leave-one-out refinement. Crosses mark the values using all reflections (red: anionic, blue: cationic). (h) Spread of $U_{ij}$ values with leave-one-out refinement without iSFAC modelling, i.e. with conventional refinement. The comparison shows that the maxima and minima outliers are within expectations for crystallographic refinement.

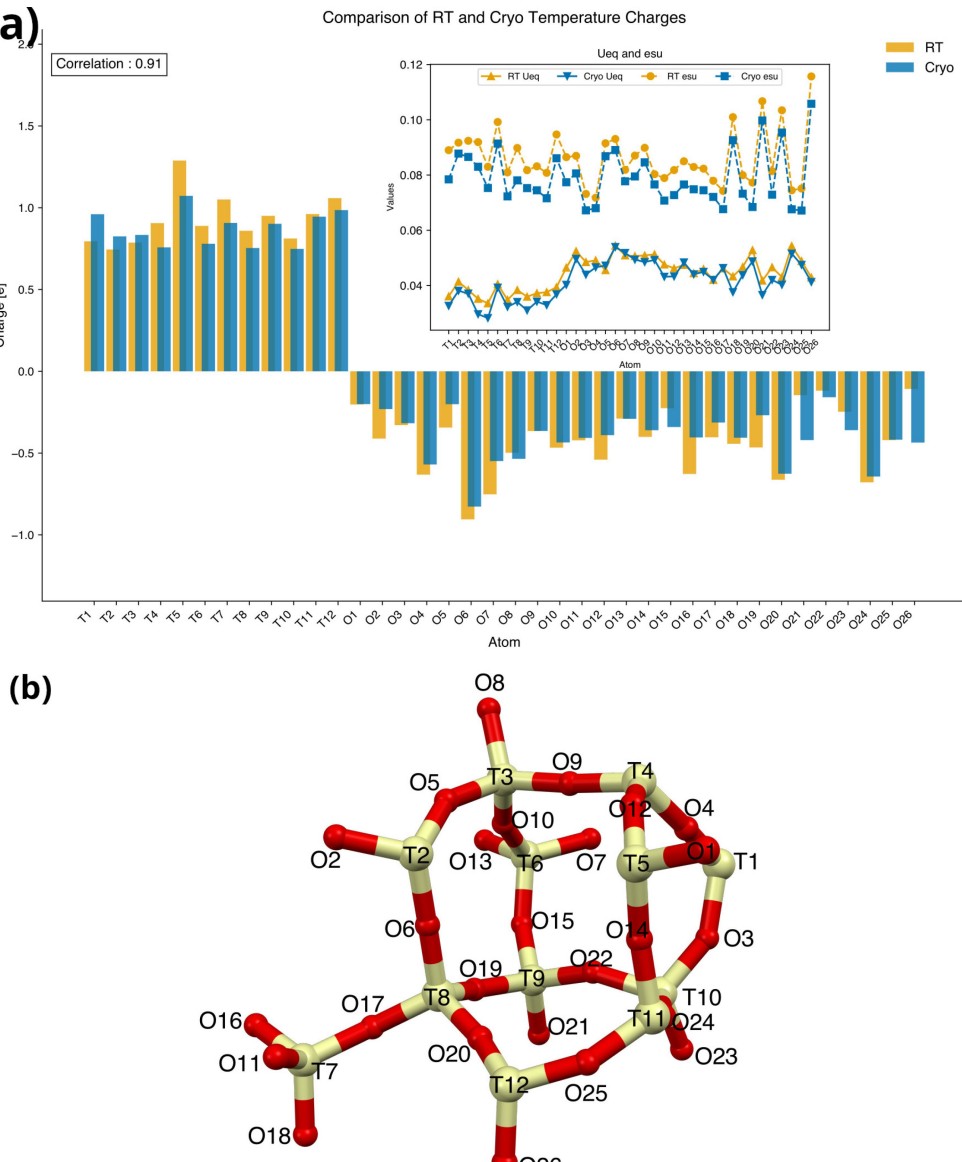

**Extended Data Fig. 4 | Partial charges for zeolite ZSM-5.** (a) Comparison of partial charges for zeolite ZSM-5 with data collected at room temperature (RT) and at −110 °C. Inset shows the estimated standard uncertainties for the partial charges, and the isotropic ADP values $U_{\mathrm{iso}}$. As expected, the errors are reduced for data collected at lower temperature. Otherwise, we do not observe any systematic trends. The Pearson correlation coefficient of 0.91, inset at the top left, compares the values for the low temperature and room temperature measurements. The correlation was computed with the moduli of the partial charges for the oxygens, becaue otherwise, the correlation would be overestimated (0.98). (b) Visual representation of the asymmetric unit of ZSM-5 zeolite, highlighting its unique framework structure characterized by 12 T-sites (tetrahedrally coordinated atoms) and 26 oxygen atoms.

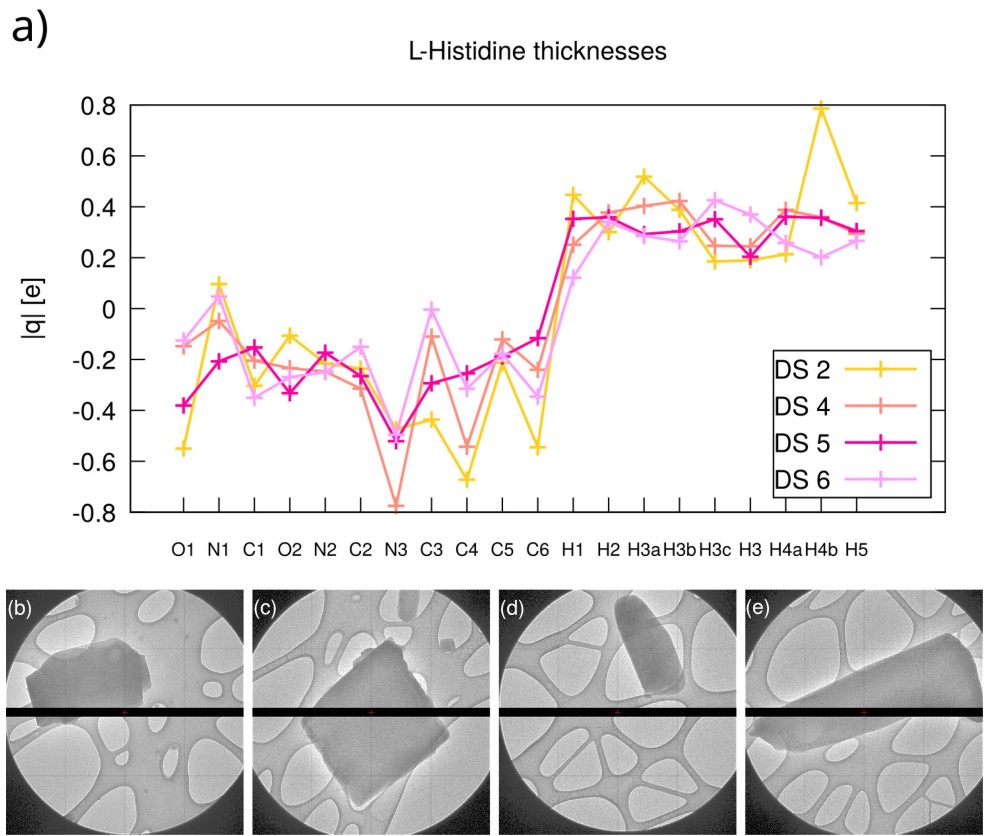

**Extended Data Fig. 5 | Variation of partial charges with crystal thicknes for L-histidine.** (a) partial charges $\delta q$ for four data sets of L-histidine of for different crystals. (b) picture of the crystal for DS2 (c) picture of the crystal for DS4 (d) picture of the crystal for DS5 (e) picture of the crystal for DS6 (c)-DS 4 and (e)-DS 6 are thickest, which the underlying lacey carbon hardly visible, (d)-DS 5 is thinnest.

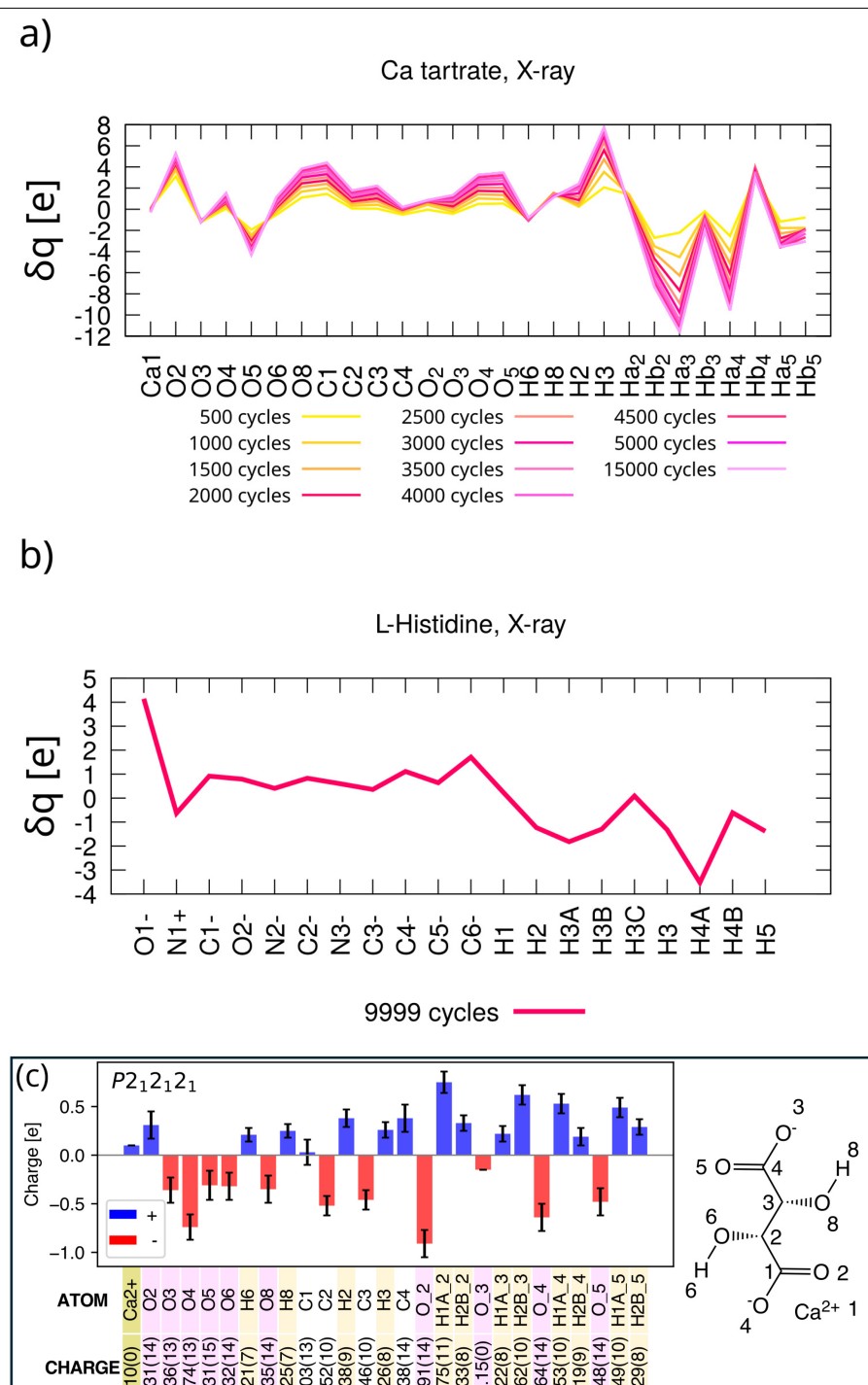

**Extended Data Fig. 6 | iSFAC modelling against X-ray data.** High quality data sets of (a) Ca tartrate and (b) L-histidine, CCDC ID KAGGEI, were refined with iSFAC modelling. In neither case refinement converges, even after thousands of cycles of least square refinement with SHELXL, there are still non-zero shifts. The resulting partial charges are outside physical plausibility $\gg 1$ and $\ll -1$, and protons with negative partial charges. (c) For comparison, iSFAC partial charges from the ED data.

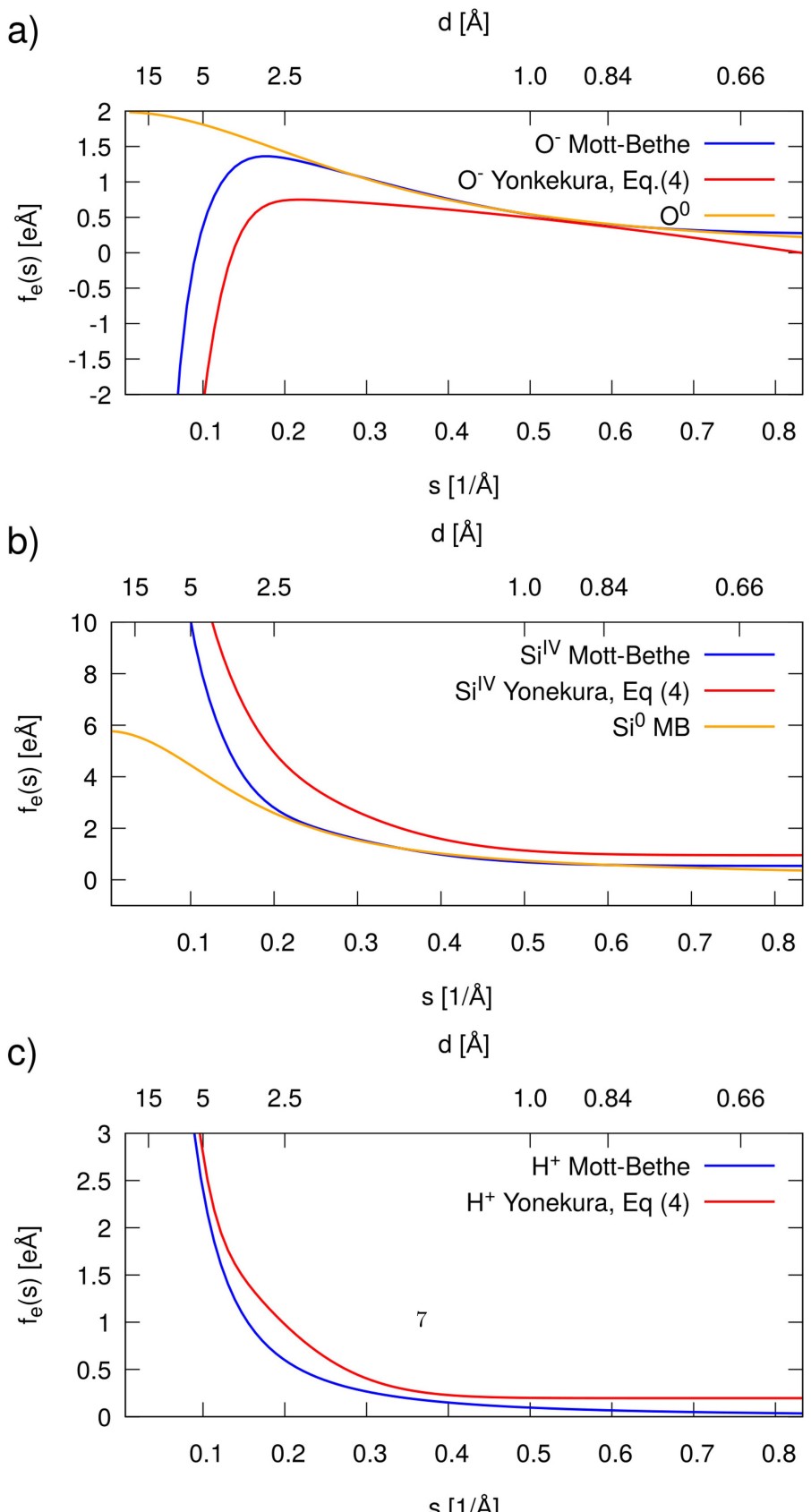

**Extended Data Fig. 7 | Comparison of electron scattering factors.** Electron scattering curves for some ions, illustrating the difference between Eq. (2), as published in[22], and the classical Mott-Bethe formula[74].

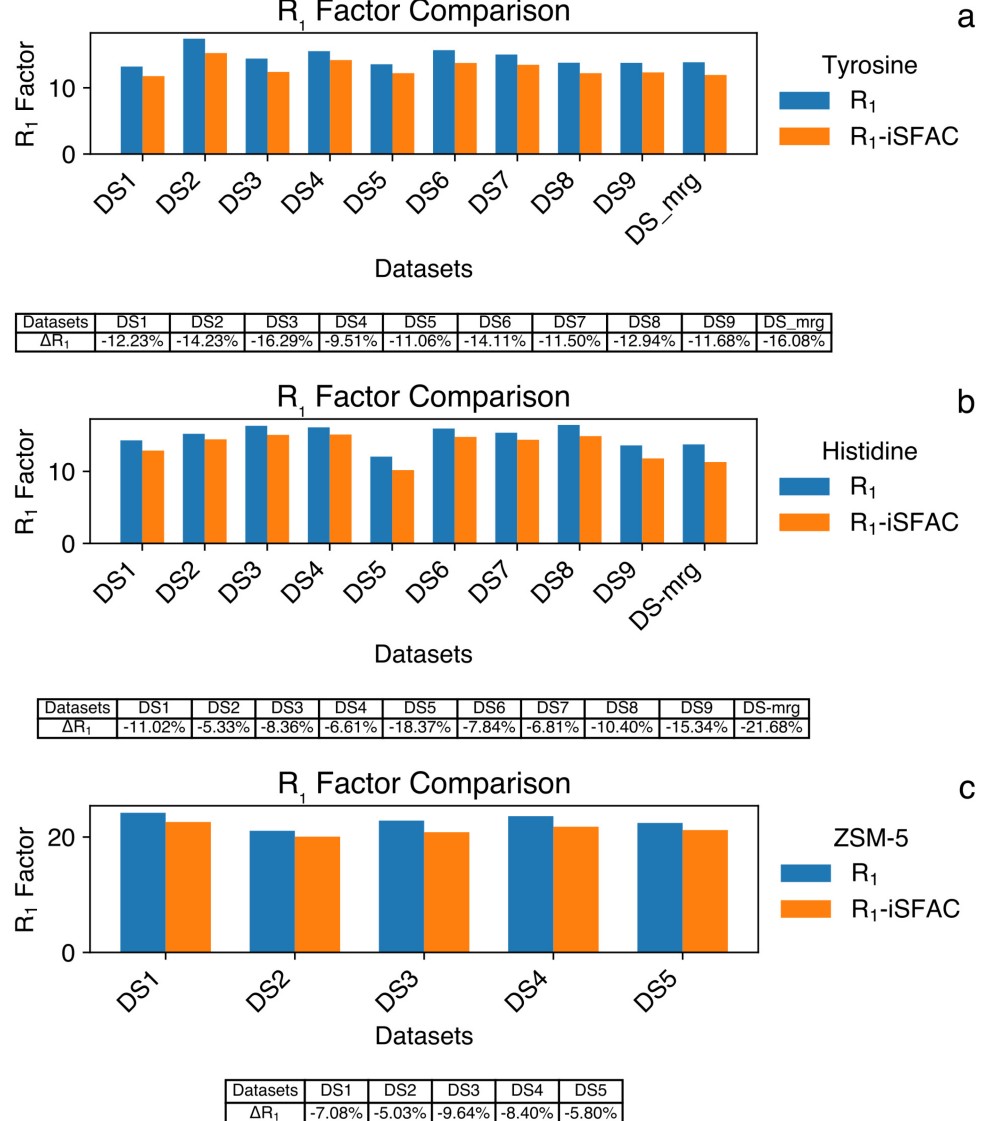

| Datasets | DS1 | DS2 | DS3 | DS4 | DS5 | DS6 | DS7 | DS8 | DS9 | DS_mrg |
|---|---|---|---|---|---|---|---|---|---|---|
| $\Delta R_1$ | -12.23% | -14.23% | -16.29% | -9.51% | -11.06% | -14.11% | -11.50% | -12.94% | -11.68% | -16.08% |

| Datasets | DS1 | DS2 | DS3 | DS4 | DS5 | DS6 | DS7 | DS8 | DS9 | DS-mrg |
|---|---|---|---|---|---|---|---|---|---|---|
| $\Delta R_1$ | -11.02% | -5.33% | -8.36% | -6.61% | -18.37% | -7.84% | -6.81% | -10.40% | -15.34% | -21.68% |

| Datasets | DS1 | DS2 | DS3 | DS4 | DS5 |
|---|---|---|---|---|---|
| $\Delta R_1$ | -7.08% | -5.03% | -9.64% | -8.40% | -5.80% |

**Extended Data Fig. 8 | Comparison of the $R_1$ factor between conventional and iSFAC modeling.** (a) Tyrosine: the upper section compares the $R_1$ factor between conventional and iSFAC models, while the table below presents the percentage change ($\Delta$) in $R_1$-iSFAC compared to conventional $R_1$. (b) Histidine: the top panel shows the $R_1$ factor comparison between conventional and iSFAC models, with the table below indicating the percentage change ($\Delta$) between the two. (c) ZSM5: similarly, the top panel compares the $R_1$ factor between conventional and iSFAC models, with the table underneath displaying the percentage change ($\Delta$) in $R_1$-iSFAC compared to $R_1$.

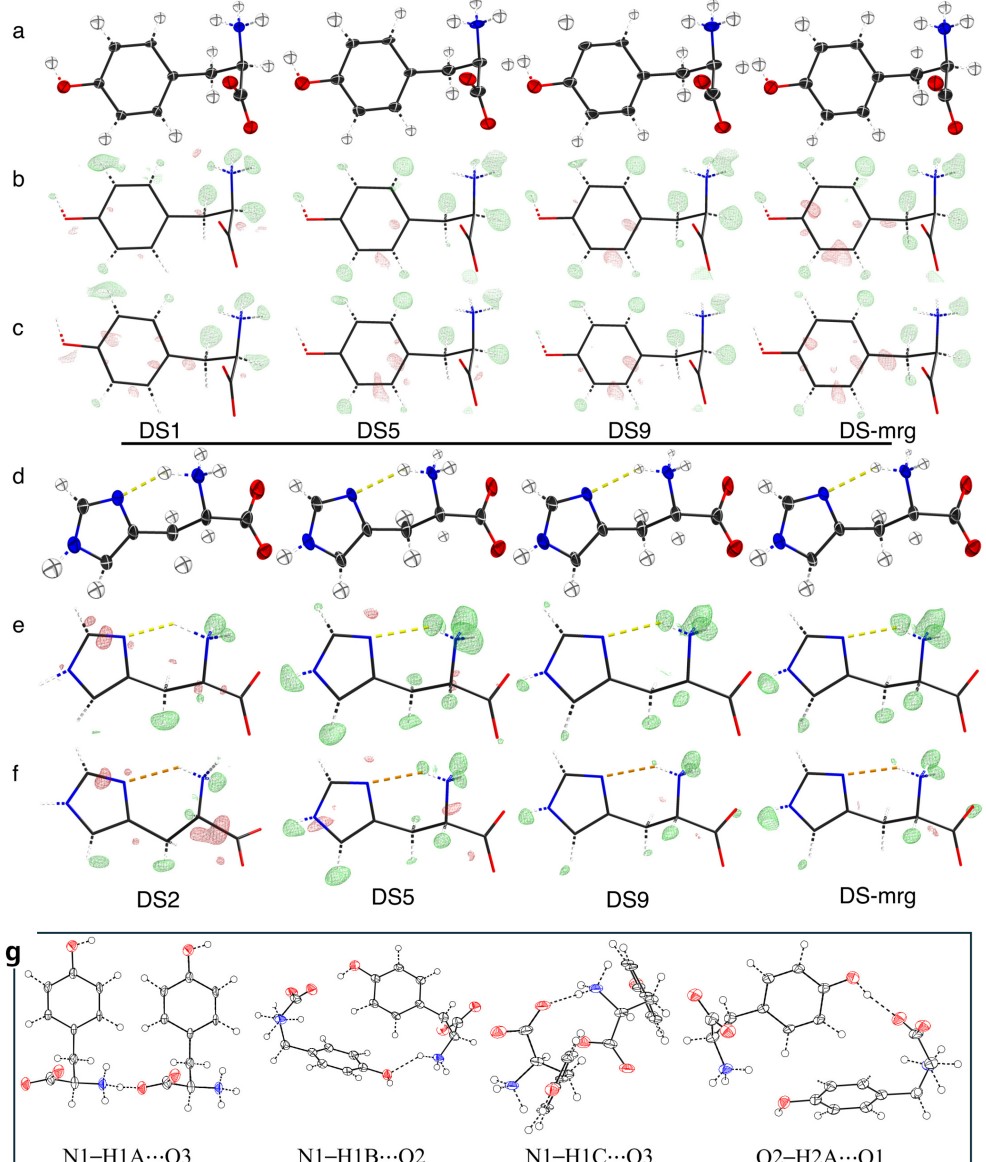

DS1 | DS5 | DS9 | DS-mrg

DS2 | DS5 | DS9 | DS-mrg

**g**

N1–H1A···O3 | N1–H1B···O2 | N1–H1C···O3 | O2–H2A···O1

**Extended Data Fig. 9 | iSFAC modelling enhances hydrogen signal.**
Hydrogen omit maps for tyrosine ((b), with iSFAC modelling; (c), without) and histidine ((e), with iSFAC modelling; (f), without) illustrate the enhanced signal of hydrogen with iSFAC modelling ((b) for tyrosine, (e) for histidine) compared with conventional modelling. OMIT maps are shown as isosurface maps at 3$\sigma$. Note: All protons in (a) and (d) were refined freely and unconstrained. (g) Hydrogen networking in tyrosine crystal packing.

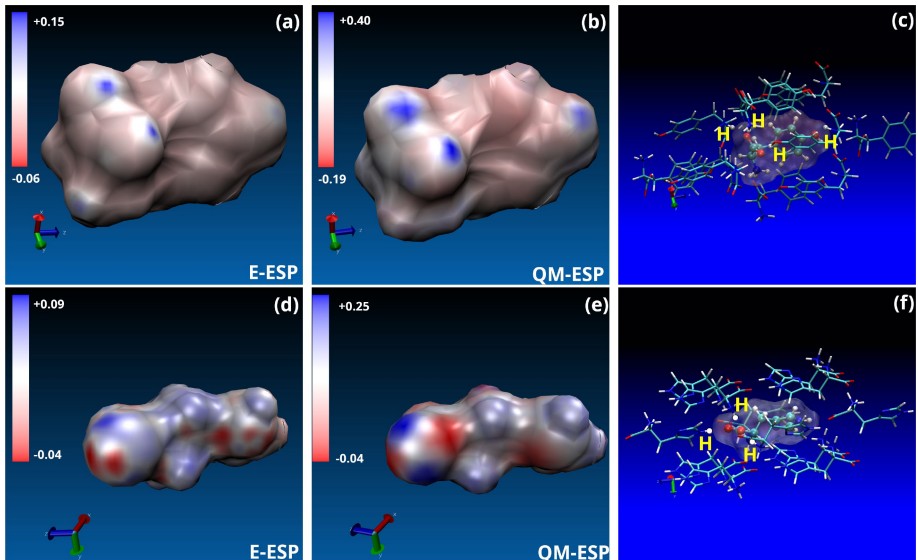

**Extended Data Fig. 10 | ESP maps.** Tyrosine (a-c) and Histidine (d-f): Comparison of the experimental electrostatic potential map (E-ESP, a/d respectively) with the quantum-mechanical electrostatic potential map (QM-ESP, b/e respectively). (c) and (f) show the orientation of the molecules, the next-neighbours composition for the QM maps, and highlight the protons (labelled yellow with "H") responsible for the positive patches of the carboxyl groups and (for tyrosine, c) the hydroxyl group. QM-ESP computed with ORCA Version 6.0.1[71] and MULTIwfn[66,70] with the B3LYP density functional and the 6-311G(d) basis set.