## [Peer Review file · Nature]

Experimental determination of partial charges with electron diffraction

Corresponding Author: Dr Tim Gruene

Version 0:

Reviewer comments:

Referee #1

(Remarks to the Author)

This manuscript describes the refinement of partial electric charges for individual atoms against rotational electron diffraction (ED) patterns. The authors utilize the standard software SHELXL without introducing any new code. Instead, their approach assumes that two atoms share the same x , y , z and ADP values but have different electron scattering factors for the neutral and charged atoms. It then refines the weighting of contributions from these scattering factors. This model, named iSFAC, is intriguing for many ED structures determined at medium resolutions, such as 1.0 to 0.7 Å, which are not sufficient for quantum crystallographic approaches like multipolar and kappa refinements. Thus, this approach will have a wide applicability. However, some inadequacies in technical aspects may impede solid verification. Addressing the following points could improve the manuscript.

1. The shapes of electron scattering factors between neutral atoms and ions differ significantly at low resolution ranges but converge from a certain resolution. This variation can have a substantial impact on refinement results for partial charge values. Therefore, these resolution-dependent effects should be carefully investigated; however, the manuscript does not address this aspect. The scattering curves for partially charged atoms appear similar for different charge values such as 0.1 to 0.3 or 0.5. How sensitive is the proposed approach in determining these charge values?
2. Adding an extra parameter often improves data statistics. It is important to verify that the refined weights between the two scattering factors are meaningful. I am curious about how this approach performs for X-ray diffraction data when using the same electron scattering factors for both neutral and charged atoms.
3. It is unclear why only the scattering curve for H^+ is used in Eq. (4), as this results in incorrect weighting for hydrogen atoms. What would be the outcome if a linear combination of H^0/H^+ were used as for non-hydrogen atoms? How can the use of Eq. (4) be justified? What is the reference for the scattering curve of H^+ ?
4. Why do the electrostatic potential distributions appear significantly different between the maps obtained using this approach and those from quantum mechanical computations, as shown in Figure 7, despite the Pearson coefficient values in Figure 8 being claimed as high? Specifically, there are red regions in the 6-membered ring and other parts in one map but not in the other. There is no description provided on how the maps were produced.
5. It is unclear how the Pearson coefficients were calculated, particularly for the electrostatic maps without iSFAC modeling. It would be helpful if the codes or scripts used for these calculations were provided. I am also interested in understanding why the ADCH calculation yields a better coefficient compared to others. Please include more detailed descriptions of the protocols throughout the manuscript.
6. Pacoste et al. (IUCrJ, 2024) have recently demonstrated that scattering curves for ligand-F(III) obtained using the transferable aspherical atom model (TAAM) yield better refinement statistics compared to those from the IAM model with ionic electron scattering factors. The scattering curves from TAAM also differ from those of the IAM model at low resolution ranges. The author's approach may not be optimal. The refinement heavily depends on the shapes of the scattering curves.

7. The weighting of charged electron scattering factors should be closely related to ADP values. Please provide ADP values both with and without iSFAC refinement.

8. "For macromolecules, qualitative discrimination between oxidation states of metals in enzymes was reported [16]". Yonekura et al., PNAS, 2015 and Ref. [36] are more relevant here.

9. "... it reveals enhanced structural details including the observation of hydrogen atoms (see Fig. S1), to a degree typically associated with dynamical structure refinement [18]". Corresponding difference maps without iSFAC are needed to justify this claim. More detailed figures, similar to Figs. S12–S15 in Ref. [18], would be helpful.

10. "... the fit, the resolution range of the fit was restricted to the resolution range of the particular data sets, 15 Å – 0.75 Å in most cases. Note that the Mott-Bethe-formula enables the parametrisation of seemingly nonphysical charges, like O⁺, C⁺, or C⁻, making it suitable for the fitting of positive and negative partial charges for any atom. We used the Levenberg-Marquardt algorithm as implemented in the program". Approximating ionic scattering curves with 4 gaussians and 1 constant over such a wide range of spatial frequencies can sometimes lead to poorer fits. For the curves of O⁻ and those nonphysical charged ions [16], it would be helpful to present Rscat values as defined in Refs. [16, 36], or to provide other metrics.

(Remarks on code availability)

This manuscript does not include any code except for the data collection GUI software, which is not related to the proposed refinement methodology. It would be helpful if the codes or scripts for calculating the Pearson coefficients and other relevant calculations were provided, along with more detailed descriptions of the protocols throughout the manuscript.

Referee #2

(Remarks to the Author)

Electrons are charged, unlike X-ray photons. Used as probes, electrons are therefore very sensitive to charge in a sample. Technically, the electrons measure the electrostatic potential (ESP, from nuclei and electrons in the probe). It is well established that such ESP maps are much more sensitive to charge and partial charge than electron density (ED) maps. Deducing partial charge from electron diffraction experiments is therefore a reasonable and promising approach. However, it is also known that the effect of partial charges on ESP map appearance is very resolution dependent. In particular, the charge effects appear much more prominent at low resolution than at high resolution, even though a high resolution map contains of course more information. To meaningfully measure experimental partial charges, this resolution or B-factor dependence has to be removed.

The authors solve this problem elegantly, by a method that they term iSFAC (ionic structure factor calculation). Instead of attempting to model the effect of partial charge directly (which is tempting, but a dead end), the authors treat an atom with partial charge as a weighted mixture of a neutral and an ionized forms. These can then separately be treated quantum-chemically. By refining the neutral and ionic form factor contributions of atoms with the SAME displacement (B-factor) and occupancy parameters, the authors get rid of B-factor and resolution effects, at least on paper. Operationally, the whole procedure is quite straightforward: based on the ED from QM-calculations, X-ray form factors can be calculated, which can then be converted to ESP form factors using the Bethe-Mott relation. The application of the Bethe-Mott relation is straightforward because the divergences for non-neutral atoms lie outside of the experimentally accessible resolution range and can therefore be disregarded).

Operationally, the authors measure charges that best explain the ESP maps in terms of an expanded (spherical) form factor set that includes neutral and ionic contributions. The authors then compare their experimental partial charges to QM-calculated partial charges: here, a problem arises from the multiple competing definitions for partial charge that are currently in use (see below).

Major issues:

=====

- On paper, the resolution (B-factor) dependence of partial charge effects on the ESP is elegantly solved by using a joint B-factor for the neutral and ionic forms. However, it would be nice to see to what extent measured partial charges are indeed independent of the resolution and B-factor. B-factors could be varied by collecting ED data at room and cryogenic temperature (which should work for small molecule crystals), and resolution could be varied by using different doses. A comparison of the deduced partial charges would show to what extent the measured partial charges are independent of crystal-specific confounding factors. Also, to make the method broadly applicable, the authors should give the reader information on what minimum quality requirements a dataset has to meet, and –if it does not– how the accuracy of partial charges degrades. For the current implementation, it seems to me that there is even a systematic resolution dependent effect (see below).

- The authors' partial charge assignment protocol can be viewed as an exercise to best model the ESP, as judged by a reciprocal-space, R-factor based weighting scheme. ChEIPG ("Charges from the Electrostatic Potential on a Grid"), RESP, RESP2 and Merz-Singh-Kollman (MK) partial charges also attempt to best model the ESP, but the weighting schemes for how the "true" and partial charge dependent ESPs are compared differ. Therefore, the comparisons between experimental and calculated charges are "apples and oranges" comparisons that confound accuracy issues with issues of how the partial charges are defined. I suggest that the authors separate these two issues. They could use QM calculations to determine a theoretical ESP. They could then determine how well they can model this theoretical ESP with their mixed form factor approach (an interesting matter in itself, not addressed in the manuscript). Most importantly, they could test which of the established partial

charge definitions is most consistent with their own charge definition in this work. I would expect that the answer to be one of the partial charge definitions revolving around approximations to the ESP (i.e. ChEIPG, RESP, RESP2 or MK). But I would not be able to guess which and I would not even be sure that another partial charge definition (e.g. Mulliken, Loewdin) would not come closer. Once the issue of partial charge definition is fully addressed, the authors could then tackle the experimental accuracy of the method.

- If I see it correctly, the practice of matching the partial charge dependent ESP and actual ESP via a reciprocal-space, R-factor dependent criterion has a problematic side effect. As a result, the DEFINITION of the partial charges becomes dependent on CRYSTAL properties, such as resolution and B-factor (insofar as it affects resolution). It seems to me that this is very undesirable and leads to unwanted variability. Shouldn't the resolution range in which the ESP Fourier transforms are scored for agreement by R-factor be fixed, and be part of the partial charge definition?

- For the partial charge definition and measurement protocol proposed in this work to be widely adopted, partial charges for the same molecule for datasets from different crystals need to be consistent. The authors have addressed the consistency issue in Fig. 6, with mixed results. I suspect that part of the problem has to do with different resolutions leading to different definitions of the partial charges (see above), and another part may be due to problems with multiple scattering in electron diffraction (due to the large scattering cross-section). It is known that electron diffraction datasets refine poorly compared to X-ray diffraction datasets, because of departures from kinematical ("first order") scattering. Strong reflections tend to be attenuated by intensity "leaking" to other reflections, and weak reflections tend to be absent, because of intensity spillover. There are crude intensity scaling protocols that repair this defect, at least on average. Are these empirical corrections a good idea for partial charge calculation? And how does more multiple scattering in thicker crystals affect the calculated partial charges?

- Some (partial) charges are >1 or <-1 , for example in metal clusters. Such partial charges cannot be modelled using form factors for neutral and singly ionized form factors alone. Presumably, in some cases, for example in metal clusters, it will not even be known in advance how positive or how negative a partial charge could be. How are such cases handled? As superpositions of different ionization states up to a predictable maximum? Or just as mixtures of two ionization states, if necessary with some trial and error to pick the correct ionization states to work with?

- The guidance of readers to implement the method could be improved. The manuscript comes with a github repo, but the code contains only detector software and is not directly relevant to the method. From the description in Methods, it is clear that the intention is to implement the method by scripting in SHELXL. The relevant section in the manuscript (section 3.4.1) states: "In SHELXL, the fraction v_j is refined through the FVAR mechanism. Equal coordinates and ADPs were imposed with the commands EXYZ and EADP, respectively". While the concept is easy to understand, many readers would struggle with this instruction. I suggest the authors include an example case with hkl reflection file, SHELXL input file, and output file as a test case that readers can run to learn and practice the method.

Minor issues:

=====

- The authors call their method "iSFAC", or "ionic scattering factor calculation", but the abbreviation could easily be mistaken for "ionic structure factor calculation" (as I originally did, which confused me). Most authors use the term "form factor" for the Fourier transform of the ED (X-rays) or ESP (electrons) of an isolated atom or ion. I suggest the authors call their method "ionic form factor calculation", or "iFFAC" to make it clear that ionization is considered at the level of individual atoms.

- How are the error bars on partial charges calculated when only a single dataset is available? Are these from full matrix inversion? Or are these from redundant data collection on the same crystal, and separate calculation of partial charges, to estimate an empirical error?

- Fig. 4: it's nice to see that the measured partial charges are qualitatively reasonable. However, without a reference, it is hard to see whether they are also quantitatively informative. Comparisons of the measured to expected partial charges (ideally for the same partial charge definition) would be helpful.

Summary:

=====

Strong sides:

Until this manuscript, partial charges were a useful, but purely theoretical concept. Although it's implicit only, this work introduces yet another partial charge definition, and presents a protocol to measure the thus defined partial charge experimentally. The approach is elegant (mixing form factors for neutral and charged forms, rather than attempting to quantify the effect of the partial charge alone). It should also be widely applicable, because it can be readily integrated into the SHELXL workflow. Judging from the results on ciprofloxacin, histidine, and tyrosine, it is clear that the method works qualitatively, and also that it improves map quality, especially for polar hydrogen atoms.

Weak sides:

In the current form, the method has a fixable systematic flaw in the dependence of the DEFINITION of the experimental partial charges on the resolution, a property of the CRYSTAL that should not affect the partial charges except for experimental error. It

is also unclear how robust the method is in the presence of crystal limitations, such as limited resolution, high B-factor, or two great crystal thickness to partially invalidate the kinematical scattering approximation. Also, handling of partial charges with absolute value > 1 is unclear. Finally, better reader guidance for how to script SHELXL for the new method is required.

P.S.: This review was (of course!) generated without any use of AI tools, not even language improvement (to avoid disclosing information to a server that is not under my control).

(Remarks on code availability)

The code is a graphical user interface for the SINGLA detector, and therefore relatively peripheral to the core idea of the paper. As I have indicated in the review, the scripting for SHELXL, which is at the core of the method, needs to be documented much better.

Referee #3

(Remarks to the Author)

The primary demonstration of this paper is the ability to derive partial atomic charges directly, readily and quantitatively from a relatively routine diffraction experiment to characterise a crystal structure. This kind of study has in principle been possible via X-ray diffraction experiments for a few decades, however so-called charge density experiments were far from routine, required ultra-high resolution and quality data and were particularly variable due to the fact an X-ray experiment probes electron density and so a 'conversion' to atomic charges was fraught with issues around partitioning the density...for these reasons, charge density experiments were not fully embraced and have essentially fallen out of favour. The introduction of the combination of experimental X-ray data with quantum mechanical approaches has revitalised the field somewhat, however the derivation of atomic charges still hinges on the same problem that its electron density that the experiment is sensitive to.

The authors of this paper show that by using electrons as the probing radiation, which interact with electrostatic potential, atomic charges can effectively be directly determined. This squarely addresses the issue with X-rays and enables the determination of meaningful experimental charges that have been sought for so long. What is more, the approach can be relatively routinely implemented for any kind of data irrespective of resolution. I can see how the developers of all crystallographic software can relatively readily implement this approach and any experimentalist can routinely acquire such insightful information - regardless of their level of expertise. I suspect several research groups developing 3D-ED methods have been thinking along these lines - and in such a rapidly moving new field this is an important 'breakthrough'. So in my view this approach represents a step change in what is achievable and will routinely provide a new level of information from solid-state structure characterisation that provides insights into fundamental chemical concepts such as structure and reactivity.

The paper presents some results from this approach which illustrate the applicability of the method to different types of chemical system, and the type of chemical information that can be gleaned. This section of the paper is important and is fairly comprehensive in terms of what is presented for each of the systems studied, however in themselves this small selection doesn't really show the full impact on the variety of chemistry that can be explored - it is not possible to present a comprehensive range of structures in the body of this paper, but I do feel that the authors could convey more of what is possible in a narrative and raise the excitement level of the paper - making it more readable to a wider audience of chemists. This is achieved in part in section 1.3 which very nicely raises the point of hydrogen atoms when subject to this treatment stop being treated as a constrained sphere attached to an atom simply to complete a model as we know they should be there - they have very varying charges in their role to balance the overall requirement for neutrality. All of a sudden they are not considered purely because of their geometry, which is the simplistic approach to examining them, but one can now really understand their key role in chemical reactivity. This has implications across most of chemistry from catalysis to crystal engineering, chemical biology, etc - again, I think the authors could convey more excitement about the potential. Importantly the authors make comparisons with the equivalent information that can be gleaned from quantum mechanical calculations. They present a statistically meaningful analysis, encompassing a range of QM methods and show how they have arrived at a good comparison.

The conclusions are robust, succinct and illustrate the potential of the approach.

I am a crystallographer and can immediately see the potential impact this approach will have - largely due to the fact that it should be implementable in routine analyses conducted by everyday crystallographers. My only criticism of this paper is that it could be written with a bit more excitement about the broad applicability in order to attract a more general readership. So I encourage the authors to review their text with this in mind and look to embellish it a little in this respect.

(Remarks on code availability)

I have looked at the GitHub repository, but not in detail at all the actual code. The repository is well organised, intuitive and navigable. I do not have the time or experience to perform any testing, but it is clear that it was written to generate the outputs in this paper and anyone with the appropriate environment and expertise wishing to confirm the findings could readily do so.

Version 1:

Reviewer comments:

Referee #1

(Remarks to the Author)

I am afraid that the revised manuscript does not sufficiently address the reviewers' comments and concerns. It appears that the authors have either failed to provide proper answers or have not responded at all. Comparison with QM-calculated charges failed to provide even a qualitative measure of validation (see below). Although the authors provided a PDF file highlighting the changes, many elements, including reference, table and figure numbers and table contents, differ from the unmarked document.

The numbering of references, figures, and tables in this review may be inaccurate due to these discrepancies. Please submit consistent files.

1. The dependence at low resolution is more critical and needs careful examination, given that the electron scattering factors for neutral atoms and ions differ significantly in low-resolution ranges but match in middle to higher resolution ranges. However, the revision focuses solely on the high-resolution cutoff. Please consider low-resolution dependencies.

2. Validating the extracted charge values from electron diffraction data is crucial in this manuscript. I do not think the previous Figure 8 to be a validation for the proposed approach, since most Pearson correlation coefficients without hydrogen charges are around 0, with some showing negative values. It is obvious that including hydrogen charges yields higher correlation values even if just arbitrary positive values are assigned. Therefore, these plots fail to provide even a qualitative measure of validation. Also, the treatment of hydrogen atoms seems problematic (see below). It would be more appropriate to show other forms of validation that do not involve hydrogen atoms. Differences in charge values between the experimental results and QM calculations should be provided for all samples. Why are the estimated charge values from QM calculations missing for the zeolite?

3. The authors claimed that “the ESP map is the first result for quantum chemical calculations, while partial charges are a derived quantity” and removed the comparison of ESP maps in the previous Figure 7. However, experimental ESP data is the primary result, and all values, including x, y, z coordinates, B-factors, and charges, are derived from the experimental ESP data. People would expect ESP to be a straightforward and correct metric for evaluating results directly from electron crystallography. Although the authors also claimed that electron-density-based calculations offer values closer to those from electron diffraction, QM-based values are just the results of how charges are assigned to each atom.

I think ESP maps should serve as a validation and the EESP and QNESP maps should exhibit greater similarity. I would like to know if the contours in the previous maps represent VDW surface rendering. If I understand correctly, the colors in the previous Figure 7 would represent potential values on the VDW surface from point charges assigned at the nucleus. If so, the additional explanation regarding the differences between the crystallographic and single-molecule environments would not be justifiable. The authors should provide more detailed descriptions.

4. Why did the authors use calcium tartrate tetrahydrate for the X-ray diffraction experiments? This compound is chemically quite different from the other examples presented in the manuscript. The samples presented in the manuscript should be tested instead.

5. In the first-round review, I asked which scattering curve was used for H⁺. If it is based on Yonekura et al., IUCrJ, 2018 [37], the difference between H⁰ and H⁺ is not small but quite large. The treatment of hydrogen seems problematic. If the difference is indeed so small, why is the refinement of hydrogen atoms unstable when using H⁰? The weight terms of H⁺ in Eq. (5) approaches 1 when the difference is small. The authors stated, “experimentally, we attribute this observation to the fact that the electron, which would make H⁺ neutral, is already refined as part of the donor atom”. I do not understand why charges in the donor atoms are refined prior to those in the hydrogen atoms. Is it possible to refine hydrogen atoms freely using only the scattering factors of H⁺ and other neutral atoms? The proposed scheme requires more thorough examination and justification.

6. Please explicitly specify the literatures from which the ionic scattering curves for hydrogen and all atoms are adopted. The description about the GNUPLOT scripts refers to refs. [39] and [62], but some essential scattering curves, such as those for H⁺, C⁻, and N⁻, are missing in these references. Ref. [37] includes these curves.

7. The difference densities for hydrogen atoms appear similarly resolved, particularly for tyrosine, with or without iSFAC refinement. Hence, the authors should not claim that the visibility of hydrogen has been enhanced by this approach.

8. This method depends on full-matrix least-squares refinement of hydrogen atoms, which is likely unstable for crystals of more complex molecules such as higher molecular weight. These samples require restraints or constraints for hydrogen atoms. However, this approach is not compatible with the standard command AFIX in SHELXL. Therefore, making strong claims about its versatility would not be appropriate.

9. I do not think the authors properly addressed my concern: “Pacoste et al. (IUCrJ, 2024) have recently demonstrated that scattering curves for ligand-F(III) ...”. This suggests that the theoretical scattering curves may not be optimal, and the refinement is heavily influenced by their shapes. Also given the low Pearson coefficients without hydrogen charges, it would be appropriate to avoid making strong claims about the determination of the absolute partial charge values.

10. The Rscat values in the new table are quite high, with some exceeding 1 - 3%. These poor fits likely introduce significant errors in estimating the contribution from ionic scattering factors. Claiming the determination of absolute partial charge values would also be too strong for this reason. Does the range 0.7 – 19 in the table refer to 0.7 Å to 19 Å?

11. Please give serious consideration to the B-factor variation test proposed by Reviewer #2, conducted at room and cryogenic temperatures. I think this test could be highly effective in distinguishing between the influences of partial charges and B-factors.

12. Multiple scattering would affect charge estimation, as Reviewer #2 concerned. Yet, the author did not address this issue except for discussing data scaling. It should be discussed in more detail, both in the main text and in the response to the reviewer.

(Remarks on code availability)

Now the GitHub addresses are included in the manuscript for codes including those for Pearson correlation coefficients, Cromer-Mann parameters and Rscat values. I am reviewing part of these codes. I would also like to check the codes and scheme used to calculate the EESP and QNESP maps in the previous Figure 7.

Referee #2

(Remarks to the Author)

The authors have meaningfully addressed my concerns regarding the resolution dependent definition of partial charges, and regarding the robustness of the partial charges towards changes of the resolution. Although the partial charge definition issue has not been addressed in a principled way, the authors make a convincing case that for resolutions better than 1 Å, their partial charges are almost resolution-independent, which is sufficient for these charges to be meaningful. In practice, the need for diffraction data to better than 1 Å resolution limits the method to crystals of small molecules, and makes it inapplicable to most if not all electron diffraction data of macromolecules. Importantly, the authors are now clear about when their method is not applicable (or not reliable). Despite the clarified limitations (that rule out some of my "favorite" use cases), the new method is still an important advance that deserves prominent publication.

(Remarks on code availability)

The documentation of the workflow with code is much improved. It is very confusing, however, that the main branch of the github repo (<https://github.com/CF-CSA/edSFAC>) for the code is actually empty (except for a license), whereas the actual code is placed in other branches. I realize that that this is due to github's convention changes regarding "main" and "master" branches (a problem for many repos and not the authors' fault), but still, it would be good if something could be done about it.

Referee #3

(Remarks to the Author)

My opinion of this submission remains, in that the authors present a novel approach to experimentally determining partial charges and that this will have widespread impact, providing new insights for a broad range of chemistry and materials science. I did feel that it was disappointing that the original version didn't include an easy method for any crystallographer or chemist to implement this method. However, I did feel that it is so significant that the developers of the main crystallographic refinement software packages would quickly implement it. I am therefore very pleased to see that scripts are now being made available to generate the appropriate SHELX commands. This will enable some crystallographers to utilise this approach immediately, but will also make it much easier for developers to implement in their software.

The tweaks to wording throughout and streamlining of parts of the paper address many of the concerns I had and I feel the paper reads rather better now. I have carefully read the comments of the other reviewers and feel that the authors have removed distracting parts and included further evidence to back up their approach - most notably regarding resolution dependence.

I believe that this paper is considerably improved and now suitable for publication.

(Remarks on code availability)

I have not been able to review the code. However I do note that further code has been made available - and that this is very important for the wider uptake of this method.

Version 2:

Reviewer comments:

Referee #1

(Remarks to the Author)

I now understand the misunderstanding by the authors. The ionic scattering factors for electrons cannot be derived from the scattering factors of neutral atoms for X-rays, except for H⁺. In Eq. 2, Z does not represent electron charges but nuclear charges, or the atomic number (please carefully refer to Yonekura et al., IUCrJ, 2018, or Peng, Acta Cryst. A, 54, 1998). I plotted the electron scattering factors of O⁻ derived from the International Tables, calculated from the X-ray scattering factor of O⁻ in Yonekura et al., IUCrJ, 2018, and from Eq. 2 of this paper. The first two are identical, but the last one showed a significant discrepancy (see the attached Plot 1). I also plotted the scattering factors for O⁺, O, O⁻ (Plot 2), C⁺, C, C⁻ (Plot 3), and H⁺ (Plot 4) using the Cromer-Mann coefficients (a1, b1, a2, b2, a3, b3, a4, b4 and c) provided in the cif file (2365033.cif) for one tyrosine dataset used in this paper. The plots for O and C matched the curves from the International Tables, but the others showed unusual shapes. The H⁺ curve from Yonekura et al., IUCrJ, 2018 matches the neutral scattering curve in higher resolutions, consistent with other atoms, while the curve used in this paper remains elevated at middle to high resolutions. Therefore, the former curve is likely correct. I then reviewed the Cromer-Mann coefficients in most of the cif files and found that all the values were either identical or very similar. The proposed scheme probably worked because the authors used their own scattering factors. Due to these incorrect ionic scattering curves, it is clear that all the calculations and results presented in this paper are fundamentally flawed.

Additionally, I am concerned that the authors have not provided direct and proper answers to my previous comments, instead offering unrelated and overly general explanations. Please address my comments directly and appropriately. For example, regarding my previous comment 3, my concerns are unrelated to comparisons between crystallography and microscopy. Though the ESP maps have been removed in the current version, I still wish to understand how the ESP maps were generated. Please provide a clear and direct response to at least this point as part of my previous comment. Regarding comment 1, I am not referring to macromolecular data. Again, the correct electron scattering factors of neutral atoms and ions differ only in the

low-resolution ranges but overlap from around 2 Å and higher resolution ranges. Even if the low-resolution parts are limited by lattice parameters, the data still include diffraction spots at lower resolutions and it is important to examine these regions, as charge information cannot be obtained from data in the higher-resolution range. I had hoped the authors would carefully consider and provide correct answers, but unfortunately, their responses have included errors and subsequent sequential corrections. I am now inclined to refrain from providing further feedback on their other responses, as commenting on results derived from incorrect scattering factors seems meaningless.

(Remarks on code availability)

No additional review this time. My previous comment regarding the calculation of ESP maps remains unchanged.

Referee #2

(Remarks to the Author)

As I felt that my most important concerns had been addressed in a previous revision, I am only commenting on the discussion between the authors and referee 1, as requested by the editors.

1) investigation of the low resolution cut-off:

=====

Referee 1 has a point regarding the low resolution differences between the form factors for the neutral and charged forms of atoms.

For MACROMOLECULAR structures, this is a serious concern. However, as already clarified by the authors in their response to me, their method is essentially inapplicable to macromolecular structures anyway, simply because the results become independent of the HIGH resolution limit only in a regime that is hard to reach for macromolecular structures. The problem with the LOW resolution cutoff spotted by referee 1 is yet another reason for why the method would be inapplicable or at least problematic for macromolecules.

For SMALL molecule crystal structures, the low resolution question seems irrelevant at first. As the authors rightly point out in their response, a low resolution cutoff is simply not applied. All reflections except for the 000 reflection (that coincides with the direct beam) can and should be measured. However, there might be a problem that both the authors and I have overlooked that referee 1 seems to have in mind. If the SAME small molecule is crystallized in two different crystal forms that BOTH diffract to very high resolution, would the partial charges determined by the authors' method be the same? The authors take it for granted that the answer would be "yes". Referee 1 seems to expect that the answer would be "no", because the effective low resolution limit is different for different crystal forms (provided cell constants and thus effective lower resolution limits are different). I don't have a clear expectation whether the "yes" or the "no" is correct. Therefore, I suggest that the authors address the question experimentally (or computationally): if indeed experimental partial charges depend on the crystal form even for well-diffracting SMALL molecule crystals, this would be a MAJOR problem for the method. Thus, the referee's concern about the effective low resolution limit should be taken seriously.

2) validating extracted charges:

=====

Referee 1 is correct that the low correlation between calculated and experimental partial charges is problematic. In also agree with the referee that including hydrogen charges yields higher correlation values almost inevitably, as long as any (remotely reasonable) positive values are used. In contrast to Referee 1, I am not surprised by the discrepancies, in fact I expected systematic discrepancies and only qualitative agreement. As I pointed out earlier in a previous refereeing round, the authors effectively introduce a NEW definition of partial charge, which is DIFFERENT from the (already many) other partial charge definitions in use in the quantum chemistry community. For me, it was sufficient that the partial charges are qualitatively reasonable, as long as they are STABLE and not dependent on experimental parameters like the resolution limit. For the HIGH resolution limit, the authors have defined conditions when this is the case. For the LOW resolution side, the situation is unclear and needs to be addressed (see point 1).

3) ESP maps as the primary experimental result:

=====

Referee 1 has a point that theoretical ESP maps calculated from the authors' partial charges should reproduce experimental ESP maps. Moreover, there should be at least reasonable agreement with ESP maps from partial charges that are calculated according to definitions that aim to make partial charges useful to calculate electrostatic potentials. I agree with the authors that ESP maps from electron diffraction are model dependent, if only because phases are model dependent. However, I do not understand how this answer addresses the referee's concern (which I also do not fully understand).

4) Comparison with X-ray diffraction:

=====

I agree with the authors that charge effects in X-ray data are small. Attempts to measure partial charges based on X-ray

diffraction would likely lead to results with large error bars. However, I don't quite see why the authors should be reluctant to do this experiment: it would likely demonstrate the superiority of electron diffraction over X-ray diffraction for the determination of experimental charges, which would strengthen the manuscript. If one WANTS to see charge effects by X-ray diffraction, then the choice of calcium tartrate tetrahydrate makes sense.

5) Hydrogen:
=====

As the authors acknowledge, the referee is right about the big difference in the electron diffraction form factors for H and H+. Like the referee, I do not understand why one would refine partial charges sequentially. I have a hard time to follow the authors justification. In particular, I do not get the statement "Chemically, we attribute this observation to the fact that the electron, which would make H+ neutral, is already refined as part of the donor atom."

6) Literature for ionic scattering curves, definition of form factors for cationic and anionic species:
=====

Equation 4 of Yonekura simply associates the partial charge with the nucleus, and calculates the form factor accordingly. By contrast, I have the impression that the referee expects that the cationic and anionic species are analyzed -like the neutral species- by calculating an electron density quantum chemically (i.e. for the cationic species with one electron less and for the anionic species with one electron more than for the neutral atom). In fact, the referee's, rather than the authors' choice would have been mine. The electron diffraction form factors calculated in the two ways are NOT expected to be equivalent. Either choice can be justified, but the results will not be the same.

7) Enhancement of hydrogen visibility:
=====

The referee is right, but the authors' response seems ok.

8) Least squares refinement of complex molecules:
=====

The referee expects instability of full matrix least squares refinement for larger molecules. Assuming a constant packing density, the number of observations should grow like the number of parameters, provided the resolution for a given resolution. Hence, the observations/parameters ratio should stay constant, and refinement should not get unstable. However, this is only true if the resolution does not degrade with molecule size. In practice, the resolution WILL degrade with molecule size, and this will make the refinement less stable. In other words, the referee has a point that the method will not generalize too well to larger molecules. From the earlier debate, it is already clear that the method is not applicable to macromolecules in practice. From this perspective too, it is reasonable to expect problems with larger "small" molecules.

9) Optimal scattering curves:
=====

The referee is right: the theoretical scattering curves are certainly not optimal (see point 6). In fact, there is a major element of choice, even between the authors' preferred (Yonekura) method, and the referee's preferred approach. Any charges calculated with either method are valid for this choice of form factors. They will be different for a different choice of form factors. I would not have a problem with this, as long as the result is STABLE, i.e. the same for any given molecule, irrespective of crystal form and diffraction limit.

10) Rscat values:
=====

Here, the referee caught an error that the authors and I had missed and that the authors appear to have corrected.

11) Influence of B-factors:
=====

The authors have carried out the experiment that I asked for earlier, and that referee 1 also wanted to see. The results suggest that partial charges deduced at room temperature and at cryogenic temperature are similar, eliminating the concern of a strong B-factor dependence of the partial charges.

12) Multiple scattering:
=====

The effect of errors of partial charges due to multiple scattering was one of my original concerns. It is true that the multiple scattering problem is not properly solved in general, and is a major concern in electron diffraction in general, not only for partial charges. Reviewer 1 is right that this concern was not thoroughly addressed so far. In an earlier round, I suggested to compare partial charges for a small crystal (multiple

scattering is less relevant) and a larger crystal (multiple scattering is a bigger problem). This experiment was not done. I gave it a pass, because I was more concerned about the resolution dependence of the partial charges. However, I concur with referee 1 that the experiment would be very useful to demonstrate robustness of the method, if indeed thicker crystals can be tolerated. The authors' response on this point is handwaving, and does not really address the issue meaningfully. Only an actual experiment with crystals of different size, of the same crystal form, of the same small molecule, could resolve this point satisfactorily.

13) ESP map availability:

=====

"ESP maps are no longer part of the manuscript, as they led to confusion amongst the referees." I am NOT happy with this approach of the authors. The ESP maps are highly relevant and should be included, suppressing them because they create "confusion" is not o.k.

Version 3:

Reviewer comments:

Referee #2

(Remarks to the Author)

Tim Gruene and co-authors have made a major effort that includes new experiments to address the remaining concerns of the unconvinced referee. As my own concerns were already satisfactorily addressed earlier, I will only comment on this debate, referring to discussion points in the order in which they are discussed in the rebuttal letter. I am satisfied with the authors' responses, with the exception of point 3.

1) investigation of the low resolution cut-off:

The authors have characterized different polymorphs of histidine, with similar and different chemical environments for atoms of interest compared to the original crystal form. The results of the partial charge analysis suggest that the partial charge is dependent on the chemical environment, but roughly preserved when only the cell constants change. In my mind, this result essentially eliminates the low resolution concern and resolves the matter satisfactorily.

2) validating extracted charges:

The response to 1) addresses this concern.

3) ESP maps as the primary experimental result:

On this issue, I am not fully satisfied with the authors' answer. The authors argue that differences between calculated ESP maps and ESP map derived from experimental partial charges stem from an inappropriate comparison between calculated ESP maps for the isolated molecule, and experimental ESP maps for the molecule in the crystalline context. The problem could easily be avoided by generating a calculated ESP map for the molecule in its crystalline context. This map should agree well with the map based on experimental partial charges. The insistence of the other referee on this point in the previous round is justified in my eyes.

4) Comparison with X-ray diffraction:

I did not expect unstable refinement when trying to model partial charges based on X-ray data. However, this observation makes the point about the superiority of electron diffraction for partial charge determination even more strongly than a comparison of partial charges obtained from X-ray diffraction and electron diffraction could. Hence, I am satisfied with the authors' answer.

5) Hydrogen:

It now seems that the discussion of this point resulted from a misunderstanding. I would regard the issue as resolved.

6) Literature for ionic scattering curves, definition of form factors for cationic and anionic species:

It is very surprising that only the Yonekura approach for modeling partial charges "works". As we now know, the authors, the other referee, and myself would all have expected otherwise. If I understand correctly, the authors have no explanation for this unexpected finding. If an explanation could be found, it would probably greatly contribute to the understanding of experimentally determined partial charges. However, this should not be a condition of publication, and I accept the authors' response.

7) Enhancement of hydrogen visibility:

This issue was already resolved in the previous round.

8) Least squares refinement of complex molecules:

The authors' response on this point is somewhat vague, but I suggest to accept it.

9) Optimal scattering curves:

There is an element of arbitrariness in the choice of scattering curves. Fig. A7 makes the point that differences between reasonable choices are not minor, especially at low resolution. I stick with my prior point that there are many valid ways to

define partial charges, and that any chemically plausible choice is acceptable, provided that it is stable. In this sense, I find the authors' answer satisfactory.

10) Rscat values:

This issue has been satisfactorily addressed in the previous round.

11) Influence of B-factors:

This issue has been satisfactorily addressed in the previous round.

12) Multiple scattering:

This authors have addressed this point experimentally, by measuring crystals of various thickness. The results do not suggest a systematic dependence of partial charges on crystal thickness. In my eyes, the additional experiments fully resolve the matter.

13) ESP map availability:

The reintroduction of the maps resolves the issue satisfactorily.

(Remarks on code availability)

This issue was resolved in previous refereeing rounds.

Version 4:

Reviewer comments:

Referee #2

(Remarks to the Author)

From the previous reviewing round, there was only one major issue left to be addressed: the discrepancy between calculated and experimental ESP maps that was attributed without proof to the neglect of the crystallographic environment for the theoretical ESP calculations. I had therefore asked the authors to do the calculations for the crystalline system. I expected the authors to use quantum chemistry software that is specifically prepared for the handling of periodic crystalline systems (such as NWChem, SIESTA, or QUANTUM ESPRESSO).

To my great surprise, the authors went ahead with software that is intended for NON-periodic systems. To approximate the crystalline environment, they simply included molecules around the amino acid of interest in their calculations. In general, this is NOT the way to treat periodic systems: it is inexact, and very computationally inefficient compared to calculations for periodic systems. No wonder that the initially planned calculations crashed after a long compute time (10 days). The authors addressed this problem by focusing on a fairly minimal system to approximate the crystalline environment.

Because of the choice of their test systems (crystals of amino acids), this approach turns out to be good enough. The agreement between calculated and experimental ESP maps is excellent. Presumably, the ESP calculations "work" because there are only local neighbor interactions, and no electrons in conduction bands. Had the authors done their tests on a conducting inorganic material, their tests would likely have failed. Perhaps a warning to readers that the authors' way of calculating theoretical ESPs is inefficient and only applicable to systems without strongly delocalized charges would be useful.

The excellent agreement between experimental and calculated ESP maps resolves the last remaining obstacle to publication. I have only one minor request. In the legend to Fig. A13, it would be helpful if the authors could state whether the general comments on quantum chemistry elsewhere in the manuscript

"Quantum-mechanical computations were conducted using Gaussian16 [25] employing the density functionals B3LYP [65] and ω B97XD [66]. The Pople basis sets 6-31G and 6-311G, along with the Karlsruhe basis set def2-tzvp, were utilised."

also apply to this system, or whether a lower level of theory had to be used.

2 Referees' comments:

2.1 Referee #1

(Remarks to the Author):

This manuscript describes the refinement of partial electric charges for individual atoms against rotational electron diffraction (ED) patterns. The authors utilize the standard software SHELXL without introducing any new code. Instead, their approach assumes that two atoms share the same x, y, z and ADP values but have different electron scattering factors for the neutral and charged atoms. It then refines the weighting of contributions from these scattering factors. This model, named *iSFAC*, is intriguing for many ED structures determined at medium resolutions, such as 1.0 to 0.7 Å, which are not sufficient for quantum crystallographic approaches like multipolar and kappa refinements. Thus, this approach will have a wide applicability. However, some inadequacies in technical aspects may impede solid verification. Addressing the following points could improve the manuscript.

1. The shapes of electron scattering factors between neutral atoms and ions differ significantly at low resolution ranges but converge from a certain resolution. This variation can have a substantial impact on refinement results for partial charge values. Therefore, these resolution-dependent effects should be carefully investigated; however, the manuscript does not address this aspect. The scattering curves for partially charged atoms appear similar for different charge values such as 0.1 to 0.3 or 0.5. How sensitive is the proposed approach in determining these charge values?

Response: This remark is similar to one of the remarks from reviewer 2. We included a section '1.5 Robustness and resolution dependence' in the main text, supported by Fig. A1 in the Extended Data section. The Figures plot the partial charges when cutting the resolution between 0.7 Å and 1.2 Å. They show a systematic trend, that is expected when reducing the resolution, and they also show that significant deviations only occur near 1.0-1.2 Å resolution – which is also the limit for atomic resolution crystal structure. Hence, these figures underline that *iSFAC* modelling is generally available and possible with data quality that coincides with current standards in crystallography. This remark was therefore of utter importance for improving our manuscript, which we are grateful for.

2. Adding an extra parameter often improves data statistics. It is important to verify that the refined weights between the two scattering factors are meaningful. I am curious about how this approach performs for X-ray diffraction data when using the same electron scattering factors for both neutral and charged atoms.

Response: The reviewer is right in that additional parameters are bound to reduce metrics like the crystallographic R-factor. For this reason, we sought for validation orthogonal to crystallography: comparison with quantum chemical computations, and the interpretation of our results in a chemical context: the clear separation between O and Si for zeolites, the differentiation between carboxylic acid and carboxylate. In addition, we carried out the suggested experiment with a data set of calcium tartrate, collected inhouse with one of our X-ray diffractometers. From the beginning, it was difficult to replace X-ray scattering

factors with electron scattering factors. The R1-factor rose from 3.84 % to 21.9 %, and all ADPs became non-positive definite - an expected behaviour, considering the different shapes of the scattering curves. With iSFAC modelling, the refinement became unstable after a couple of cycles. When stopped after 10 cycles, the free variables for the partial charges were already far over the roof, with chemical non-sense values:

Ca ²⁺	12e	O2	+65e	O3	+89e	O4	+63e	O5	+85e	
	O6	77e	O8	+86e	C1	+53e	C2	+63e	C3	+59e
	C4	+64e	O	+99e						

None of the hydrogen atoms could be refined freely, and the crystallographic R1-factor raised above 40 %.

We are not sure how this could be included in the manuscript. If the reviewer finds it helpful, we could add the “exploding” RES file to the github repository https://github.com/CF-CSA/iSFAC_code.git?

3. It is unclear why only the scattering curve for H^+ is used in Eq. (4), as this results in incorrect weighting for hydrogen atoms. What would be the outcome if a linear combination of H^0/H^+ were used for non-hydrogen atoms? How can the use of Eq. (4) be justified? What is the reference for the scattering curve of H^+ ?

Response: This was an oversight in Section 3.4.2 — we forgot to explain the reason for the special treatment of H atoms. When hydrogen are refined just like other atoms, refinement does not stabilise. Crystallographically, we assume that, although iSFAC modelling enhances the role of hydrogen atoms in crystallographic models, the difference between H^0 and H^+ is still small compared with the other elements. However, there is also a chemical reason, since the electron of the H^0 is shared with the donor atom, and hence modelled by the partial charge contribution of the donor. This is only one observation that stunned us during this project- it is amazing, how much chemical information is revealed with iSFAC modelling.

quote from section 3.4.2:

When hydrogen atoms are refined as linear superposition, similar to all other atoms, the refinement does not converge and stops as unstable. Theoretically, the difference between the electron scattering factors for H^0 and H^+ is most likely too small. Experimentally, we attribute this observation to the fact that the electron, which would make H^+ neutral, is already refined as part of the donor atom.

4. Why do the electrostatic potential distributions appear significantly different between the maps obtained using this approach and those from quantum mechanical computations, as shown in Figure 7, despite the Pearson coefficient values in Figure 8 being claimed as high? Specifically, there are red regions in the 6-membered ring and other parts in one map but not in the other. There is no description provided on how the maps were produced.

Response: It appears that showing the electrostatic potential maps (Fig. 7) was rather confusing, as judged by this comment, and reviewer 2. We finally decided to omit this presentation. Here, we would like to explain why, and the reasons for discrepancies:

The ESP map is the first result for quantum chemical calculations, while partial charges are a derived quantity. As we show, this derived quantity from the QM calculated maps can have significant variations, depending on the chosen approach. For the experimental map, the situation is inverted, so to speak: the partial charge information is the experimental result derived directly from the observed data, while the map is a side product from refinement, and it combines observed data with phases computed from the model. Hence, the experimental ESP is only a derived quantity from the partial charges. This is also reflected in X-ray charge density studies, where the ESP cannot be computed all from the IAM approximation. Still, a multipole model or a database approach like TAAM or INVARIOM is required.

In addition to the above explanation, the experimental map shows the molecule within the crystallographic environment, including the interaction with neighbouring molecules, while the quantum chemically computed model is computed from a single molecule and from a “simulated crystal” with only a few unit cells.

However, as pointed out, the maps were only an add-on to the manuscript and apparently lead to distraction from the main message, the assignment of partial charges to the individual atoms.

5. It is unclear how the Pearson coefficients were calculated, particularly for the electrostatic maps without iSFAC modelling. It would be helpful if the codes or scripts used for these calculations were provided. I am also interested in understanding why the ADCH calculation yields a better coefficient compared to others. Please include more detailed descriptions of the protocols throughout the manuscript.

Response: We created a github repository https://github.com/CF-CSA/iSFAC_code.git for various codes used in the manuscript, including the scripts for computing Pearson coefficients. We would like to point out that the maps are only shown for completeness, and the correlation coefficients are computed for the partial charge values per atom. The maps are only derived information from the crystallographic refinement, while the partial charge values are direct information. It seems, also from one of the comments of Reviewer 2, that the ESP maps in Fig. 8 can easily be misunderstood. As explained above, we decided to omit this figure from the manuscript.

6. Pacoste et al. (IUCrJ, 2024) have recently demonstrated that scattering curves for ligand-F(III) obtained using the transferable aspherical atom model (TAAM) yield better refinement statistics compared to those from the IAM model with ionic electron scattering factors. The scattering curves from TAAM also differ from those of the IAM model at low-resolution ranges. The author’s approach may not be optimal. The refinement heavily depends on the shapes of the scattering curves.

Response: We agree with the reviewer in that the community in ED is discussing which are suitable scattering factors (the roots for iSFAC modelling were laid by a comment on refining scattering factors by Richard Henderson to author TG). For this reason, our manuscript goes in depths into validation of the results, both chemically (carboxylate in the amino acids vs. carboxylic acid in ciprofloxacin) and theoretically by the quantum

chemical computations. There are database approaches like TAAM or INVARIOM, that attempt to circumvent limitations that arise, when normal resolution data are available, as opposed to ultra-high resolution data required for multipole refinement. As far as we understand, such approaches are limited in that fragments need to be precalculated, which requires special expertise (this was the case for the INVARIOM data base, when author TG was discussing with Birger Dittrich), and they are limited by compounds that escape such computations. E.g., the building block of zeolites is as simple as SO_4 , yet 12 independent T-site in ZSM-5, plus the enormous variety of zeolites, implies that the T-sites all have different environments, that prohibits computations. At least this was the result of a discussion with the theoretician in the van Bokhoven group a couple of years ago.

7. The weighting of charged electron scattering factors should be closely related to ADP values. Please provide ADP values both with and without iSFAC refinement.

Response: The program SHELXL prints the largest correlation coefficients in its log file. The strongest correlations are below 60 % between the weighting of the partial charge for hydrogen atoms and their ADP values. For non-hydrogen atoms, the correlation is at or below 30 %. Tables for comparison of ADP values between iSFAC modelling and conventional refinement have been added for each of the four compounds as tables S3, S4, S7, and S9.

8. “For macromolecules, qualitative discrimination between oxidation states of metals in enzymes was reported [16]”. Yonekura et al., PNAS, 2015 and Ref. [36] are more relevant here.

Response: We updated the references.

For macromolecules, qualitative discrimination between oxidation states of metals in enzymes was reported [16, 17].

9. “... it reveals enhanced structural details including the observation of hydrogen atoms (see Fig. S1), to a degree typically associated with dynamical structure refinement [18]”. Corresponding difference maps without iSFAC are needed to justify this claim. More detailed figures, similar to Figs. S12–S15 in Ref. [18], would be helpful.

Response: This is a very helpful suggestion. We extended Fig. S1 (now Fig. A4 in the Extended Data Section) with hydrogen omit maps generated from refinement without iSFAC modelling.

10. “... the fit, the resolution range of the fit was restricted to the resolution range of the particular data sets, $15 \text{ \AA} - 0.75 \text{ \AA}$ in most cases. Note that the Mott-Bethe-formula enables the parametrisation of seemingly nonphysical charges, like O^+ , C^+ , or C^- , making it suitable for the fitting of positive and negative partial charges for any atom. We used the Levenberg-Marquardt algorithm as implemented in the program”. Approximating ionic scattering curves with 4 gaussians and 1 constant over such a wide range of spatial frequencies can sometimes lead to poorer fits. For the curves of O^- and those nonphysical charged ions [16], it would be helpful to present R_{scat} values as defined in Refs. [16, 36], or to provide other metrics.

Response: We added a list of Rscat values for the ionic scattering factors in the supplement, Table S2. The python script for how Rscat was calculated is included in the github repository https://github.com/CF-CSA/iSFAC_code.git, with the link provided in the data availability section.

Remarks on code availability:

This manuscript does not include any code except for the data collection GUI software, which is not related to the proposed refinement methodology. It would be helpful if the codes or scripts for calculating the Pearson coefficients and other relevant calculations were provided, along with more detailed descriptions of the protocols throughout the manuscript.

Response: We created a github repository for this manuscript that includes the various codes we used, and a separate github repository for the computation for the SHELXL SFAC commands. Both repositories are listed in the 'data availability section'.

Referee #2 :

2.1.1 Referee #2 (Remarks to the Author):

Electrons are charged, unlike X-ray photons. Used as probes, electrons are therefore very sensitive to charge in a sample. Technically, the electrons measure the electrostatic potential (ESP, from nuclei and electrons in the probe). It is well established that such ESP maps are much more sensitive to charge and partial charge than electron density (ED) maps. Deducing partial charge from electron diffraction experiments is therefore a reasonable and promising approach. However, it is also known that the effect of partial charges on ESP map appearance is very resolution dependent. In particular, the charge effects appear much more prominent at low resolution than at high resolution, even though a high resolution map contains of course more information. To meaningfully measure experimental partial charges, this resolution or B-factor dependence has to be removed.

The authors solve this problem elegantly, by a method that they term iSFAC (ionic structure factor calculation). Instead of attempting to model the effect of partial charge directly (which is tempting, but a dead end), the authors treat an atom with partial charge as a weighted mixture of a neutral and an ionized forms. These can then separately be treated quantum-chemically. By refining the neutral and ionic form factor contributions of atoms with the SAME displacement (B-factor) and occupancy parameters, the authors get rid of B-factor and resolution effects, at least on paper. Operationally, the whole procedure is quite straightforward: based on the ED from QM-calculations, X-ray form factors can be calculated, which can then be converted to ESP form factors using the Bethe-Mott relation. The application of the Bethe-Mott relation is straightforward because the divergences for non-neutral atoms lie outside of the experimentally accessible resolution range and can therefore be disregarded.

Operationally, the authors measure charges that best explain the ESP maps in terms of an expanded (spherical) form factor set that includes neutral and ionic contributions. The authors then compare their experimental partial charges to QM-calculated partial charges: here, a problem arises from the multiple competing definitions for partial charge that are currently in use (see below).

Major issues:

- On paper, the resolution (B-factor) dependence of partial charge effects on the ESP is elegantly solved by using a joint B-factor for the neutral and ionic forms. However, it would be nice to see to what extent measured partial charges are indeed independent of the resolution and B-factor. B-factors could be varied by collecting ED data at room and cryogenic temperature (which should work for small molecule crystals), and resolution could be varied by using different doses. A comparison of the deduced partial charges would show to what extent the measured partial charges are independent of crystal-specific confounding factors. Also, to make the method broadly applicable, the authors should give the reader information on what minimum quality requirements a dataset has to meet, and –if it does not– how the accuracy of partial charges degrades. For the current implementation, it seems to me that there is even a systematic resolution dependent effect (see below).

Response: This remark is similar to the first remark of reviewer 1, and we extended the manuscript with a section 'Robustness and resolution dependence'. We attempted to collect data that would mimic various crystal qualities with varying beam intensity - when we tried with histidine, the crystal was too damaged after the first, low intensity data collection, which was meant to simulate a low resolution data set. Encouraged by the suggestion of reviewer 1, we refined the data sets with varying resolution cut offs and prepared respective graphs. The results are very good, and up to 0.9 Å resolution, the variation is less than one would expect from a crystallographic refinement.

- The authors' partial charge assignment protocol can be viewed as an exercise to best model the ESP, as judged by a reciprocal-space, R-factor based weighting scheme. ChEIPG ("Charges from the Electrostatic Potential on a Grid"), RESP, RESP2 and Merz-Singh-Kollman (MK) partial charges also attempt to best model the ESP, but the weighting schemes for how the "true" and partial charge dependent ESPs are compared differ. Therefore, the comparisons between experimental and calculated charges are "apples and oranges" comparisons that confound accuracy issues with issues of how the partial charges are defined. I suggest that the authors separate these two issues. They could use QM calculations to determine a theoretical ESP. They could then determine how well they can model this theoretical ESP with their mixed form factor approach (an interesting matter in itself, not addressed in the manuscript). Most importantly, they could test which of the established partial charge definitions is most consistent with their own charge definition in this work. I would expect that the answer to be one of the partial charge definitions revolving around approximations to the ESP (i.e. ChEIPG, RESP, RESP2 or MK). But I would not be able to guess which and I would not even be sure that another partial charge definition (e.g. Mulliken, Loewdin) would not come closer. Once the issue of partial charge definition is fully addressed, the authors could then tackle the experimental accuracy of the method.

Response: The primary focus of this manuscript is the determination of partial charges from experimental data. Therefore, our analysis centers on comparing these data with quantum-mechanical results at this level directly. While we recognize that various methods, with differing levels of success, have been developed to derive partial charges from the electrostatic potential, the aim of this work is neither to evaluate these quantum-mechanical

methods nor to improve them. Instead, our goal is to present a straightforward methodology for deriving meaningful partial charges directly from experimental data. Our findings indicate that experimental results align more closely with methods based on electron density rather than those relying on the electrostatic potential. Consequently, the latter property does not seem the best level for comparison. From the experimental perspective, partial charges represent a fundamental output, while the electrostatic map is a derived property. This contrasts with the approach in quantum-mechanical methodologies, where the order is reversed.

- If I see it correctly, the practice of matching the partial charge dependent ESP and actual ESP via a reciprocal-space, R-factor dependent criterion has a problematic side effect. As a result, the DEFINITION of the partial charges becomes dependent on CRYSTAL properties, such as resolution and B-factor (insofar as it affects resolution). It seems to me that this is very undesirable and leads to unwanted variability. Shouldn't the resolution range in which the ESP Fourier transforms are scored for agreement by R-factor be fixed, and be part of the partial charge definition?

Response: It appears that the presentation of ESP maps in Fig. 7 distracts from the actual workflow of our method. As explained in greater detail in responses to Reviewer 1, we decided to remove this figure. The experimental ESP is only a derived quantity from crystallographic refinement, as side product, so to speak. The partial charges are a result of the refinement, without intermittent steps. As responded above, the new section on robustness and resolution (in-)dependence illustrates that iSFAC partial charges are independent of data resolution, at least within the range of what is commonly considered a good crystal structure, i.e. better than 1 Å resolution. With worse resolution, reliable partial charges can be produced when refinement is stabilised with restraints (as one anyway needs to do at such resolution range).

- For the partial charge definition and measurement protocol proposed in this work to be widely adopted, partial charges for the same molecule for datasets from different crystals need to be consistent. The authors have addressed the consistency issue in Fig. 6, with mixed results. I suspect that part of the problem has to do with different resolutions leading to different definitions of the partial charges (see above), and another part may be due to problems with multiple scattering in electron diffraction (due to the large scattering cross-section). It is known that electron diffraction datasets refine poorly compared to X-ray diffraction datasets, because of departures from kinematical ("first order") scattering. Strong reflections tend to be attenuated by intensity "leaking" to other reflections, and weak reflections tend to be absent, because of intensity spillover. There are crude intensity scaling protocols that repair this defect, at least on average. Are these empirical corrections a good idea for partial charge calculation? And how does more multiple scattering in thicker crystals affect the calculated partial charges?

Response: The question of resolution dependence was also raised by reviewer 1, and we added data to the manuscript that illustrate how little iSFAC modelling depends on resolution. Even less, than we expected. Only at worse than 1–1.2~, significant changes occur. However, this is a resolution range where the data to parameter ratio becomes rather small

and even conventional modelling, even with X-ray data, cannot be carried out without substantial support by restraints or constraints. We are grateful to both reviewers, because this additional information will be a great assurance for the general reader of the manuscript. Regarding the question about scaling: We have always scaled our ED data, as implemented in the CORRECT step of the program XDS. We do this routinely, whether or not it is for iSFAC modelling. This scaling step can be turned off (with the option SNRC=50), but in all cases we tried this made data quality significantly worse to the extent that structure solution would failed. We also tried scaling with the program SADABS instead of XSCALE, but again, the effect was usually negative. We do not know the exact reason. However, this discussion is independent from iSFAC modelling, and we made this observation several years ago. Therefore, we would rather not include this very technical discussion in the manuscript, in particular it is based on our experience without rigorous scientific investigations.

- Some (partial) charges are >1 or <-1 , for example in metal clusters. Such partial charges cannot be modelled using form factors for neutral and singly ionized form factors alone. Presumably, in some cases, for example in metal clusters, it will not even be known in advance how positive or how negative a partial charge could be. How are such cases handled? As superpositions of different ionization states up to a predictable maximum? Or just as mixtures of two ionization states, if necessary with some trial and error to pick the correct ionization states to work with?

Response: We amended section 3.4.1 with the explanation of these situations, and also show the Mott-Bethe formula explicitly in section 3.4.

- The guidance of readers to implement the method could be improved. The manuscript comes with a github repo, but the code contains only detector software and is not directly relevant to the method. From the description in Methods, it is clear that the intention is to implement the method by scripting in SHELXL. The relevant section in the manuscript (section 3.4.1) states: “In SHELXL, the fraction ν_j is refined through the FVAR mechanism. Equal coordinates and ADPs were imposed with the commands EXYZ and EADP, respectively”. While the concept is easy to understand, many readers would struggle with this instruction. I suggest the authors include an example case with hkl reflection file, SHELXL input file, and output file as a test case that readers can run to learn and practice the method.

Response: We created a video on how to prepare the SHELXL input file, which we are happy to provide along with the manuscript. We also extended the github repositories to provide the various source codes that we used in this work. We presume that the developers of Olex2 will pick up our manuscript, once published, and make the file preparations much more automated, while we prepared the input files manually.

Minor issues:

- The authors call their method “iSFAC”, or “ionic scattering factor calculation”, but the abbreviation could easily be mistaken for “ionic structure factor calculation” (as I originally

did, which confused me). Most authors use the term “form factor” for the Fourier transform of the ED (X-rays) or ESP (electrons) of an isolated atom or ion. I suggest the authors call their method “ionic form factor calculation”, or “iFFAC” to make it clear that ionization is considered at the level of individual atoms.

Response: The term “iSFAC” is a mnemonic to the SHELXL command “SFAC”, the short hand for the Cromer-Mann parametrisation of the atomic scattering factor $f(s)$. We would like to keep this mnemonic. The term “atom form factor” is commonly used. Some literature also name the atom form factor as atom scattering factor, e.g. the textbooks by Massa or Giacovazzo. Therefore, iSFAC, while being an artificial term, is not too far from the terminology used in crystallography.

- How are the error bars on partial charges calculated when only a single dataset is available? Are these from full matrix inversion? Or are these from redundant data collection on the same crystal, and separate calculation of partial charges, to estimate an empirical error?

Response: The reviewer is right, we used estimated uncertainties resulting from full matrix inversion. We added this explanation to section 3.3 (Data processing and model refinement)

- Fig. 4: it’s nice to see that the measured partial charges are qualitatively reasonable. However, without a reference, it is hard to see whether they are also quantitatively informative. Comparisons of the measured to expected partial charges (ideally for the same partial charge definition) would be helpful.

Response: We are not aware of a method that determines partial charges that we could compare our results with. This is why we are so excited about the results with iSFAC modelling. iSFAC modelling itself is quantitative on an absolute scale, because of the use of ionic scattering factors: The scattering factor for S^{IV} represent a charge of $+4e$, that of O^- is $-1e$. Hence, the fractions derived via iSFAC modelling are fractions of these absolute values. We went into great strengths to validate our results.

Summary:

Strong sides:

Until this manuscript, partial charges were a useful, but purely theoretical concept. Although it’s implicit only, this work introduces yet another partial charge definition, and presents a protocol to measure the thus defined partial charge experimentally. The approach is elegant (mixing form factors for neutral and charged forms, rather than attempting to quantify the effect of the partial charge alone). It should also be widely applicable, because it can be readily integrated into the SHELXL workflow. Judging from the results on ciprofloxacin, histidine, and tyrosine, it is clear that the method works qualitatively, and also that it improves map quality, especially for polar hydrogen atoms.

Remark: We are grateful for the very positive judgement of our manuscript. It was difficult to explain the results in a transparent, general manner, and it seems that we succeeded - even better now since we took all reviewers’ comments into account.

Weak sides:

In the current form, the method has a fixable systematic flaw in the dependence of the DEFINITION of the experimental partial charges on the resolution, a property of the CRYSTAL that should not affect the partial charges except for experimental error. It is also unclear how robust the method is in the presence of crystal limitations, such as limited resolution, high B-factor, or two great crystal thickness to partially invalidate the kinematical scattering approximation. Also, handling of partial charges with absolute value > 1 is unclear. Finally, better reader guidance for how to script SHELXL for the new method is required.

Remark: We believe that the revised version of our manuscript addresses all of the points raised. The plot comparing resolution cut-offs illustrate that variations are indeed due to experimental error, and we added a section on this topic of robustness, as well as clarified the question on partial charges $|q| > 1e$. These were all extremely useful comments.

P.S.:

This review was (of course!) generated without any use of AI tools, not even language improvement (to avoid disclosing information to a server that is not under my control).

Referee #2 (Remarks on code availability):

The code is a graphical user interface for the SINGLA detector, and therefore relatively peripheral to the core idea of the paper. As I have indicated in the review, the scripting for SHELXL, which is at the core of the method, needs to be documented much better.

Response: We provide more repositories on github, including the scripts for generating SHELXL SFAC commands, and we also created two videos, one for how to use the script (unfortunately, the volume is very low, even though we tried twice), and one for the refinement process with SHELXL / SHELXLE.

Referee #3 :

Response: We are grateful by the overwhelmingly positive comments by Reviewer #3. We did revise the Abstract and Conclusions of the paper to raise more excitement from the reader.

(Remarks to the Author):

The primary demonstration of this paper is the ability to derive partial atomic charges directly, readily and quantitatively from a relatively routine diffraction experiment to characterise a crystal structure. This kind of study has in principle been possible via X-ray diffraction experiments for a few decades, however so-called charge density experiments were far from routine, required ultra-high resolution and quality data, and were particularly variable due to the fact that an X-ray experiment probes electron density. Therefore, a 'conversion' to atomic charges was fraught with issues around partitioning the density. For these reasons, charge density experiments were not fully embraced and have essentially fallen out of favour.

The introduction of the combination of experimental X-ray data with quantum mechanical approaches has revitalised the field somewhat, however, the derivation of atomic charges still hinges on the same problem that its electron density that the experiment is sensitive to. The authors of this paper show that by using electrons as the probing radiation, which interact with electrostatic potential, atomic charges can effectively be directly determined. This squarely addresses the issue with X-rays and enables the determination of meaningful experimental charges that have been sought for so long. What is more, the approach can be relatively routinely implemented for any kind of data irrespective of resolution. I can see how the developers of all crystallographic software can relatively readily implement this approach and any experimentalist can routinely acquire such insightful information — regardless of their level of expertise. I suspect several research groups developing 3D-ED methods have been thinking along these lines — and in such a rapidly moving new field, this is an important 'breakthrough'. So in my view, this approach represents a step change in what is achievable and will routinely provide a new level of information from solid-state structure characterisation that provides insights into fundamental chemical concepts such as structure and reactivity.

The paper presents some results from this approach which illustrate the applicability of the method to different types of chemical systems, and the type of chemical information that can be gleaned. This section of the paper is important and is fairly comprehensive in terms of what is presented for each of the systems studied; however, in themselves, this small selection doesn't really show the full impact on the variety of chemistry that can be explored. It is not possible to present a comprehensive range of structures in the body of this paper, but I do feel that the authors could convey more of what is possible in a narrative and raise the excitement level of the paper — making it more readable to a wider audience of chemists.

This is achieved in part in section 1.3 which very nicely raises the point that hydrogen atoms, when subject to this treatment, stop being treated as a constrained sphere attached to an atom simply to complete a model, as we know they should be there. Instead, they have very varying charges in their role to balance the overall requirement for neutrality. All of a sudden, they are not considered purely because of their geometry, which is the simplistic approach to examining them, but one can now really understand their key role in chemical reactivity. This has implications across most of chemistry, from catalysis to crystal engineering, chemical biology, etc. Again, I think the authors could convey more excitement about the potential.

Importantly, the authors make comparisons with the equivalent information that can be gleaned from quantum mechanical calculations. They present a statistically meaningful analysis, encompassing a range of QM methods, and show how they have arrived at a good comparison. The conclusions are robust, succinct, and illustrate the potential of the approach.

I am a crystallographer and can immediately see the potential impact this approach will have — largely due to the fact that it should be implementable in routine analyses conducted by everyday crystallographers. My only criticism of this paper is that it could be written with a bit more excitement about the broad applicability in order to attract a more general readership. So I encourage the authors to review their text with this in mind and look to embellish it a little in this respect.

Referee #3 (Remarks on code availability):

I have looked at the GitHub repository, but not in detail at all the actual code. The repository is well organised, intuitive, and navigable. I do not have the time or experience to perform any testing, but it is clear that it was written to generate the outputs in this paper and anyone with the appropriate environment and expertise wishing to confirm the findings could readily do so.

Author response letter

Tim Gruene with all co-authors

9th December 2024

1 Referees' comments

1.1 Referee #1 (Remarks to the Author)

I am afraid that the revised manuscript does not sufficiently address the reviewers' comments and concerns. It appears that the authors have either failed to provide proper answers or have not responded at all. Comparison with QM-calculated charges failed to provide even a qualitative measure of validation (see below). Although the authors provided a PDF file highlighting the changes, many elements, including reference, table and figure numbers and table contents, differ from the unmarked document. The numbering of references, figures, and tables in this review may be inaccurate due to these discrepancies. Please submit consistent files.

Response: We are sorry about the discrepancy between the marked-up document and the article file. Nature asked us to reformat the manuscript according to the Nature template. This included heavy restructuring, resulting to the discrepancies between the revised version and the PDF that highlighted the differences to the original submission. E.g. our Supplement was moved up into the Methods and Materials section, according to the Nature guidelines. Since this revision is based on the properly formatted revision 1, the mark-up PDF is now better consistent.

1. The dependence at low resolution is more critical and needs careful examination, given that the electron scattering factors for neutral atoms and ions differ significantly in low-resolution ranges but match in middle to higher resolution ranges. However, the revision focuses solely on the high-resolution cutoff. Please consider low-resolution dependencies.

Response: We are not sure what the referee has in mind considering low resolution. Our data sets have different unit cells, thus the lowest possible resolution also differs. Referee 2 indicates an interest in macromolecular structures. Maybe referee 1 also has macromolecular structures in mind? Some technical details of macromolecular crystallography are quite different from small-molecule crystallography, e.g. small-molecule crystallography can afford least-squares minimisation, and structures often need no, or

few restraints. While the high-resolution cut-off is subject to the judgement of the crystallographer, based in quality criteria like I/σ_I or $CC_{1/2}$, the low-resolution limit is set fixed by the unit cell dimensions. We usually aim to measure all low-resolution reflections. If they cannot be captured by a single run, we aim for a low-resolution run at increased detector distance. To reflect this, we amended Section 2.5, including a reference of Zbigniew Dauter, an expert and excellent teacher concerning the collection of high quality data. The amendment reads on page 10, second paragraph before Section 2.6:

Since the charge information of scattering factors of ions is greatest at low resolution (at $s \rightarrow 0 \text{ \AA}^{-1}$, see e.g. Fig. 2), one needs to ensure high data completeness at low resolution. In case the detector cannot capture the entire resolution range, the high-resolution pass should be followed by a low-resolution pass, as is common practice for good data collection [26].

We sincerely hope this captures the referee's remark.

2. Validating the extracted charge values from electron diffraction data is crucial in this manuscript. I do not think the previous Figure 8 to be a validation for the proposed approach, since most Pearson correlation coefficients without hydrogen charges are around 0, with some showing negative values. It is obvious that including hydrogen charges yields higher correlation values even if just arbitrary positive values are assigned. Therefore, these plots fail to provide even a qualitative measure of validation. Also, the treatment of hydrogen atoms seems problematic (see below). It would be more appropriate to show other forms of validation that do not involve hydrogen atoms. Differences in charge values between the experimental results and QM calculations should be provided for all samples. Why are the estimated charge values from QM calculations missing for the zeolite?

Response: During the course of the work, we spent much thought into how to validate the results, which hasn't been easy, introducing an experimental method without real precedence. There is chemical validation with the discussion on carboxylate vs. carboxylic acid (Fig. 1 on p. 3 and top paragraph on page 5), as well as with the zeolite ZSM-5 with the clear distinction between Si and O. The refinement program has no reason to make Si positive and O negative, but it does, even when starting with opposite partial charges into the refinement. Hence the information is in the data, and not just some coincidence. To explain the comparison with computation better, we refined Figure A2 based on the referee's comment. The result is described in Sec. 2.6, page 11, starting line 10:

The poor performance of the partial charges without the inclusion of iSFAC treatment for hydrogens may be attributed to an imbalanced dataset, where the majority of non-hydrogen atoms exhibit negative charges. As a result, a linear regression focused solely on non-hydrogen partial charges is inherently more constrained compared to one that includes all partial charges. To address this issue, the average charge of hydrogens was calculated for the experimental partial charge sets not containing hydrogen atoms, effectively

counterbalancing the charges of the non-hydrogen atoms. A Pearson correlation analysis was then re-conducted using an augmented dataset that included both hydrogen and non-hydrogen atoms, represented by the orange boxes in Fig. A2. While incorporating the average hydrogen partial charges significantly improved the Pearson coefficients, the green boxes still demonstrated superior correlations, as indicated by higher average Pearson coefficients, reduced spreads of the corresponding boxes, and lower standard deviations. The success of the average hydrogen partial charges (orange boxes) can be attributed to the relatively narrow range of hydrogen partial charges ($0.1e \leq q_H \leq 0.4e$). Consequently, the actual partial charges of hydrogens do not deviate substantially from the computed average for all hydrogen atoms. However, the partial charges of non-hydrogen atoms are considerably influenced by whether or not hydrogen atoms are included in the analysis. Specifically, the average Pearson coefficient of non-hydrogen partial charges between these two partial charge analyses is 0.63 in histidine, underscoring that the iSFAC model, which incorporates hydrogen atoms, not only improves the accuracy of hydrogen charges but also enhances the reliability of partial charges for all other atoms.

Concerning ZSM-5: zeolites are an example of compounds that are unsuitable for QM computations, they are too complex (organic compounds are “easy”).

The referee’s comment was very helpful to make us add this intermediate step of the analysis.

3. The authors claimed that “the ESP map is the first result for quantum chemical calculations, while partial charges are a derived quantity” and removed the comparison of ESP maps in the previous Figure 7. However, experimental ESP data is the primary result, and all values, including x, y, z coordinates, B-factors, and charges, are derived from the experimental ESP data. People would expect ESP to be a straightforward and correct metric for evaluating results directly from electron crystallography. Although the authors also claimed that electron-density-based calculations offer values closer to those from electron diffraction, QM-based values are just the results of how charges are assigned to each atom.

I think ESP maps should serve as a validation and the EESP and QNESP maps should exhibit greater similarity. I would like to know if the contours in the previous maps represent VDW surface rendering. If I understand correctly, the colors in the previous Figure 7 would represent potential values on the VDW surface from point charges assigned at the nucleus. If so, the additional explanation regarding the differences between the crystallographic and single-molecule environments would not be justifiable. The authors should provide more detailed descriptions.

Response: We do not fully comprehend this comment. It is a shame that people would expect ESP to be a straightforward metric. Probably this is due to Tamir Gonen who, in a large number of his publications, uses the expression “the cryo-electron microscopy

method microED". This may confuse people into thinking that microscopy and crystallography were similar in their procedures, which is of course not so. We highlight the difference between electron crystallography and electron microscopy. (Macromolecular) electron microscopy (EM) results in an experimental electrostatic potential map, into which a model is built. During the model building process, this EM map does not change. This is very different from crystallography. In crystallography, the data are the intensities of the reflections. In order to calculate the electrostatic potential map, phases are needed in addition to the experimental intensities, and these phases are computed from the chemical model. Therefore, the map improves as the model improves: the map is only indirect information.

In addition to the fact that the electrostatic potential map is only indirect information, a graphic presentation of a crystallographic map is usually affected by subjectivity, as there are parameters that the creator of the picture needs to decide on, in particular the colour level. The comments from both reviewer 1 and reviewer 2 in the previous submission pointed out how subjective this step can be: for us, the ESPs showed, qualitatively, a reasonable match between QM and experimental map, while both referees pointed at the areas of discrepancies. Because of this subjectivity, we removed the map from the manuscript. We kindly ask the reviewer to respect that this figure is no longer part of the manuscript.

We added the following to the manuscript, page 11, final paragraph of section 2.6:

When comparing iSFAC modelling with computational methods, one needs to bear in mind that a comparison of the electrostatic potential maps is tampered by the fact that a crystallographic map depends on the phases, which are computed from the model, and are not part of the experimental data. This is different from electron microscopy, where the model is created from the map, and where the map is not influenced by the model.

4. Why did the authors use calcium tartrate tetrahydrate for the X-ray diffraction experiments? This compound is chemically quite different from the other examples presented in the manuscript. The samples presented in the manuscript should be tested instead.

Response: X-ray scattering factors for ions differ only very little from X-ray scattering factors for neutral atoms. Hence, the suggested comparison was bound to fail, because of the extremely high correlation between these two types of scattering factors. We chose Ca tartrate, because we have a very good data set at hand (we use this compound routinely for calibration of our single crystal X-ray diffractometers), with very high multiplicity, and because Ca(II) was a promising candidate with a strong difference between Ca(0) and Ca(II). If the suggested experiment would work, it would work with such a data set. Despite the very high data quality, iSFAC modelling against X-ray data let the model parameters explode, yielding no result. Tartrate is actually not so much different from ciprofloxacin: both are organic ligands with an metal ion as counter-ion (Cl^- and Ca^{2+}).

5. In the first-round review, I asked which scattering curve was used for H⁺. If it is based on Yonekura et al., IUCrJ, 2018 [37], the difference between H⁰ and H⁺ is not small but quite large. The treatment of hydrogen seems problematic. If the difference is indeed so small, why is the refinement of hydrogen atoms unstable when using H⁰? The weight terms of H⁺ in Eq. (5) approaches 1 when the difference is small. The authors stated, “experimentally, we attribute this observation to the fact that the electron, which would make H⁺ neutral, is already refined as part of the donor atom”. I do not understand why charges in the donor atoms are refined prior to those in the hydrogen atoms. Is it possible to refine hydrogen atoms freely using only the scattering factors of H⁺ and other neutral atoms? The proposed scheme requires more thorough examination and justification.

Response: The referee is correct, the scattering factors between H^+ and H are very different, our original response was incorrect. The special treatment of H in iSFAC modelling is based on observation rather than theoretical considerations. To underline, how we developed the concept of iSFAC modelling, we rewrote the Section 3.4.2 “iSFAC modelling: hydrogen atoms”, pages 14/15:

In course of our work we realised that hydrogen atoms have to be included in iSFAC modelling. Otherwise, the resulting partial charges are not very meaningful (cf. Fig. A2, dark red boxes, and Section 2.6). However, when hydrogen atoms are refined as linear superposition, similar to all other atoms with Eq. 4, the refinement does not converge and stops as unstable. Possibly, the scattering factor of H^0 is too weak for superposition with the scattering factor of H^+ . We needed to modify Eq. 4 to include hydrogen atoms in iSFAC modelling:

$$F_{\text{calc}}(hkl) = \sum_{\text{non-H atoms } j} (\nu_j f_j^{\text{ionic}}(s) + (1 - \nu_j) f_j^{\text{neutral}}(s)) e^{-8\pi^2 U_j(s)} e^{-2\pi i \vec{h} \vec{x}_j} + \sum_{\text{H atoms } j} \nu_j f_j^{H^+}(s) e^{-8\pi^2 U_j(s)} e^{-2\pi i \vec{h} \vec{x}_j}$$

Chemically, we attribute this observation to the fact that the electron, which would make H^+ neutral, is already refined as part of the donor atom. E.g., the valence electron donated by H to a C-H bond is included in the C^- ionic scattering factor, even though this may be a naïve explanation. This treatment revealed that iSFAC modelling stabilizes H-atoms, and their parameters can be freely refined without constraints (possibly supported by mild restraints).

We also express the sentence quoted by the referee slightly differently (“chemically” instead of “experimentally”), as this sentence refers to a chemical interpretation of our observation. The justification of the proposed scheme lies in the observation that a linear superposition is not stable, at least with the mechanisms available in SHELXL. Hopefully, the rewriting of the section clarifies the situation better.

6. Please explicitly specify the literatures from which the ionic scattering curves for hydrogen and all atoms are adopted. The description about the GNUPLOT scripts refers to

refs. [39] and [62], but some essential scattering curves, such as those for H^+ , C^- , and N^- , are missing in these references. Ref. [37] includes these curves.

Response: We answered this question in the first round, but since it is now repeated, we considered that maybe there is something behind this question that at first we did not realise. After thinking more about this remark, I believe it is the interpretation of Eq. (4) in Yonekura et al (IUCrJ, 2018, which we repeat in section 3.4, page 13, Eq. (2)). The referee might interpret $f_X(s)$ in this formula to be the scattering factor of the ionic species. This would explain why the referee is asking for the reference of ionic X-ray scattering factors for H^+ , C^- , and N^- , which we did not use. In Eq. (2) of our manuscript, $f_X(s)$ refers to the X-ray scattering factor of the neutral species (which are tabulated in the Int. Tables Vol. C, Table 6.1.1.1). The beauty of the equation published by Yonekura et al. (IUCrJ, 2018) is that it provides an elegant work-around about the fact that not all ionic X-ray (or electron) scattering factors are accessible. Quoting Yonekura et al: (IUCrJ, 2018): " ΔZ is the ionic charge and is defined as $\Delta Z = Z - Z_0$ ". The ionic state is expressed via Z , avoiding the need for X-ray scattering factors for ionic species, which are not available for all cases. If the referee assumes that, in addition to Z , $f_X(s)$ would be taken as the scattering factor of the ionic species, the total charge would be wrong. As example, consider O^- , where $Z_0 = 8$, $\Delta Z = -1$, i.e. $Z = 7$. If $f_X(s)$ were the scattering factor of O^- , it would contain 9 electrons, i.e. a charge of -9 . Thus, the formula would compute a scattering factor for a species with charge of $7 + (-9) = -2$, which is obviously not the wanted species. The same would apply for, say, C^+ with $Z = 6 + 1 = 7$ and $f_X(s)$ for C^+ with 5 electrons (charge -5), i.e., the formula would compute a scattering factor for a species with charge $7 + (-5) = 2$.

To avoid confusion, the manuscript by inserting "of the respective atom (not ion)" on page 13, Section 3.4, second paragraph:

"[...] is the X-ray scattering factor of the respective atom (not ion) evaluated at the magnitude of the scattering [...]"

At the beginning of this project, we made this mistake, until we realised that modifying both Z and $f_X(s)$ doubles the charge of the species.

7. The difference densities for hydrogen atoms appear similarly resolved, particularly for tyrosine, with or without iSFAC refinement. Hence, the authors should not claim that the visibility of hydrogen has been enhanced by this approach.

Response: Many people like omit maps, although they often are only pictures, and not a numerical support for a result. We thank the referee for the comment and realise now that the term 'visibility' may be misleading. In crystallography (both small molecules and macromolecules) atoms positions are refined in reciprocal space against diffraction intensities, and not placed based on actual maps. This is different from, e.g. electron microscopy (cf. response to item 3). Considering Figure A4 on page 25, we deliberately placed the figure into the extended data section. In particular for His, there are either more green blobs or larger green blobs in row (e) than in row (f), while for Tyrosine, the differences are less obvious, but still visible. We do refer to Figure A4 in the results

section, page 4, but merely to support the actual observation, that the observation of H-atoms is enhanced. and that iSFAC modelling enables the refinement of coordinates and U-values for *H*, while conventional modelling usually uses the AFIX constraints for *H* atoms, where U-values and coordinates are computed from the neighbouring atoms.

We rephrased this part at the end of the Results section, page 4 as:

[...] and it reveals enhanced structural details including the possibility to refine coordinates and ADP-values of hydrogen atoms (see Fig. A4), which is rather associated with dynamical structure refinement [20].

To clarify this in the manuscript, we changed the caption of Fig. A4 from

"[...] show enhanced hydrogen visibility with iSFAC [...]"

to

"[...] illustrate the enhanced signal of hydrogen with iSFAC [...]"

The title of the section A.4 was changed from "Enhanced visibility of hydrogen atoms (Tyrosine and Histidine)" to "Enhanced role of hydrogen atoms (Tyrosine and Histidine)"

8. This method depends on full-matrix least-squares refinement of hydrogen atoms, which is likely unstable for crystals of more complex molecules such as higher molecular weight. These samples require restraints or constraints for hydrogen atoms. However, this approach is not compatible with the standard command AFIX in SHELXL. Therefore, making strong claims about its versatility would not be appropriate.

Response: It is not clear why the referee believes that full-matrix least-squares refinement was not available for more complex molecules. Kai Tittmann and Piotr Neumann used full-matrix least-squares to get standard uncertainties of a carbene in a thiamin enzyme (10.1038/nchembio.1275). At that time, the available RAM of computers was the limiting factor, but not the complexity of the molecule. The PDB entry 4KGD list 603 amino acids.

Similarly for AFIX.

The need for AFIX is not so much related to the complexity of the model, but the data to parameter ratio (which is closely linked to the resolution of the data set) and data quality. If resolution is not sufficient, restraints and constraints can be added to stabilise the model being refined.

That said, it is not clear why AFIX should not be compatible with iSFAC modelling. iSFAC modelling refines a free variable for each atom/ion pair, and a free variable for the modelled H⁺ ion. With AFIX, the crystallographer can fix atom coordinates, and can also fix the ADP value. Whether or not the crystallographer decides to do so, the free variable in the occupancy of the respective ion is unaffected by AFIX. AFIX has two parameters. The first one can be used to compute the coordinates (and ADP value) of an atom based on the geometry to its next neighbours (probably this is what the referee refers to), the second parameter controls how much freedom is given to the computed

parameters. They can remain fixed (i.e. constrained), but they can also be freely refined. Independently from the use of AFIX, any atoms, also H-atoms or H^+ ions can be restrained by geometry commands like DFIX, DANG, SADI, or FLAT. Some of these, like SADI or FLAT, do not even need absolute values. Where distances are required, like for DFIX or DANG, values from neutron diffraction could be used, e.g. as published by one of the authors

(<https://homepage.univie.ac.at/tim.gruene/research/shelxl-nmx/nmx-hydrogen-restraints.dfx>)

Several people have been using neutron distances (including us in our *Angewandte* paper <https://doi.org/10.1002/anie.201811318>), but of course further development may produce restraints better tailored for electron diffraction. It needs to be born in mind the 3D electron diffraction can still be considered a young field, with active development in many directions.

In order to make the option better understandable to the reader, we added a reference to P. Muller's reference book on refinement with SHELXL in the section about hydrogen atoms, page 7, third line from the bottom:

[...] enables their free refinement, like any non-hydrogen atom. When needed, e.g. when data resolution is not quite atomic, mild restraints can be applied [24].

Even more do we anticipate that our work will stimulate the implementation of iSFAC modelling in modern programs like Olex2. Hence, developers will find suitable values, and not the users need to decide on these details.

9. I do not think the authors properly addressed my concern: "Pacoste et al. (IUCrJ, 2024) have recently demonstrated that scattering curves for ligand-F(III) ...". This suggests that the theoretical scattering curves may not be optimal, and the refinement is heavily influenced by their shapes. Also given the low Pearson coefficients without hydrogen charges, it would be appropriate to avoid making strong claims about the determination of the absolute partial charge values.

Response: The concern seems "that the theoretical scattering curves may not be optimal". We do not claim that the scattering curves are optimal, but we used those that are available. We are sure that, in case better scattering curves become available, also iSFAC modelling will benefit from such future developments We amended Section 3.4, page 13, last sentence:

[...] GNUPLOT [43] for fitting. Future development may result in even better scattering factors, leading to better results from iSFAC modelling [44].

The partial charges are on an absolute scale based on the scattering factors, as explained before to referee 2. Their uncertainties are comparable to other parameters determined from crystallography, like distances between non-bonded atoms. To us, the claim does not go beyond what crystallography usually claims with respect to the accuracy of the parameters determined by crystallography.

Considering the Pearson coefficient we hope that we clarified the concerns of low and high Pearson coefficient with the improved Figure A2, and discussion in Sec. 2.6, as described in response to item 2 above.

10. The Rscat values in the new table are quite high, with some exceeding 1 - 3%. These poor fits likely introduce significant errors in estimating the contribution from ionic scattering factors. Claiming the determination of absolute partial charge values would also be too strong for this reason. Does the range 0.7 – 19 in the table refer to 0.7 Å to 19 Å?

Response: We were alarmed by this comment and double-checked our calculations. The highest numbers were due to typo's, e.g. we inserted 7 for Z0 of C, or 6 for Z0 of N. The table has been corrected, and the actual values are much lower. In addition, we added a column that better illustrates the noise in the fit. R_{scat} is accumulative, i.e. adding more data points increases its value, even if the data were getting better. Also, R_{scat} does not differentiate between systematic and random errors. Random errors, i.e. fluctuations of the values for $f_{\text{CM}}(s)$ about the values of $f_{\text{MB}}(s)$ along s would not lead to the same level of errors as systematic errors. Table S10 list this value (we named it $R_{\text{nl}} := \frac{|\sum_s f_{\text{MB}}(s) - f_{\text{CM}}(s)|}{\sum_s |f_{\text{MB}}(s)|}$ as in "noise level", as opposed to $R_{\text{scat}} = \frac{\sum_s |f_{\text{MB}}(s) - f_{\text{CM}}(s)|}{\sum_s |f_{\text{MB}}(s)|}$), and the values are between one and two orders of magnitude lower than R_{scat} , indicating fluctuations rather than a systematic difference. **Note that Table S10 lists R_{scat} as %, while R_{nl} is given as ‰.**

We think these error levels are not too bad, considering these are crystallographic data. Even errors of 1 %–3 % are not so rare in crystallography. E.g., the residual $R1$, is hardly ever that low — the CCDC currently contains 164489 structures with an R-factor better than 3 %, with a total of 1'325'975, i.e. only about 12 % of all structures have an error better than 3 %.

Yes, the range of resolution is given in Å. We added the unit to the three lines in the table.

11. Please give serious consideration to the B-factor variation test proposed by Reviewer #2, conducted at room and cryogenic temperatures. I think this test could be highly effective in distinguishing between the influences of partial charges and B-factors.

Response: Collecting data at room temperature for the organic compounds was not possible since the compounds suffered too much from radiation damage. We now recollected data for zeolite ZSM-5, both at room temperature and at -110°. The comparison is shown in Figure A7. The inset shows the estimated standard uncertainties of the free variables associated with the respective partial charge as upper curve, and the lower curve shows U_{eq} (the term used in small molecule crystallography, $B = -8\pi^2 U_{\text{eq}}$). As expected, the esu's are lower for data collected at cryo temperatures, and as expected, the U_{eq} tend to be smaller for data collected at cryo-temperature. For the partial charges, we do not see a clear trend, some values seem a little higher, some seem a little lower, but not to an extent that would be outside the typical variation of crystallographic parameters, and we added this to the manuscript, page 10, paragraph before section 2.6:

This also applies to the temperature of data collection, since cryogenic temperatures generally improve data quality and protect better against radiation damage. In case of the zeolite ZSM-5, we collected data both at room temperature and at -110° C. Except for the expected variation, and reduction in estimated standard uncertainties, there is no significant trend for the partial charges (Fig. A7).

These data were collected with the upgraded JUNGFRU detector, which is part of an ongoing research project, therefore, the respective researchers, K. Ferjaoui, E. Frojdh, A. Mozzanica, and K. Takaba have been included in the list of authors.

12. Multiple scattering would affect charge estimation, as Reviewer #2 concerned. Yet, the author did not address this issue except for discussing data scaling. It should be discussed in more detail, both in the main text and in the response to the reviewer.

Response: Dynamic diffraction is a theory that describes the intensities of a diffraction experiment. It is more detailed than kinematic diffraction, which is an alternative description of the diffraction experiment. In X-ray diffraction, the kinematic theory has been used for a long time, and researchers have become used to the theory (sometimes without knowing). “Getting used to” includes a feeling for error levels, reliability and the experience to differentiate between good data and poor data quality. One often sees the expression of ‘single scattering’ as synonym for the kinematic theory, and multiple scattering as synonym of the dynamic theory. In electron diffraction, the deviation of measured intensities from the intensities calculated based on the kinematic theory is much greater than in case of X-ray diffraction. This results in (classical) quality indicators for electron diffraction experiments being much worse than for X-ray diffraction experiments. This applies both to data quality indicators and model quality indicators. It is worth noting that it has been much discussed that most quality indicators, like R_{int} , etc, are measures of precision, rather than measures of accuracy. The programs that we used in this work (XDS for data processing, SHELXL for refinement) both are based on the kinematic theory of diffraction. Hence, values like R1, wR2, GooF, etc. are generally greater than what crystallography experience from X-ray diffraction experiments. Nevertheless, very detailed information can be concluded from electron diffraction data even when using programs that only use the kinematic theory of diffraction. We demonstrated the sensitivity of our hardware and software with the reference for the first version of the JUNGFRU detector by the discrimination of aluminum from silicon in ED data, despite R-values that X-ray crystallographers might call “horrendous” (DOI 10.3390/cryst10121148).

It would be an issue, if, at the present stage, it would be possible to provide more accurate results than what we present. In the lack of knowing what is correct, however, this a pure speculation. There are of course programs that are based on the dynamic theory of diffraction, notably Lukas Palatinus’ JANA, and, recently, also Oleg Dolomanov’s Olex2. It will be a great recognition of our work, when our results will be implemented into these programs and thus be made available to the community. For the moment, the steps we undertake in SHELXL, while not being very complex, require much diligence

to assign all the free variables correctly (we double and triple checked this in the course of the work).

Using the model of multiple scattering, it can be said that strong reflections “lose” intensity compared with the kinematic model, and weak reflections “gain” intensity compared with the kinematic model. Since strong reflections dominate the low-resolution region, and reflections get weaker towards high resolution, SHELXL would overestimate strong reflections, and underestimate weak ones. The algorithm could compensate for this with a reduced ADP value, but this would affect all atoms. Another scenario would be that SHELXL emphasises negative ionic scattering factors and reduces the impact of positive ones. This scenario, however, is falsified by the result on ZSM-5, because in this scenario, all atoms, including T-sites should be preferred with negative ionic scattering factors. However, we can invert the sign of the scattering factor of T-sites, starting refinement from a negative partial charge assignment but refinement with SHELXL pulls it back into the positive, showing that the data really want this distribution between negative and positive charges, and not in order compensate for some unrelated effect.

All this, however, is purely speculation, which would be suitable for a review, but not so much for a research article, which is meant to be concise and meant to report results, rather than speculations.

1.1.1 Referee #1 (Remarks on code availability)

Now the GitHub addresses are included in the manuscript for codes including those for Pearson correlation coefficients, Cromer-Mann parameters and Rscat values. I am reviewing part of these codes. I would also like to check the codes and scheme used to calculate the EESP and QNESP maps in the previous Figure 7.

Response: As stated above, we would like to ask the referee to respect that the ESP maps are no longer part of the manuscript, as they led to confusion amongst the referees.

1.2 Referee #2 (Remarks to the Author)

The authors have meaningfully addressed my concerns regarding the resolution dependent definition of partial charges, and regarding the robustness of the partial charges towards changes of the resolution. Although the partial charge definition issue has not been addressed in a principled way, the authors make a convincing case that for resolutions better than 1 Å, their partial charges are almost resolution-independent, which is sufficient for these charges to be meaningful. In practice, the need for diffraction data to better than 1 Å resolution limits the method to crystals of small molecules, and makes it inapplicable to most if not all electron diffraction data of macromolecules. Importantly, the authors are now clear about when their method is not applicable (or not reliable). Despite the clarified limitations (that rule out some of my “favorite” use cases), the new method is still an important advance that deserves prominent publication.

Response: We appreciate the referee’s comments and are very pleased to hear that we could bring across the value of our work. We would not exclude iSFAC modelling from macromolec-

ules. The refinement of macromolecules is quite different from small molecules, mainly due to the need of restraints. With SHELXL, combining iSFAC modelling and restraints, it would be a lot(!) of bookkeeping. However, for developers like Garib Murshudov (REFMAC5) or Pavel Afonine (phenix), it should be possible to do, and thus iSFAC modelling might, in the future, also become available for macromolecules.

1.2.1 Referee #2 (Remarks on code availability)

The documentation of the workflow with code is much improved. It is very confusing, however, that the main branch of the github repo (<https://github.com/CF-CSA/edSFAC>) for the code is actually empty (except for a license), whereas the actual code is placed in other branches. I realize that that this is due to github's convention changes regarding "main" and "master" branches (a problem for many repos and not the authors' fault), but still, it would be good if something could be done about it.

Response: We fully agree. The two branches have different histories, because "main" was created as empty project with the github project and "master" is the original repository from the author's local work. This prevents merging of the two branches. We now populated "main" from a copy of "master" and pushed it to the github server. Now, the scripts and codes are directly visible on the github site, without the need to select a different branch.

1.3 Referee #3 (Remarks to the Author)

My opinion of this submission remains, in that the authors present a novel approach to experimentally determining partial charges and that this will have widespread impact, providing new insights for a broad range of chemistry and materials science.

I did feel that it was disappointing that the original version didnt include an easy method for any crystallographer or chemist to implement this method. However, I did feel that is it so significant that the developers of the main crystallographic refinement software packages would quickly implement it. I am therefore very pleased to see that scripts are now being made available to generate the appropriate SHELX commands. This will enable some crystallographers to utilise this approach immediately, but will also make it much easier for developers to implement in their software.

The tweaks to wording throughout and streamlining of parts of the paper address many of the concerns I had and I feel the paper reads rather better now. I have carefully read the comments of the other reviewers and feel that the authors have removed distracting parts and included further evidence to back up their approach - most notably regarding resolution dependance.

I believe that this paper is considerably improved and now suitable for publication.

Response: We are very grateful to read that the manuscript as improved, and that the referee repeats their support for our work. Kiyofumi Takaba, now listed as co-author has, in the meantime, implemented a python script that prepares a SHELXL res file for iSFAC modelling. It is not yet part of the github repository, because we are going to test it a bit before. Maybe by the time of publication, this script is sufficiently stable (for the preparation of this manuscript, we

did not use it; however, if the script becomes available before the manuscript is fully accepted, we will be happy to add a reference to it).

1.3.1 Referee #3 (Remarks on code availability)

I have not been able to review the code. However I do note that further code has been made available - and that this is very important for the wider uptake of this method.

Revision 3 — Author response letter

Tim Gruene with all co-authors

18th March 2025

Dear Dr. Editor,

thank you very much for your consideration! We greatly appreciate the effort of Referee #2, for a clear and helpful overview of the open issues with our manuscript. The majority of questions could be answered swiftly, with respective improvements to the manuscript detailed below. Three questions were more difficult to answer, as we are concerned we might not understand completely - hence we aimed to answer as clearly and as complete as possible. We do hope that we understood the concerns correctly and that our responses address them in a satisfying manner. We conducted additional experiments, so that the manuscript received a valuable round-up improvements.

1 Referees' comments:

Referee #1 (Remarks to the Author):

I now understand the misunderstanding by the authors. The ionic scattering factors for electrons cannot be derived from the scattering factors of neutral atoms for X-rays, except for H^+ . In Eq. 2, Z does not represent electron charges but nuclear charges, or the atomic number (please carefully refer to Yonekura et al., IUCrJ, 2018, or Peng, Acta Cryst. A, 54, 1998). I plotted the electron scattering factors of O^- derived from the International Tables, calculated from the X-ray scattering factor of O^- in Yonekura et al., IUCrJ, 2018, and from Eq. 2 of this paper. The first two are identical, but the last one showed a significant discrepancy (see the attached Plot 1). I also plotted the scattering factors for O^+ , O , O^- (Plot 2), C^+ , C , C^- (Plot 3), and H^+ (Plot 4) using the Cromer-Mann coefficients (a_1 , b_1 , a_2 , b_2 , a_3 , b_3 , a_4 , b_4 and c) provided in the cif file (2365033.cif) for one tyrosine dataset used in this paper. The plots for O and C matched the curves from the International Tables, but the others showed unusual shapes. The H^+ curve from Yonekura et al., IUCrJ, 2018 matches the neutral scattering curve in higher resolutions, consistent with other atoms, while the curve used in this paper remains elevated at middle to high resolutions. Therefore, the former curve is likely correct. I then reviewed the Cromer-Mann coefficients in most of the cif files and found that all the values were either identical or very similar. The proposed scheme probably worked because the authors used

their own scattering factors. Due to these incorrect ionic scattering curves, it is clear that all the calculations and results presented in this paper are fundamentally flawed.

Response: Please refer to our response to Referee 2

Additionally, I am concerned that the authors have not provided direct and proper answers to my previous comments, instead offering unrelated and overly general explanations. Please address my comments directly and appropriately. For example, regarding my previous comment 3, my concerns are unrelated to comparisons between crystallography and microscopy. Though the ESP maps have been removed in the current version, I still wish to understand how the ESP maps were generated. Please provide a clear and direct response to at least this point as part of my previous comment. Regarding comment 1, I am not referring to macromolecular data. Again, the correct electron scattering factors of neutral atoms and ions differ only in the low-resolution ranges but overlap from around 2 Å and higher resolution ranges. Even if the low-resolution parts are limited by lattice parameters, the data still include diffraction spots at lower resolutions and it is important to examine these regions, as charge information cannot be obtained from data in the higher-resolution range. I had hoped the authors would carefully consider and provide correct answers, but unfortunately, their responses have included errors and subsequent sequential corrections. I am now inclined to refrain from providing further feedback on their other responses, as commenting on results derived from incorrect scattering factors seems meaningless.

Referee #1 (Remarks on code availability):

No additional review this time. My previous comment regarding the calculation of ESP maps remains unchanged.

Referee #2 (Remarks to the Author):

As I felt that my most important concerns had been addressed in a previous revision, I am only commenting on the discussion between the authors and referee 1, as requested by the editors.

Response: This summary was a lot of work. It helped a lot to understand the raised points, and it helped to find good answers and to make the manuscript richer. We greatly appreciate the referee's effort.

1) investigation of the low resolution cut-off:

Referee 1 has a point regarding the low resolution differences between the form factors for the neutral and charged forms of atoms.

For MACROMOLECULAR structures, this is a serious concern. However, as already clarified by the authors in their response to me, their method is essentially inapplicable to macromolecular structures anyway, simply because the results become independent of the HIGH resolution limit only in a regime that is hard to reach for macromolecular structures. The problem with the LOW resolution cutoff spotted by referee 1 is yet another reason for why the method would be inapplicable or at least problematic for macromolecules.

For SMALL molecule crystal structures, the low resolution question seems irrelevant at first. As the authors rightly point out in their response, a low resolution cutoff is simply not applied. All reflections except for the 000 reflection (that coincides with the direct beam) can and should be measured.

Response: We appreciate this thoughtful consideration. Maybe our explanation w.r.t. low-resolution was too simplified: In ED, it is not always possible to collect all data, since the goniometers are not as flexible as those for X-ray diffractometers, and because the crystals cannot be manipulated as easily in their orientation on the TEM grid. Rightly, of course, all effort should be made to collect as complete data as possible with as high quality and multiplicity as possible. Our finding may stimulate further development in this direction.

However, there might be a problem that both the authors and I have overlooked that referee 1 seems to have in mind. If the SAME small molecule is crystallized in two different crystal forms that BOTH diffract to very high resolution, would the partial charges determined by the authors' method be the same? The authors take it for granted that the answer would be "yes". Referee 1 seems to expect that the answer would be "no", because the effective low resolution limit is different for different crystal forms (provided cell constants and thus effective lower resolution limits are different).

Response: In case of different unit cell dimensions (and thus different low resolution limits), the chemical environment of the molecule within the crystal lattice is different. Thus we expect the answer is 'no'. Rather than a major problem, but consider this a powerful feature of iSFAC modelling. We hope that the modifications (see below) will convince the referee of this strength of iSFAC modelling.

I don't have a clear expectation whether the "yes" or the "no" is correct. Therefore, I suggest that the authors address the question experimentally (or computationally): if indeed experimental partial charges depend on the crystal form even for well-diffracting SMALL molecule crystals, this would be a MAJOR problem for the method. Thus, the referee's concern about the effective low resolution limit should be taken seriously.

Response: The referee is concerned about the reliability/ reproducibility of partial charge values and how the values are affected by low resolution data points. In particular, the referee refers to the (common) case of two different crystal forms, that naturally have different low resolution reflections, but in value and in number. We amended the manuscript to cover two aspects of this concern: (1) a second crystal form of histidine with different unit cell constants is analysed, and (2) the possibility of missing low resolution reflections is analysed:

1. Considering different low resolution reflections because of different unit cell constants, we collected additional data for L-histidine, which crystallises in spacegroup $P2(1)$ and a c-axis about half the length compared with the $P2(1)2(1)2(1)$ polymorph ($V=350\text{\AA}^3$ instead of $V=733\text{\AA}^3$, and the lowest resolution reflection at 7.42\AA rather than 9.47\AA). These different low resolution limits will have an effect on the (expected) different partial charges of histidine with two different chemical environments. This is actually one of the strengths of iSFAC modelling. To clarify this strength, we added to the conclusions, page 20: (modification in italics):

A significant advantage of iSFAC modelling is its ability to determine partial charges within the true chemical environment of the molecule, cap-

turing essential chemical interactions rather than modelling the molecule as an isolated entity in a vacuum. *Different crystal forms have different chemical environments, resulting in different partial charges. This provides much deeper understanding of molecules and their functional groups than a conventional crystal structure.*

The comparison between the two crystal forms of histidine is found at the end of Section 2.2.2 together with Fig. A1 on page 23:

We also collected data from a different polymorph with spacegroup $P2_1$ and a c-axis half the length of the first polymorph. The chemical environment in this polymorph is similar, but not identical. Consequently, the partial charges are similar, but not identical (Fig. A1). This observation highlights the sensitivity of iSFAC modelling not only to the molecule itself, but also to its chemical environment.

2. To investigate the effect of low resolution reflections (independently of the crystal form), we chose a rigorous approach: for the $P2_12_12_1$ form of histidine with 1710 reflections, we performed 1710 refinement runs, each with one particular reflection omitted (such a leave-one-out procedure is part of the computation of R_{complete}). This captures not only the low-resolution reflections, but the effect of any reflection in the dataset. The result is added to section 2.5 “Robustness and resolution dependence” page 10, together with Fig. A4 a) and b) on page 26:

Another indicator for the stability of iSFAC refinement results from the computation of R_{complete} [29], where each reflection in turn is omitted from the refinement (leave-one-out cross validation). In case of histidine, this results in monitoring the effect of each of its 1,710 data points on the values of the partial charges. The boxplot Fig. 4a shows $\pm 3\sigma$ boxes about the mean value $\langle \delta q \rangle$ for each atom, as well as the respective maximum and minimum values. The $\pm 3\sigma$ boxes are tiny, as the variance of the partial charge is minute. The other plot, Fig. 4b shows the same type of analysis, not for the partial charge values, but for the atomic displacement parameters U_{ij} in case of a conventional refinement without iSFAC modelling. The spread of maximum and minimum outliers for U_{ij} compares with those of the partial charge values, showing that missing certain reflections does not affect the reliability of iSFAC modelling beyond a standard crystallographic refinement.

While the σ range is very tight around the mean value, apparently some individual reflections have strong impact on the resulting value. At first, we were alarmed considering the reliability of values: if the wrong reflection could not be measured, quite different partial charge values might be obtained. However, we never saw such type of analysis before, on any crystallographic parameter, not just those on partial charges. Therefore, the same plot was prepared, but this time for a conventional refinement (without iSFAC modelling), and the boxplot is generated for the U_{ij} values of each

atom. As can be seen in Fig. A4a and A4b, the maximum and minimum outliers for U_{ij} with conventional modelling exceed in parts the outliers for partial charge values from iSFAC modelling. Hence, the variations shown in Fig. A4 are intrinsic for crystallographic refinement, and do not compromise the accuracy of the crystal structure beyond the common precision of crystal structures.

2) validating extracted charges:

Referee 1 is correct that the low correlation between calculated and experimental partial charges is problematic. I also agree with the referee that including hydrogen charges yields higher correlation values almost inevitably, as long as any (remotely reasonable) positive values are used. In contrast to Referee 1, I am not surprised by the discrepancies, in fact I expected systematic discrepancies and only qualitative agreement. As I pointed out earlier in a previous refereeing round, the authors effectively introduce a NEW definition of partial charge, which is DIFFERENT from the (already many) other partial charge definitions in use in the quantum chemistry community. For me, it was sufficient that the partial charges are qualitatively reasonable, as long as they are STABLE and not dependent on experimental parameters like the resolution limit. For the HIGH resolution limit, the authors have defined conditions when this is the case. For the LOW resolution side, the situation is unclear and needs to be addressed (see point 1).

Response: We appreciate the view and the summary of the referee. We appreciate that the referee acknowledges and accepts our efforts to demonstrate stability of the results with respect to the high resolution limit (See previous answer and the results of R_{complete} refinement). We sincerely hope that the additional experiments included in this revision address the concerns about the low resolution side to the satisfaction of the referee. The following addition on page 16 defines the term partial charge used in this manuscript:

The fraction ν_j of the ionic scattering factor for the j^{th} atom in Eq. 3, and the respective charge offset ΔZ from Eq. 2 that was used to calculate its ionic scattering factor $f_j^{\text{ionic}}(s)$ define the partial charge of the j^{th} atom as

$$\delta q_j = \nu_j \Delta Z \quad (1)$$

3) ESP maps as the primary experimental result:

Referee 1 has a point that theoretical ESP maps calculated from the authors' partial charges should reproduce experimental ESP maps. Moreover, there should be at least reasonable agreement with ESP maps from partial charges that are calculated according to definitions that aim to make partial charges useful to calculate electrostatic potentials. I agree with the authors that ESP maps from electron diffraction are model dependent, if only because phases are model dependent. However, I do not understand how this answer addresses the referee's concern (which I also do not fully understand).

Response: The discussion and interpretation of the ESP maps seems a little subjective and causing misunderstandings. In our eyes (surely biased as authors of the manuscript),

the experimental ESPs and the calculated ESPs are quite similar in many parts of the molecules, and differences make sense considering the different chemical environments: The experimental map reflects the contacts within the crystal lattice, whereas the QM map is calculated for an isolated molecule. The ESPs show negatively charged carboxylates, positively charged amine groups for both histidine and tyrosine. The N-H of the imidazole group in histidine is less positively charged in the experimental ESP, which is plausible, because it is compensated by the H-bond to the carboxylate of the next histidine molecule in the crystal structure, while the QM-map does not have such a neighbouring molecule that could compensate the positive charge at the N-H group. In both histidine and tyrosine, the residue side chain seems less charged overall, which is plausible for a molecule embedded in the network of a crystal structure. Charges want to be compensated. The QM ESP presents the isolated molecule before such compensation, the experimental ESP present the real molecule in its chemical environment of the crystal lattice. The figures of the ESPs are now back in the manuscript, including a detailed description of how they were generated (Section 3.6 "Generation of electrostatic potential maps", see also remark (13) below).

4) Comparison with X-ray diffraction:

I agree with the authors that charge effects in X-ray data are small. Attempts to measure partial charges based on X-ray diffraction would likely lead to results with large error bars. However, I don't quite see why the authors should be reluctant to do this experiment: it would likely demonstrate the superiority of electron diffraction over X-ray diffraction for the determination of experimental charges, which would strengthen the manuscript. If one WANTS to see charge effects by X-ray diffraction, then the choice of calcium tartrate tetrahydrate makes sense.

Response: It appears that our response was not clear. We were not reluctant to do the experiments, we did carry out the suggested experiment with a data set of Ca tartrate. However, we did not succeed, because the refinement program SHELXL would not produce any output - it would stop with the statement "REFINEMENT NOT STABLE". We now looked back into the data and tried to at least avoid this situation with introducing additional and much stronger restraints. We managed to avoid unstable refinement by tightening some restraints, yet, the refinement does not converge (non-zero shifts for most parameters reported by SHELXL), despite 15'000 cycles of refinement. At this point, many of the partial charges exceed +1e or -1 by far, which is non-physical. This is presented in extended Fig. A3 in the manuscript. The non-ending drift is linked to a number of parameters that are strongly correlated, e.g. several coordinates of hydrogen atoms are correlated to more than 90 %. In contrast, for the ciprofloxacin data set, the highest correlation, as reported in the SHELXL lst file, is 0.6 for the z-coordinate of H1 and the U_{11} value for O3. With the X-ray data of Ca tartrate, there are over a dozen of pairs of parameters with a correlation greater than 0.9 or and anti-correlation less than -0.9. We provide the respective CIF file for the referee's convenience. We collected ED data for Ca tartrate, also included iSFAC modelling on X-ray data for a published high resolution structure of histidine (so that results can be compared with our ED results for histidine) and we added a paragraph 2.6 "iSFAC's Reliance on electron diffraction over X-ray diffraction",

We also tested whether iSFAC-modelling would be suitable with X-ray data.

We chose a very high quality data set of Ca^{2+} tartrate, as well as a published high quality structure of L-histidine (CCDC ID KAGGEI, [32]). While with electron diffraction data, iSFAC modelling converges quickly to stable values for the partial charge, iSFAC modelling against X-ray data does not converge, leaving non-zero shifts of the parameters even after thousands of cycles, with partial charges much greater than $1e$, and negatively charged hydrogen atoms, neither of which is chemically plausible (cf. Fig. A6).

5) Hydrogen:

As the authors acknowledge, the referee is right about the big difference in the electron diffraction form factors for H and H^+ . Like the referee, I do not understand why one would refine partial charges sequentially. I have a hard time to follow the authors justification. In particular, I do not get the statement "Chemically, we attribute this observation to the fact that the electron, which would make H^+ neutral, is already refined as part of the donor atom."

Response: The method description in 3.4.1 and 3.4.2 used to present the chronologic order how we developed our idea of iSFAC modelling. Most likely this is why the referee speaks of "sequential" refinement of partial charges. This is not the case, there is no sequential refinement in iSFAC modelling. We removed the intermediate step and merged sections 3.4.1 and 3.4.2 into one section 3.4.1 "iSFAC modelling: quantification of partial charges". With this, we also removed the sentence quoted above, "Chemically, we attribute this observation to the fact that the electron, which would make H^+ neutral, is already refined as part of the donor atom." Hopefully this section is now better to understand.

6) Literature for ionic scattering curves, definition of form factors for cationic and anionic species:

Equation 4 of Yonekura simply associates the partial charge with the nucleus, and calculates the form factor accordingly. By contrast, I have the impression that the referee expects that the cationic and anionic species are analyzed -like the neutral species- by calculating an electron density quantum chemically (i.e. for the cationic species with one electron less and for the anionic species with one electron more than for the neutral atom). In fact, the referee's, rather than the authors' choice would have been mine. The electron diffraction form factors calculated in the two ways are NOT expected to be equivalent. Either choice can be justified, but the results will not be the same.

Response: The referees' choice also was our choice at first, several years ago, when we started working on partial charges. It seems obvious, because it is derived from physical principles and Coulomb's law. However, despite a lot of work, we never came to reasonable results. Since this has been several years, we forgot about the importance of this aspect. Thanks to the referee's clarification, this important aspect is now described in the manuscript page 14-15, supported visually with Fig. A7 (p. 29):

Eq. 2 has the advantage of being suitable for all ions, without dependency on computed X-ray scattering factors for ions like C^- or N^+ , which may not always

be available. Examples of the scattering curves $f(s)$ are shown in Fig. A7. Note the offset at high resolution between neutral and ionic scattering curves, which may contribute to the stability of iSFAC modelling. Originally, our first attempts with iSFAC modelling computed $f_e(s)$ with the classical Mott-Bethe formula, where the ionic charge is expressed by $f_X(s)$ of the ion instead through ΔZ . This, however, did not produce plausible results. In fact, modelling the oxygen of the hydroxyphenyl side chain of tyrosine with either the scattering factor for O^- or O^+ results in stable refinement, which would mean a negative or a positive partial charge depending on the crystallographers choice (data not shown). It was only when scattering factors $f_e(s)$ were computed based on Eq. 2 that iSFAC modelling would unambiguously result in a stable value for the partial charge for all atoms in the model.

7) Enhancement of hydrogen visibility:

The referee is right, but the authors' response seems ok.

Response: We acknowledge this opinion and won't take further action.

8) Least squares refinement of complex molecules:

The referee expects instability of full matrix least squares refinement for larger molecules. Assuming a constant packing density, the number of observations should grow like the number of parameters, provided the resolution for a given resolution. Hence, the observations/parameters ratio should stay constant, and refinement should not get unstable. However, this is only true if the resolution does not degrade with molecule size. In practice, the resolution WILL degrade with molecule size, and this will make the refinement less stable. In other words, the referee has a point that the method will not generalize too well to larger molecules. From the earlier debate, it is already clear that the method is not applicable to macromolecules in practice. From this perspective too, it is reasonable to expect problems with larger "small" molecules.

Response: For any data set, no matter its size, we would suggest to just try. According to Isabel Uson, maintainer of the SHELXL source code, there is no limit to the number of free variables in SHELXL. The reduction of data to parameter ratio, as observed for low resolution structures, is typically compensated with restraints, not only by SHELXL, but also by other refinement programs like phenix.refine, refmac5, busterTNT, If refinement becomes unstable, the program writes this to the output, and the operator can take measures to stabilise by introducing appropriate restraints. For the moment, where this step is not automated, this would largely increase the book-keeping and increase the risk of oversights due to human errors. To combine both the referee's and our view, we added a sentence to the conclusions (addition in italics):

With remarkable versatility, iSFAC modelling is applicable to any crystal structure resolved through electron diffraction at atomic resolution, spanning a broad range of compounds, including inorganic, bio-inorganic, and organic molecules.

What will be the ultimate limitation of application of iSFAC modelling, like the size of accessible molecules, remains to be determined.

We were actually surprised that iSFAC modelling works for e.g. for ciprofloxacin, and for the very radiation sensitive data for tartrate (only 30-40° of data per crystal), although radiation damage in tartrate required constraints on two partial charges.

9) Optimal scattering curves:

The referee is right: the theoretical scattering curves are certainly not optimal (see point 6). In fact, there is a major element of choice, even between the authors' preferred (Yonekura) method, and the referee's preferred approach. Any charges calculated with either method are valid for this choice of form factors. They will be different for a different choice of form factors. I would not have a problem with this, as long as the result is STABLE, i.e. the same for any given molecule, irrespective of crystal form and diffraction limit.

Response: We greatly appreciate the referee's opinion, and, as pointed out, our work may lead to a convergence about the discussion of appropriate form factors in the community, as anticipated by Richard Henderson long ago (cf. Acknowledgements). We address the stability with respect to the parameters crystal form and diffraction limit in this revision. We hope that the more detailed discussion in section 3.4 including the extended Figure A7 with curves of form factors, is an appropriate response

10) Rscat values:

Here, the referee caught an error that the authors and I had missed and that the authors appear to have corrected.

Response: We appreciate the confirmation and take no further action

11) Influence of B-factors:

The authors have carried out the experiment that I asked for earlier, and that referee 1 also wanted to see. The results suggest that partial charges deduced at room temperature and at cryogenic temperature are similar, eliminating the concern of a strong B-factor dependence of the partial charges.

Response: We greatly appreciate the approval of our experiments to address the stability of iSFAC modelling with respect to B-factors, and hope that our updates concerning the stability with respect to low resolution data (remark No. 1 and 12) and crystal thickness (remark No. 12) completes the experiments with respect to robustness and stability.

12) Multiple scattering:

The effect of errors of partial charges due to multiple scattering was one of my original concerns. It is true that the multiple scattering problem is not properly solved in general, and is a major concern in electron diffraction in general, not only for partial charges. Reviewer 1 is right that this concern was not thoroughly addressed so far. In an earlier round, I suggested

to compare partial charges for a small crystal (multiple scattering is less relevant) and a larger crystal (multiple scattering is a bigger problem). This experiment was not done. I gave it a pass, because I was more concerned about the resolution dependence of the partial charges. However, I concur with referee 1 that the experiment would be very useful to demonstrate robustness of the method, if indeed thicker crystals can be tolerated. The authors' response on this point is handwaving, and does not really address the issue meaningfully. Only an actual experiment with crystals of different size, of the same crystal form, of the same small molecule, could resolve this point satisfactorily.

Response: This comment helps us to better understand the concerns raised before: if two crystals of the same kind, but different thickness are measured, does the thickness affect the partial charges? We included these data with crystals from L-Histidine as extended Fig. A5 (page 27), described on page 10:

For electron diffraction data, the dynamical theory of diffraction produces significantly more accurate calculated intensity values, than the kinematical theory of diffraction [30], but only kinematical refinement is available with the program SHELXL, that was used for this study. The discrepancy between dynamical and kinematical refinement increases with sample thickness. Therefore, Fig. A5 displays the partial charges of four individual histidine crystals with varying thickness. Although the absolute thickness of these crystals is unknown, the transparency to the underlying lacey carbon, and the crystal thickness, varies between these samples. Within the range of selected crystal size the plot of partial charges in Fig. A5a) does not indicate systematic errors related to the crystal thickness, although the variations do underline once more the beneficial effect of merging data from several crystals for more reliable results [31].

With crystals for electron diffraction, we could not measure the thickness of the samples, nor could we preselect the samples, before inserting the TEM grid into the TEM, simply because these crystals are too small to be inspected with a light microscope. Unlike in X-ray diffraction, where crystals can be manipulated with needles or scalpels through a light microscope, crystals for ED cannot be modified, or re-oriented.

13) ESP map availability:

"ESP maps are no longer part of the manuscript, as they led to confusion amongst the referees." I am NOT happy with this approach of the authors. The ESP maps are highly relevant and should be included, suppressing them because they create "confusion" is not o.k.

Response: We returned the ESP maps into the manuscript as Fig. A13 (page 35) and include a detailed description of how they were generated. The section 2.6 "Comparison with quantum-mechanical calculations" refers to the maps:

Fig. A13 juxtaposes the experimental with the computed electrostatic potential maps for Tyrosine and Histidine. The comparison reflects the similarities, e.g. for the imidazole group in Histidine, but also highlights some differences, e.g. for O2 of the carboxylate group in Histidine, which is involved in the N1–H1···O2 hydrogen bond discussed in Sec. 2.3

We also included a section 3.6 “Generation of electrostatic potential maps” that describes how we produced these maps.

Revision 4 — Author response letter

Tim Gruene with all co-authors

12th May 2025

Dear Dr. Editor,

we were very pleased to read the referee's comments. Following the suggestion, we re-computed the QM ESP maps for tyrosine and histidine (extended Fig. 13, page 36) with a simulated crystallographic environment. It turns out that the QM map is amazingly similar to the experiments ESP derived from iSFAC modelling. We provide more comments in the response to the reviewer. We greatly appreciate the constructive help which led to a substantial improvement of our manuscript. The correction in the caption of Fig. 3, shown in the PDF `diff_rev3_rev4.pdf` was only as slip, the example explanation of the short error notation "0.15(6)" referred to the tyrosine in Fig. 4, and is now properly exemplified with the C11 of ciprofloxacin, shown in Fig. 3.

We look forward to hear whether we could satisfy all concerns regarding our manuscript on iSFAC modelling.

Very best regards, Tim Gruene

1 Referees' comments:

Referee #2 (Remarks to the Author):

Tim Gruene and co-authors have made a major effort that includes new experiments to address the remaining concerns of the unconvinced referee. As my own concerns were already satisfactorily addressed earlier, I will only comment on this debate, referring to discussion points in the order in which they are discussed in the rebuttal letter. I am satisfied with the authors' responses, with the exception of point 3.

1) investigation of the low resolution cut-off:

The authors have characterized different polymorphs of histidine, with similar and different chemical environments for atoms of interest compared to the original crystal form. The results of the partial charge analysis suggest that the partial charge is dependent on the chemical environment, but roughly preserved when only the cell constants change. In my mind, this result essentially eliminates the low resolution concern and resolves the matter satisfactorily.

2) validating extracted charges:

The response to 1) addresses this concern.

3) ESP maps as the primary experimental result:

On this issue, I am not fully satisfied with the authors' answer. The authors argue that differences between calculated ESP maps and ESP map derived from experimental partial charges stem from an inappropriate comparison between calculated ESP maps for the isolated molecule, and experimental ESP maps for the molecule in the crystalline context. The problem could easily be avoided by generating a calculated ESP map for the molecule in its crystalline context. This map should agree well with the map based on experimental partial charges. The insistence of the other referee on this point in the previous round is justified in my eyes.

Response: Motivated by the referee's expectation, we tried again and created a small crystallographic environment with a central molecule and all directly adjacent molecules. We did this for both histidine and tyrosine and computed the electrostatic potential map for this composition. It turns out the referee is right: In contrast to the ESP map calculated on an isolated molecule, these computed ESP maps (QM-ESP) are strikingly similar to the experimental ESP maps (E-ESP). The improved maps are shown in extended Fig. 13 a-f on page 36, including the "crystallographic environment" (c/f). The figure is referred to at the end of section 2.7, pages 12/13:

Partial charges are the direct result from iSFAC modelling. Together with the phases computed from the crystallographic model, Fourier transformation of the electron diffraction data produces a map that corresponds to the electrostatic potential map (ESP). Quantum chemical calculations calculate the electron density map first, and the partial charges are derived second (cf. Sec. 3.5). Fig. A13 juxtaposes the experimental with the computed electrostatic potential maps for tyrosine and histidine. In order to mimic the chemical environment of the crystal lattice, the central molecule together with its direct neighbours were extracted from the crystallographic model and used for the computation of the computational electrostatic potential, indicated in Fig. A13(c) and (f). Both for tyrosine (Fig. A13(a)/(b)) and histidine (Fig. A13(d) and (e)), the carboxylate group shows positive patches induced by hydrogen bonds. This is also reflected by the hydroxyl group of tyrosine. The respective proton are highlighted in Fig. A13(c) and (f). The strong match of the experimental ESP from iSFAC modelling with the computational ESP is underlined by strong Pearson correlation coefficient between the QM-ESP and the experimental ESP are 69.0 % for tyrosine and 71.4 % for histidine respectively.

Originally, we attempted to mimic the crystallographic environment with a $3 \times 3 \times 1$ unit cell for histidine, which forms approximately a cubic volume. This composition, however, crashed after 10 days of computing time on a dual-CPU AMD EPYC with 48 cores and 756 GB RAM. We are very grateful for the referee's insistence on this topic, and we are positively surprised

that such a small molecular environment, consisting only of direct neighbours, is sufficient for such good agreement between the experimental maps and the quantum chemical maps.

4) Comparison with X-ray diffraction:

I did not expect unstable refinement when trying to model partial charges based on X-ray data. However, this observation makes the point about the superiority of electron diffraction for partial charge determination even more strongly than a comparison of partial charges obtained from X-ray diffraction and electron diffraction could. Hence, I am satisfied with the authors' answer.

5) Hydrogen:

It now seems that the discussion of this point resulted from a misunderstanding. I would regard the issue as resolved.

6) Literature for ionic scattering curves, definition of form factors for cationic and anionic species:

It is very surprising that only the Yonekura approach for modeling partial charges "works". As we now know, the authors, the other referee, and myself would all have expected otherwise. If I understand correctly, the authors have no explanation for this unexpected finding. If an explanation could be found, it would probably greatly contribute to the understanding of experimentally determined partial charges. However, this should not be a condition of publication, and I accept the authors' response.

7) Enhancement of hydrogen visibility:

This issue was already resolved in the previous round.

8) Least squares refinement of complex molecules:

The authors' response on this point is somewhat vague, but I suggest to accept it.

9) Optimal scattering curves:

There is an element of arbitrariness in the choice of scattering curves. Fig. A7 makes the point that differences between reasonable choices are not minor, especially at low resolution. I stick with my prior point that there are many valid ways to define partial charges, and that any chemically plausible choice is acceptable, provided that it is stable. In this sense, I find the authors' answer satisfactory.

10) Rscat values:

This issue has been satisfactorily addressed in the previous round.

11) Influence of B-factors:

This issue has been satisfactorily addressed in the previous round.

12) Multiple scattering:

This authors have addressed this point experimentally, by measuring crystals of various thickness. The results do not suggest a systematic dependence of partial charges on crystal thickness. In my eyes, the additional experiments fully resolve the matter.

13) ESP map availability:

The reintroduction of the maps resolves the issue satisfactorily.

Referee #2 (Remarks on code availability):

This issue was resolved in previous refereeing rounds.